# Global soil organic carbon removal by water erosion under climate change and land use change during 1850-2005 AD

Victoria Naipal[1], Philippe Ciais[1], Yilong Wang[1], Ronny Lauerwald[2, 3], Bertrand Guenet[1], Kristof Van Oost[4]

[1]Laboratoire des Sciences du Climat et de l'Environnement, CEA CNRS UVSQ, Gif-sur-Yvette 91191, France
[2]Department of Geoscience, Environment & Society, Université Libre de Bruxelles, Brussels, Belgium
[3]Department of Mathematics, College of Engineering, Mathematics and Physical Sciences, University of Exeter, Exeter, UK
[4]Université catholique de Louvain,TECLIM - Georges Lemaître Centre for Earth and Climate Research, Louvain-la-Neuve, Belgium

*Correspondence to*: Victoria Naipal (victoria.naipal@lsce.ipsl.fr)

## Abstract

Erosion is an Earth System process that transports carbon laterally across the land surface, and is currently accelerated by anthropogenic activities. Anthropogenic land cover change has accelerated soil erosion rates by rainfall and runoff substantially, mobilizing vast quantities of soil organic carbon (SOC) globally. At timescales of decennia to millennia this mobilized SOC can significantly alter previously estimated carbon emissions from land use change (LUC). However, a full understanding of the impact of erosion on land-atmosphere carbon exchange is still missing. The aim of this study is to better constrain the terrestrial carbon fluxes by developing methods compatible with Land Surface Models (LSMs) in order to explicitly represent the links between soil erosion by rainfall and runoff and carbon dynamics. For this we use an emulator that represents the carbon cycle of a LSM, in combination with the Revised Universal Soil Loss Equation model. We applied this modeling framework at the global scale to evaluate the effects of potential soil erosion (soil removal only) in the presence of other perturbations of the carbon cycle: elevated atmospheric $CO_2$, climate variability, and LUC. We find that over the period 1850-2005 AD acceleration of soil erosion leads to a total potential SOC removal flux of 74±18 Pg C of which 79-85% occurs on agricultural-and grassland. Using our best estimates for soil erosion we find that including soil erosion in the SOC-dynamics scheme results in an increase of 62% of the cumulative loss of SOC over 1850 – 2005 due to the combined effects of climate variability, increasing atmospheric $CO_2$ and LUC. This additional erosional loss decreases the cumulative global carbon sink on land by 2 Pg of carbon for this specific period, with the largest effects found for the tropics, where deforestation and agricultural expansion increased soil erosion rates significantly. We conclude that the potential effect of soil erosion on the global SOC stock is comparable to the

effects of climate or LUC on the carbon cycle. It is thus necessary to include soil erosion in assessments of LUC and evaluations of the terrestrial carbon cycle.

## 1 Introduction

Carbon emissions from land use change (LUC), recently estimated as $1.0\pm0.5$ Pg C yr$^{-1}$, form the second largest anthropogenic source of atmospheric $CO_2$ (Le Quéré *et al.*, 2016). However, their uncertainty range is large, making it difficult to constrain the net land-atmosphere carbon fluxes and reduce the biases in the global carbon budget (Goll *et al.,* 2017; Houghton and Nassikas, 2017; Le Quéré *et al.,* 2016). The absence of soil erosion in assessments of LUC is an important part of this uncertainty, as soil erosion is strongly connected to LUC (Van Oost *et al.*, 2012;

Wang *et al.*, 2017).

The expansion of agriculture has accelerated soil erosion by rainfall and runoff significantly, mobilizing around 783 ± 243 Pg of soil organic carbon (SOC) globally over the past 8000 years (Wang *et al*., 2017). Most of the mobilized SOC is redeposited in alluvial and colluvial soils, where it is stabilized and buried for decades to millennia (Hoffmann *et al.*, 2013a; Hoffmann *et al.*, 2013b; Wang *et al.*, 2017). Together with dynamic replacement of

removed SOC by new litter input at the eroding sites, and the progressive exposure of carbon-poor deep soils, this translocated and buried SOC can lead to a net carbon sink at the catchment scale, potentially offsetting a large part of the carbon emissions from LUC (Berhe *et al.*, 2007; Bouchoms *et al.*, 2017; Harden *et al.*, 1999; Hoffmann *et al*., 2013a; Lal, 2003; Stallard, 1998; Wang *et al.*, 2017).

On eroding sites, soil erosion keeps the SOC stocks below a steady-state (Van Oost *et al.,* 2012) and can lead to

substantial $CO_2$ emissions in certain regions (Billings *et al.*, 2010; Worrall *et al*., 2016; Lal, 2003). $CO_2$ emission from soil erosion can take place during the breakdown of soil aggregates by erosion and during the transport of the eroded SOC by runoff and later also by streams and rivers.

LUC emissions are usually quantified using bookkeeping models and LSMs that represent the impacts of LUC activities on the terrestrial carbon cycle (Le Quere *et al*., 2016) only through processes leading to a local imbalance

between NPP and heterotrophic respiration, ignoring lateral displacement. Currently, LSMs consider only the carbon fluxes following LUC resulting from changes in vegetation, soil carbon and sometimes wood products (Van Oost *et al.*, 2012; Stocker *et al.*, 2014). The additional carbon fluxes associated with the human action of LUC from the removal and lateral transport of SOC by erosion are largely ignored.

In addition, the absence of lateral SOC transport by erosion in LSMs complicates the quantification of the human

perturbation of the carbon flux from land to inland waters (Regnier *et al*., 2013). Recent studies have been investigating the Dissolved Organic Carbon (DOC) transfers along the terrestrial-aquatic continuum in order to better quantify $CO_2$ evasion from inland waters and to constrain the lateral carbon flux from the land to the ocean (Lauerwald *et al*., 2017; Regnier *et al.*, 2013). They point out that an explicit representation of soil erosion and transport of particulate organic matter (POC) – in addition to DOC leaching and transport - in future LSMs is

essential to be able to better constrain the flux from land to ocean. This is true, since the transfer of particulate organic carbon from eroded SOC also matters for estimating carbon inputs to rivers.

The slow pace of carbon sequestration by soil erosion and deposition (Van Oost *et al.*, 2012; Wang *et al.*, 2017) and the slowly decomposing SOC pools require the simulation of soil erosion at timescales longer than a few decades to fully quantify its impacts on the SOC dynamics. This, and the high spatial resolution that soil erosion models typically require, complicates the introduction of soil erosion and related processes in LSMs that use short time steps ($\approx 30$ min) for simulating energy fluxes and require intensive computing resources.

Previous approaches used to explicitly couple soil erosion and SOC turnover have been applying different erosion and carbon dynamic models at different spatial and temporal scales. Some studies coupled process-oriented soil erosion models with C turnover models calibrated for specific micro-catchments on timescales of a few decades to a millennium, (Billings *et al.*, 2010; Van Oost *et al.*, 2012; Nadeu *et al.*, 2015; Wang *et al.*, 2015a; Zhao *et al.*, 2015; Bouchoms *et al.*, 2017). Other studies focused on the application of parsimonious erosion-SOC dynamics models using the RUSLE approach together with sediment transport methods at regional or continental spatial scales (Chappell *et al.*, 2015; Lugato *et al.*, 2016; Yue *et al.*, 2016; Zhang *et al.*, 2014). However, the modeling approaches used in these studies apply erosion models that still require many variables and data input that is often not available at the global scale or for the past or the future time period. These models also run on a much higher spatial resolution than LSMs, making it difficult to integrate them with LSMs. The study of Ito (2007) was one of the first studies to couple water erosion to the carbon cycle at the global scale, using a simple modelling approach that combined the RUSLE model with a global ecosystem carbon cycle model. However, there are several unaddressed uncertainties related to his modelling approach, such as the application of the RUSLE at the global scale without adjusting its parameters.

Despite all the differences between the studies that coupled soil erosion to the carbon cycle, they all agree that soil erosion by rainfall and runoff is an essential component of the carbon cycle. Therefore, to better constrain the land-atmosphere and the land-ocean carbon fluxes, it is necessary to develop new LSM-compatible methods that explicitly represent the link between soil erosion and carbon dynamics at regional to global scales *and* over long timescales. Based on this, our study introduces a 4D modeling approach that consists of 1) an emulator that simulates the carbon dynamics like in the ORCHIDEE LSM (Krinner *et al.*, 2005), 2) the Revised Universal Soil Loss (Adj.RUSLE) model that has been adjusted to simulate global soil removal rates based on coarse resolution data input from climate models (Naipal *et al.*, 2015), and 3) a spatially explicit representation for LUC. This approach represents explicitly and consistently the links between the perturbation of the terrestrial carbon cycle by elevated atmospheric $CO_2$ and variability (temperature and precipitation change), the perturbation of the carbon cycle by LUC and the effect of soil erosion at the global scale.

The main goal of our study is to use this new modeling approach to determine the potential effects of long-term soil erosion by rainfall and runoff without deposition or transport on the global SOC stocks under LUC, climate variability and increasing atmospheric $CO_2$ levels. In order to be able to determine if global soil erosion is a net carbon source or sink, it is essential to study first how soil erosion, without deposition or transport, interacts with the terrestrial carbon cycle. Therefore, we also aim to understand the links between the different perturbations to the carbon cycle in the presence of soil erosion and to identify relevant changes in the spatial variability of SOC stocks under erosion.

## 2 Materials and methods

### 2.1 Modeling framework concept

We used the LSM ORCHIDEE-MICT (Guimberteau *et al.*, 2017; Zhu *et al.*, 2016) (in the following simply referred to as ORCHIDEE) to construct a carbon emulator that describes the carbon pools and fluxes exactly as in ORCHIDEE (Fig. 1A). MICT stands for aMeliorated Interactions between Carbon and Temperature, and this version of ORCHIDEE has several major modifications and improvements for especially the high-latitudes. ORCHIDEE has 8 biomass pools, 4 litter pools, of which 2 are above-ground and 2 are below-ground and 3 SOC pools for each land cover type (Fig. 1A). It has been extensively validated using observations on energy, water and carbon fluxes at various eddy-covariance sites, and with measurements of atmospheric $CO_2$ concentration (Piao *et al*., 2009). The land cover types are represented by 12 plant functional types (PFT's) and an additional type for bare soil. 10 PFT's represent natural vegetation and 2 represent agricultural land (C3 and C4 crop). The turnover times for each of the PFT-specific litter and SOC pools depend on their residence time modified by local soil texture, humidity, and temperature conditions (Krinner *et al.*, 2005). The loss of biomass and litter carbon by fire is represented by the parameterization of the Spitfire model from Thonicke *et al.* (2011) in the full ORCHIDEE model, and currently cannot be modified in our version of the emulator. Carbon losses by fire here are considered to contribute directly to the $CO_2$ emissions from land.

At face value, the emulator merely copies the ORCHIDEE carbon pool dynamics, and for each new atmospheric $CO_2$- and climate-scenario a new run of the original LSM is required to build the emulator. The emulator thus reproduces exactly the carbon pool dynamics of the full LSM. The change in carbon over time for each pool of the original model is represented in the emulator by the following general mass-balance approach:

$$\frac{dC}{dt} = I(t) - k * C(t) \tag{1}$$

Here, $\frac{dC}{dt}$ represents the change in carbon stock of a certain pool over time, calculated by the difference between the incoming flux ($I(t)$), and the outgoing flux ($k*C(t)$) to the respective pool, where $k$ is the turnover rate. Although originally calculated by complex equations, the dynamic evolution of each pool can be described using the first-order model of Eq. 1. Complex equations, such as photosynthesis and hydrological processes, are needed to simulate realistically the carbohydrates input to carbon pools and the moisture and temperature conditions controlling litter and soil carbon decomposition over time. All the processes that determine surface- and soil temperature and soil moisture, are calculated by the ORCHIDEE LSM on a 30 minute time-step. Such a time-step is needed for coupled simulations with a climate model, but makes the LSM model CPU intensive. However, there is no need for such hightemporal resolution calculations of 'fast' carbon and energy fluxes to account for erosion-induced effects on SOC stocks. The addition of erosion is here supposed to impact only carbon pools, and to have no feedbacks on soil moisture, soil temperature and photosynthesis. Therefore, we decided to use the emulator concept rather than incorporating erosion processes directly into ORCHIDEE. For each carbon pool the stock and all the incoming and outgoing fluxes are derived at a daily time step from a single simulation performed with the ORCHIDEE LSM.

Based on the daily output stock and fluxes, the values of the turnover rates are calculated and archived together with the input fluxes to build the emulator. Then, the emulator can be run to simulate the dynamics of all pools over long time scales without having to re-compute carbon fluxes at each time-step. In this way the emulator reduces the computation time of the complex ORCHIDEE model significantly and allows us to easily add and study erosion-related processes affecting the carbon dynamics of the soil. Our main objective here is to present a tool able to evaluate erosion-related carbon fluxes at global scale using a 'state-of-art' LSM output and to estimate the drivers of carbon erosion at the global scale.

ORCHIDEE also includes crop harvest, defined as the harvest of above-ground biomass of agricultural PFTs, and calculated based on the concept of the harvest index (Krinner *et al*., 2005). The harvest index is defined as the yield of crop expressed as a fraction of the total above-ground dry matter production (Hay, 1995). ORCHIDEE uses a fixed harvest index for crop of 0.45. However, Hay (1995) showed that the harvest index has increased significantly since 1900 for C3 crop such as wheat. In the emulator, and also in the full ORCHIDEE model, the carbon balance of agricultural lands is sensitive to crop harvest. Based on this we use the findings of Hay (1995) to change the harvest index of C3 crops to be temporally variable over the period 1850 – 2005 in the emulator, with values ranging between 0.26 and 0.46. This means that more crop biomass is harvested against what becomes litter. We only changed the HI of C3 plants, because Hay (1995) mentioned that C4 plants, such as maize, had already a high HI at the start of the last century. It should be noted that the harvest index does not vary spatially in our emulator, and harvesting is then done constantly at each time step.

## 2.2 Net land use change

Land use change is not taken into account in the ORCHIDEE LSM version we are using in this study to build the emulator, but is represented by a net-land use change routine in the emulator that includes past agricultural and grassland expansion over natural PFTs (Fig. 1B). This makes it possible to switch the LUC routine on or off in the emulator or to change LUC scenarios when needed without having to re-run ORCHIDEE. We verified that the LUC routine added to the emulator conserves the mass of all carbon pools for lands in transition to a new land use type. When LUC takes place, the fractions of PFTs in each grid cell are updated every year given prescribed annual maps of agricultural and natural PFTs (Peng *et al.*, 2017). The carbon stocks of the litter and SOC pools of all the shrinking PFTs are then summed and allocated proportionally to the expanding or new PFTs, maintaining the mass-balance (Houghton and Nassikas, 2017; Piao *et al.,* 2009). When natural vegetation is reduced by LUC, the heartwood and sapwood biomass pools are harvested and transformed to 3 wood products with turnover times of 1 year, 10 years and 100 years. The other biomass pools (leafs, roots, sapwood below-ground, fruits, heartwood belowground) are transformed to metabolic or structural litter and allocated to the respective litter pools of the expanding PFTs (Piao *et al*., 2009).

## 2.3 Soil carbon dynamics

The change in the carbon content of the PFT-specific SOC pools in the emulator without soil erosion can be described with the following differential equations:

$$\frac{dSOC_a(t)}{dt} = lit_a(t) + k_{pa} * SOC_p(t) + k_{sa} * SOC_s(t) - \left(k_{ap} + k_{as} + k0_a\right) * SOC_a(t) \tag{2}$$

$$\frac{dSOC_s(t)}{dt} = lit_s(t) + k_{as} * SOC_a(t) - \left(k_{sa} + k_{sp} + k0_s\right) * SOC_a(t) \tag{3}$$

$$\frac{dSOC_p(t)}{dt} = k_{ap} * SOC_a(t) + k_{sp} * SOC_s(t) - \left(k_{pa} + k0_p\right) * SOC_p(t) \tag{4}$$

Where $SOC_a$, $SOC_s$, and $SOC_p$ (g C m$^{-2}$) are the active (unprotected), slow (physically or chemically protected) and passive (biochemically recalcitrant) SOC, respectively. The SOC pools are based on the study of Parton *et al.* (1987) and are defined by their residence times. The active SOC pool has the lowest residence time (~1.5 years) and the passive the highest (~1000 years). $lit_a$ and $lit_s$ (g C m$^{-2}$ day$^{-1}$) are the litter input rates to the active and slow SOC pools, respectively; $k0_a$, $k0_s$ and $k0_p$ (day$^{-1}$) are the respiration rates of the active, slow and passive pools, respectively; $k_{as}$, $k_{ap}$ , $k_{pa}$, $k_{sa}$, $k_{sp}$ are the coefficients determining the flux from the active to the slow pool, from the active to the passive pool, from the passive to the active pool, from the slow to the active pool and from the slow to the passive pool, respectively (Fig. 1A).

The SOC pools are not vertically discretized in the version of the ORCHIDEE LSM used to build the emulator, so we implemented a simple vertical discretization scheme for the SOC pools in the emulator based on the concept of Wang *et al.* (2015a,b). In this scheme the carbon dynamics of each soil layer are calculated separately, based on layer-dependent litter input and respiration rates (Fig. 1A). The vertical discretization scheme of the emulator does not change the total input and respiration as simulated by ORCHIDEE in the case where erosion and land use change processes are switched off. We apply the same scheme for all three SOC pools, assuming that each SOC pool  is equally distributed across all layers of the soil profile, while  the ratios between the pools per soil layer are equal to those from ORCHIDEE. We base this assumption on the fact that there is very little information or data to constrain the pool ratios globally, mainly because the three SOC pools cannot be directly related to measurements (Elliott E.T., Paustian K., 1996). Furthermore, neither the emulator nor the ORCHIDEE LSM model we used include soil processes that may affect these pool ratios with depth, such as vertical mixing by soil organisms, diffusion, leaching, changes in soil texture (SOC protection and stabilization by clay particles), limitations by oxygen and by access to deep organic matter by microbes. It is also uncertain how sensitive SOC is to these processes. For example, the study of Huang *et al.* (in revision for the journal Advances in Modeling Earth Systems) who implemented a matric-based approach to assess the sensitivity of SOC showed that equilibrium SOC stocks are more sensitive to input than to mixing for soils in the temperate and high-latitude regions.

In the vertical discretization scheme of the emulator, the soil profile is divided into thin layers of 1 cm thickness down to a depth of 2 m, which is the soil depth used by ORCHIDEE to calculate SOC. The first 10 cm of the soil profile are referred to as the "topsoil", where we assume that the SOC content is homogeneously distributed. The rest of the soil profile is referred to as the subsoil. The topsoil receives carbon from above- and below-ground litter, and each of the soil layers in the topsoil receives an equal fraction of both litter types.

The below-ground litter input for the active SOC pool is the sum of a fraction of the below-ground structural and metabolic litter pools of ORCHIDEE being re-calculated by the emulator, while the below-ground litter input for

slow SOC pool is equal to a fraction of the below-ground structural litter pool only. This setting is consistent with the structure of the SOC module of the ORCHIDEE LSM to ensure that the emulator reproduces the same C pool dynamics than the LSM. The litter fractions are based on the Century model as introduced by Parton *et al.* (1987) and later implemented inside ORCHIDEE (Krinner *et al.*, 2005). We assume that the subsoil receives carbon only from below-ground litter, and that this input decreases exponentially with depth following the root profile of ORCHIDEE. This discretization of the total below-ground litter input ($lit_{be}$) is the same for both SOC pools and can then be represented as:

$$lit_{be} = \int_{z=0}^{z=z_{max}} I_{0be} * e^{-r*z} \, dz \tag{5}$$

where $I_{0be}$ is the below-ground litter input to the surface layer, and is equal to:

$$I_{0be} = lit_{be} * \frac{r}{1-e^{-r*z_{max}}} \tag{6}$$

The homogeneously distributed below-ground litter input ($I_{be}$) to the layers of the topsoil is equal to:

$$\frac{\sum_{z=0}^{z=10} I_{be0} * e^{-r*z}}{0.1} \tag{7a}$$

The below-ground litter input to the layers of the subsoil is equal to:

$$I_{be}(z) = I_{be0} * e^{-r*z} \tag{7b}$$

where $z_{max}$ is the maximum soil depth equal to 2 m, and $dz$ is the soil layer discretization (1 cm); $r$ is the PFT-specific vertical root-density attenuation coefficient as used in ORCHIDEE.

The SOC respiration rates for the topsoil layers are equal to those from ORCHIDEE and are determined by average soil temperature, moisture and texture. For the rest of the soil profile the respiration rates of all three SOC pools decrease exponentially with depth:

$$k_i(z) = k_{0\,i} * e^{-r_e z} \tag{8}$$

Here $k_{0\,i}$ is the SOC respiration rate at the surface layer for each SOC pool (i = a, s, p), and $r_e$ (m$^{-1}$) is a coefficient representing the impact of external factors, such as oxygen availability, which reduce SOC mineralization rate with depth ($z$). To ensure that the total soil respiration of the emulator is similar to that of the ORCHIDEE LSM model for each grid cell, each PFT, and each SOC pool, we have calibrated the exponent '*re*' and variable '$k_{0i}$' of equation (8) for each grid cell and PFT under equilibrium conditions. First we selected a default value for '*re*' between 0 and 5, and calculated the respiration rate of the surface soil layer ($k_0$) when all SOC pools are in an equilibrium state, with the following equation:

$$SOC_{orchidee} = \sum_{z=0}^{z=n} \frac{L(z)}{k0*e^{-re*z}} \tag{9}$$

Where, $SOC_{orchidee}$ is the total equilibrium SOC stock derived from ORCHIDEE for a certain grid cell and PFT. *L(z)* is the total litter input to the soil for a certain soil layer discretized according to the root profile. Then we derived the equilibrium SOC stocks per soil layer as:

$$SOC(z) = \frac{L(z)}{k0*e^{-re*z}} \tag{10}$$

Assuming that the ratios between the active, slow and passive SOC pools do not change with depth and are equal to the ratios derived from ORCHIDEE, we calculated the SOC stocks of each pool with the following equation:

$$1 + \frac{soil_s(z)}{soil_a(z)} + \frac{soil_p(z)}{soil_a(z)} = \frac{SOC(z)}{soil_a(z)} \tag{11}$$

Where, $soil_a(z)$, $soil_s(z)$, $soil_p(z)$ are the emulator derived active, slow and passive SOC stock per soil layer, grid cell and PFT. Now, for the equilibrium state the input is equal to the output, allowing us to derive $k_{0a}$, $k_{0s}$ and $k_{0p}$ with the following equations:

$$\sum_{z=0}^{z=n}\left(\frac{L_a(z)+k_{sa}*soil_s(z)+k_{pa}*soil_p(z)}{k_{0a}*e^{-re*z}+k_{as}+k_{ap}}\right) = SOC_a \tag{12a}$$

$$\sum_{z=0}^{z=n}\left(\frac{L_s(z)+k_{as}*soil_a(z)}{k_{0s}*e^{-re*z}+k_{sa}+k_{sp}}\right) = SOC_s \tag{12b}$$

$$\sum_{z=0}^{z=n}\left(\frac{k_{sp}*soil_s(z)+k_{ap}*soil_a(z)}{k_{0p}*e^{-re*z}+k_{pa}}\right) = SOC_p \tag{12c}$$

Where, $L_a$ is the total litter input to the active SOC pool, $L_s$ is the total litter input to the slow SOC pool. $SOC_a$, $SOC_s$, $SOC_p$ are the total active, slow and passive SOC stocks per grid cell and PFT, respectively, derived from ORCHIDEE. $k_{as}$, $k_{ap}$, $k_{sa}$, $k_{sp}$, $k_{pa}$ are the coefficients determining the fluxes between the SOC pools. After the derivation of '$k_{0i}$' we tested if the difference in the SOC stocks between the emulator and the original ORCHIDEE LSM is less than 1 g m$^{-2}$ per grid cell and PFT. If this was not the case we increased or decreased the value of '$re$' and repeated the calibration cycle. If we did not find an optimized value for both '$re$' and '$k_{0i}$' that meet this criteria, we used values that minimized the difference in SOC stocks between the emulator and the original ORCHIDEE LSM.

For the transient period (without LUC or erosion) we assumed a time-constant '$re$', where the values are equal to those at equilibrium. Using the mass-balance approach we calculated the daily values for $k_{0a}$, $k_{0s}$, $k_{0p}$ per grid cell and PFT with:

$$\frac{dSOC_a}{dt} = \sum_{z=0}^{z=n}(L_a(z,t)+k_{sa}*soil_s(z,t-1)+k_{pa}*soil_p(z,t-1)-\left(k_{0a}(t)*e^{-re*z}+k_{as}+k_{ap}\right)*soil_a(z,t-1)) \tag{13a}$$

$$\frac{dSOC_s}{dt} = \sum_{z=0}^{z=n}(L_s(z,t)+k_{as}*soil_a(z,t-1)-\left(k_{0s}(t)*e^{-re*z}+k_{sa}+k_{sp}\right)*soil_s(z,t-1)) \tag{13b}$$

$$\frac{dSOC_p}{dt} = \sum_{z=0}^{z=n}(k_{sp}*soil_s(z,t-1)+k_{ap}*soil_a(z,t-1)-\left(k_{0p}(t)*e^{-re*z}+k_{pa}\right)*soil_p(z,t-1)) \tag{13c}$$

In case there was no solution for the '$k_{0i}$' at a certain time-step we took the values from the previous time step.

The annual average soil erosion rate (E, t ha$^{-1}$ year$^{-1}$) is calculated by the Adj.RUSLE (Naipal et al., 2015; Naipal et al., 2016) according to:

$$E = S * R * K * C \tag{14}$$

Where $R$ is the rainfall erosivity factor (MJ mm ha$^{-1}$ h$^{-1}$ year$^{-1}$), $K$ is the soil erodibility factor (t ha h ha$^{-1}$ MJ$^{-1}$ mm$^{-1}$), $C$ is the land cover factor (dimensionless), $S$ is the slope steepness factor (dimensionless). The S-factor is calculated using the slope from a 1km resolution digital elevation model (DEM) that has been scaled using the fractal method to a resolution of 150m (Naipal *et al.,* 2015). In this way the spatial variability of a high-resolution slope dataset can be captured. The computation of the R factor has been adjusted to use coarse resolution input data on precipitation and to provide reasonable global erosivity values. For this Naipal *et al.* (2015) derived regression equations for different climate zones of the Köppen–Geiger climate classification (Peel *et al.*, 2007). The results from the Adj.RUSLE model have been tested against empirical large-scale assessments of soil erosion and rainfall erosivity (Naipal *et al.,* 2015, 2016). The original RUSLE model as described by Renard *et al.* (1997) also includes the slope-

length (L) and support-practice (P) factors. Although these factors can strongly affect soil erosion in certain regions, the Adj.RUSLE does not include these factors due to several reasons. Firstly, (Doetterl *et al*., 2012) showed that these factors do not significantly contribute to the variation in soil erosion at the continental to global scales, in comparison to the other RUSLE factors. Secondly, data on the L and P factors and methods to estimate them at the global scale are very limited. Thus, including them in global soil erosion estimations would result in large uncertainties. Finally, the focus of this study is to show the effects of potential soil erosion on the terrestrial carbon cycle, without the explicit effect of management practices such as covered by the P-factor. For more information on the validation of our erosivity values and a more detailed description of the calculation of each of the RUSLE factors see supporting material section S1.

The Adj. RUSLE model is not imbedded in the C emulator but is run separately on a 5arcmin spatial resolution and at a yearly timestep. The resulting soil erosion rates are then read by the C emulator at each time step and used to calculate the daily SOC erosion rate of a certain SOC pool $i$ ($Ce_i$ in g C m$^{-2}$ day$^{-1}$) at the surface layer by:

$$Ce_i = SOC_i * \frac{\frac{E}{365}*100}{BD_{top}*dz*10^6} \tag{15}$$

where BD$_{top}$ is the bulk density of the surface layer (g cm$^{-3}$). We assume that the enrichment ratio, i.e. the volume ratio of the carbon content in the eroded soil to that of the source soil material, is equal to 1 here, which implies that our estimates of SOC mobilization are likely conservative (Chappell *et al*., 2015; Nadeu *et al.*, 2015).

When erosion takes place, the surface layer is truncated by the erosion height, and at the same time an amount of SOC corresponding to this erosion height is removed. As we assume that the soil layer thickness does not change, part of the SOC of the next soil layer is allocated to the surface layer proportional to the erosion height and the SOC concentration (per volume) of the next layer. In this way, the SOC from all the following soil layers move upward and become exposed to erosion in the surface layer at some point in time (Fig. 1A). To preserve mass balance, we assume that there is no SOC below the 2 m soil profile represented in the emulator and new substrate replacing the material of the last soil layer is SOC free, so that SOC in the bottom layer will decrease towards zero after erosion has started.

### 2.4 Input datasets

### 2.4.1 for ORCHIDEE

We used 6-hourly climate data supplied by CRU-NCEP (version 5.3.2) global database (https://crudata.uea.ac.uk/cru/data/ncep/) available at 0.5 degree resolution to perform simulations with the full ORCHIDEE model for constructing the emulator. CRU-NCEP climate data was only available for the period 1901-2012. To be able to run ORCHIDEE for the period 1850-1900, we randomly projected the climate forcing after 1900 to the years before 1900. The random projection of the climate data is necessary to avoid the risk of including the effects of extreme climate conditions multiple times when only a certain decade is used repeatedly.

The historical changes of PFT fractions were derived from the historical annual PFT maps of Peng *et al*. (2017). These PFT maps were available at a resolution of 0.25 degrees (Fig. 2), and were re-gridded to the resolution of the ORCHIDEE emulator, which is 2.5 x 3.75 degrees, using the nearest neighbor approach.

**2.4.2 for the Adj.RUSLE**

Due to the resolution of the Adj.RUSLE, which is 5 arcmin (~0.0833 degree), all the RUSLE factors had to be regridded or calculated at this specific resolution before calculating the soil erosion rates.

The land cover fractions from the historical 0.25 degree PFT maps were used in combination with the LAI values from ORCHIDEE at the resolution of 2.5° x 3.75° to derive the values for the C-factor of the RUSLE model. We first regridded the yearly average LAI to the resolution of the PFT maps before calculating the land cover factor of RUSLE (C-factor) at the resolution of 0.25 degree. The C values were then regridded using the nearest neighbor method to the resolution of the Adj.RUSLE model. We used the nearest neighbor approach here, because the C-

factor is strongly dependent on the land cover class.

Daily precipitation data for the period 1850-2005 to calculate soil erosion rates is derived from the Inter-Sectoral Impact Model Intercomparison Project (ISIMIP), product ISIMIP2b (Frieler *et al*., 2017). This data is based on model output of the Coupled Model Intercomparison Project Phase 5 (CMIP5 output of IPSL-CM5A-LR (Taylor *et al*., 2012), which are bias corrected using observational datasets and the method of Hempel *et al.* (2013), and made

available at a resolution of 0.5 degrees (Fig. 2). We chose this data as input to the Adj.RUSLE model, because the dataset extended to 1850, in contrast to the CRU-NCEP data. Also, this dataset being bias-corrected, provides a better distribution of extreme events and frequencies of dry and wet days (Frieler *et al*., 2017), which is important for the calculation of rainfall erosivity (R factor). The ISIMIP precipitation data was regridded using the bilinear interpolation method to the resolution of the Adj.RUSLE model, before being used to calculate the R-factor. This

was necessary because the erosivity equations from the Adj.RUSLE model are calibrated at this specific resolution (Naipal *et al.*, 2015).

Data on soil bulk density and other soil parameters to calculate the soil erodibility factor (K), available at the resolution of 1 km, have been taken from the Global Soil Dataset for use in Earth System Models (GSDE) (Shangguan *et al*, 2014). The K factor has been calculated at the resolution of 1 km before being regridded to 5

arcmin using the bilinear interpolation method. We also used the SOC concentration in the soil from GSDE, which was derived using the "aggregating first" approach, to compare to our SOC stocks from simulations with the emulator. Finally, the slope steepness factor (S), which was originally estimated at the resolution of 1 km, was also regridded to the resolution of 5 arcmin using the bilinear interpolation method.

Using the above-mentioned data, soil erosion rates were first calculated at the resolution of 5 arcmin, and afterwards

aggregated to the coarse resolution of the emulator (2.5° x 3.75° ) to calculate daily SOC erosion rates.

**2.5 Model simulations**

To be able to understand and estimate the different direct and indirect effects of soil erosion on the SOC dynamics, we propose a factorial simulation framework (Fig. 3 and Table 1). This framework allows isolating or combining the main processes that link soil erosion to the SOC pool, namely the influence from climate variability, LUC, and atmospheric $CO_2$ increase. The different model simulations described in this section will be based on this framework.

We performed two different simulations with the full ORCHIDEE model to produce the required data input for the emulator for the period 1850-2005. For this we first performed a spinup with ORCHIDEE to get stead-state carbon pools for the year 1850. We chose the period 1850-2005 based on the ISIMIP2b precipitation data availability and the fact that this period underwent a significant intensification in agriculture globally and a substantial rise in atmospheric $CO_2$ concentrations. In the first simulation of ORCHIDEE the global atmospheric $CO_2$ concentration was fixed to the year 1850 to calculate time-varying NPP not impacted by CO2 fertilization and subsequent carbon pools, while in the second simulation the atmospheric $CO_2$ concentration was made variable. In both simulations, climate is variable from CRU-NCEP (Fig. 3).

Furthermore, we performed 7 simulations with the Adj.RUSLE model to pre-calculate the soil erosion rates that will be used as input to the ORCHIDEE emulator. Three of the 7 erosion simulations used best estimates for each model parameter, and the rest used either the minimum or maximum values for the R- and C-factors to derive an uncertainty range for our soil erosion rates and to analyze the sensitivity of the emulator. In the first simulation with the best estimated model parameters we kept the climate and land cover variable through time (the "CC+LUC" simulation). In the second simulation we only varied the climate through time and kept land cover fractions fixed to 1850 (the "CC" simulation, Fig. 3). In the third simulation we only varied the land cover through time and kept the climate constant to the average cyclic variability of the period 1850-1859 (the "LUC" simulation, Fig. 3).The erosion simulations with either minimum or maximum model parameters were either a "CC+LUC" or a "CC" type of simulation.

From the two simulations of ORCHIDEE (with variable and constant $CO_2$) and the 7 soil erosion simulations of the Adj.RUSLE, we constructed 4 versions of the emulator to perform 8 main simulations and 4 sensitivity simulations. The different simulations and their description are given in table 1 and figure 3. In the simulations without LUC (S2, S4, S6 and S8), the PFT fractions and the harvest index are constant and equal to those in the year 1850. In the simulations with LUC (S1, S3, S5 and S7) the harvest index increases and the PFT fraction change with time during 1850 - 2005. In each emulator-simulation we first calculated the equilibrium carbon stocks analytically before calculating the change of the carbon stocks in time depending on the perturbations during the transient period (1851-2005). In simulations with erosion, the equilibrium state of the SOC pools has been calculated using the average erosion rates of the period 1850-1859, assuming erosion to be constant before 1850 and a steady state condition where erosion fluxes are equal to new input from litter.

## 3 Results

### 3.1 Erosion versus no erosion

After including soil erosion in the ORCHIDEE emulator we obtain a total global soil loss flux of 47.6±10 Pg C y$^{-1}$ for the year 2005 of which 20 to 29% is attributed to agricultural land and 51 to 55% to grassland. This global soil loss flux (here 'loss' meaning horizontal removal of SOC by erosion) leads to a total SOC loss flux of 0.52±0.14 Pg C y$^{-1}$ of which 26 to 33% are attributed to agricultural land and 54 to 64% to grassland (CTR, Fig 4). Grassland and agricultural land thus have much larger annual average soil and SOC erosion rates compared to forest (Table 2).

The total soil and SOC losses in the year 2005 show an increase of 11-19% and 23-35%, respectively, compared to 1850 (CTR, Fig. 4) with the largest increases found in the tropics (Fig. 5B, D). The largest increase in soil and SOC erosion during 1850 – 2005 is found in South-America (Table 3) despite of the significant decreases in simulated precipitation leading to less intense erosion rates in this region. One should keep in mind that due to uncertainties in the simulated LUC and climate variability for certain regions and the assumptions made in our modeling framework, these trends in soil and SOC erosion rates are linked to some uncertainty. However, it is difficult to assess this uncertainty, mainly due to the lack of observations for the past in regions such as the tropics and the lack of model-testing in these regions.

We found that the total soil erosion flux on agricultural land increased by 55-58% in the year 2005 compared to 1850, while the SOC erosion flux increased by only 11-70% (Fig. 4) and led to a cumulative SOC removal of 22±5 Pg C (CTR). On grassland the soil erosion flux increased only with 8-20%, while the SOC erosion flux increased with 44-54% (Fig. 4) and led to a cumulative SOC mobilization of 38±7 Pg C since 1850. It is evident that on agricultural land the uncertainty range of soil erosion leads to a large uncertainty range in SOC erosion compared to grassland. The increase in SOC erosion is much larger than the increase in soil erosion for grasslands because in our model, LUC (without erosion) leads to a significant increase in SOC on grassland amplifying the increasing trend in SOC erosion for grassland. This simulated increase in SOC stocks on grasslands after LUC is not unrealistic, as it is observed from paired chronosequences worldwide where grasslands have higher SOC densities than forests for instance (Li *et al.*, 2017).

In total 7183±1662 Pg of soil and 74±18 Pg of SOC is mobilized across all PFTs by erosion during the period 1850 - 2005, which is equal to approximately 46-74% of the total net flux of carbon lost as $CO_2$ to the atmosphere due to LUC (net LUC flux) over the same period estimated by our study (S1-S2). In this study, we do not address the fate of this large amount of eroded SOC, be it partly sequestered (Wang *et al*., 2017) or released to the atmosphere as $CO_2$.

**3.2 Validation of model results**

We calculated a total global SOC stock for 2005 in the absence of soil erosion (S3) of 1284 Pg C, which is a factor of 0.73 lower than the total SOC stock from GSCE (Shangguan *et al*., 2014) for a soil depth of 2m (Table 5). SOC stocks of forest in our model contribute the most to this overestimation. Including soil erosion (S1, minimum, maximum) leads to a total SOC stock of 1001±58 Pg C for the year 2005 (Table 5). We also find that including soil erosion in the SOC-dynamics scheme slightly improves the root mean square error (RMSE) between the simulated

SOC stocks and those from GSDE for the top 30cm of the soil profile. This improvement in the RMSE occurs especially in highly erosive regions.. Furthermore, the total SOC stock of agricultural land is significantly lower than that of the GSDE, because we assume a steady-state landscape at 1850, where soil erosion losses are equal to the carbon input to the soil. We did not perform a more in-depth comparison with SOC global observations as our emulator and the original ORCHIDEE LSM do not include various soil processes that have been proven to affect SOC substantially such as vertical mixing, diffusion, priming, changes in soil texture, carbon rich organic soils formation, etc. The ORCHIDEE LSM model we use to build the emulator also lacks processes such as nitrogen and phosphorus limitations and priming, which affect the productivity and SOC decomposition (Goll *et al.*, 2017; Guenet *et al.*, 2016). The emulator also simulates only the removal of SOC but not the subsequent SOC transport and deposition after erosion, and there is a general uncertainty in the simulation of underlying processes that govern the SOC dynamics (Todd-Brown *et al.*, 2014). Finally, large uncertainties in the global soil databases (Hengl *et al.*, 2014; Scharlemann *et al.*, 2014; Tifafi *et al.*, 2018), complicate the exact quantification of the uncertainties of the resulting SOC dynamics simulated by our emulator.

Using the Adj.RUSLE model to estimate agricultural soil loss by water erosion for the year 2005 resulted in a global soil loss flux of $12.28\pm4.62$ Pg $y^{-1}$ (Fig. 4). This flux is paralleled by a SOC loss flux of $0.16\pm0.06$ Pg C $y^{-1}$ after including soil erosion in the CTR simulation (Fig. 4). This soil loss flux is in the same order of magnitude as earlier high-resolution assessments of this flux, while the SOC removal flux is slightly lower compared to previously published high-resolution estimates, but within the uncertainty (Table 6). We also find a fair agreement between our model estimates of recent agricultural soil and SOC erosion fluxes per continent and the high-resolution estimates (excluding tillage erosion) from the study of Doetterl *et al.* (2012) (Table 7). However, the continental SOC erosion fluxes from our study are generally lower, because of the lower SOC stocks on agricultural land. Only South-America shows a higher SOC flux for present-day compared to the high-resolution estimates of Doetterl *et al.* (2012), which is the result of the simulated high productivity of crops in the tropics.

Furthermore, we find a cumulative soil loss of $1888\pm753$ Pg and cumulative SOC removal flux of $22\pm5$ Pg C from agricultural land over the entire time period (CTR simulation). This soil loss flux lies in the range of $2480\pm720$ Pg found by Wang *et al.* (2017) for the same time period, while the SOC removal flux is significantly lower than the $63 \pm 19$ Pg C found by Wang *et al.* (2017) . Wang *et al.* (2017) used only recent climate data in his study while we explicitly include the effects of changes in precipitation and temperature on global soil erosion rates and the SOC stocks in our study, which may explain this difference.

## 4 Discussions

### 4.1 Significance of including soil erosion in the ORCHIDEE emulator

To estimate the net effect of soil erosion on the global SOC stocks under all perturbations we compare the cumulative SOC stock change from simulation S3 (no erosion; Table 1) with that of the CTR simulation with all factors included, that is, land use, $CO_2$, and climate. When considering our best estimated soil erosion rates and

assuming that the SOC mobilized by soil erosion in the CTR simulation is all respired, we find an overall global SOC stock decrease that is 62 % larger compared to a world without soil erosion during the period 1850 – 2005 (Fig. 6A). This assumption is certainly an extreme and unrealistic assumption, as in reality a fraction of the mobilized SOC will remain stored on land, but we take this assumption as an extreme scenario. Including soil erosion in the SOC cycling scheme under the previously mentioned assumption thus reduces the global land C sink

with the largest impact observed for Asia, where the decrease in the total SOC stock is 156% larger when the effects of soil erosion are taken into account (Table 4, Fig. 8A). Some regions, such as Western Europe show instead a smaller SOC loss when erosion is taken into account. This is because we assumed a steady state in 1850, where carbon losses by erosion are equal to the carbon input by litter. And as soil erosion decreased during 1850 - 2005 in Western Europe, mainly due to a decreasing trend in precipitation since 1965, less intense expansion of agricultural

and grasslands (Fig. 5B) and agricultural abandonment, it partly offsets the decrease of SOC by LUC (Fig. 8).
The significantly smaller increase in SOC stocks on agricultural land when the best estimated soil erosion rates are taken into account (Fig. 7) explains the larger decrease in the global SOC stock during 1850 – 2005 (S1) compared to a world without soil erosion (Fig. 6A). Due to the slow response of the global SOC stocks to perturbations, this impact of soil erosion can be even larger at longer timescales. The effect of soil erosion on the SOC stocks is also

influenced by the mechanism where removal of SOC causes a sink in soils that tend to return to equilibrium.
Furthermore, we find that the variability in the temporal trend of global SOC erosion is mainly determined by the variability in soil erosion rates and less by climate and rising atmospheric $CO_2$ that are affecting SOC stocks (Fig. 4). Also the spatial variability in SOC erosion rates for the year 2005 and the spatial variability in the change of SOC erosion during 1850 – 2005 follow closely the spatial variability of soil erosion rates (Fig. 5B, D). This can be

explained by the slow response of the SOC pools to changes in NPP and decomposition caused by $CO_2$ and climate in contrast to the fast response of soil erosion to changes in land cover and climate.

**4.2 LUC versus precipitation and temperature change**

Although the variability in the temporal trends of soil and SOC erosion is dominated by the variability in precipitation changes, the overall trend follows the increase in agricultural land and grassland. The global decrease in precipitation in many regions worldwide, especially in the Amazon, as simulated by ISIMIP2b, leads to a slight decrease in soil and SOC erosion rates (Fig. 4). At the same time precipitation is very variable and might not lead to a significant global net change in soil erosion rates over the total period 1850-2005. This result might be

contradictory to the fact that major soil erosion events are caused by storms. But in this study we only simulate rill and interril erosion, which are usually slow processes. In addition, previous studies (Lal, 2003; Montgomery, 2007; Van Oost *et al.*, 2007) have shown that land use change is usually the main driver behind accelerated rates of these types of soil erosion. Our study confirms this observation.
If we separate the effects of LUC and climate variability co-varying with soil erosion we find that in the "LUC"

erosion scenario with constant climate (see section 2.5), the total global soil loss from erosion increases by a factor of 1.27 since 1850, while in the "CC" erosion scenario with constant LUC at the level of 1850 the soil loss flux from

erosion decreases by a factor of 1.12 (Fig 4). Analyzing the effects of LUC and climate variability separately on SOC erosion we find that in the LUC-only scenario (S2-S1) the total global SOC loss increases by a factor of 1.35 since 1850, while in the climate-change-only scenario (S2) SOC loss decreases by a factor of 1.12 (Fig 4). This

shows that LUC slightly dominates the trend in both soil and SOC erosion fluxes on the global scale during 1850 - 2005.

For soil erosion, however, LUC dominates the temporal trend less than for SOC erosion. This effect is especially clear for grasslands, where we find that climate variability offsets a large part of the increase in soil erosion rates by LUC, but not in the case of SOC erosion. This is the due to the fact that LUC has a much stronger effect on the

carbon content in the soil than the effect of climate and $CO_2$ change on the timescale of the last 200 years. Also, intense soil erosion is typically found in mountainous areas where climate variability has significant impacts, while at the same time these regions are usually poor in SOC due to unfavorable environmental conditions for plant productivity.

Regionally, there are significant differences in the relative contributions of LUC versus climate variability to the

total soil erosion flux (Fig. 2 & 5). In the tropics in South-America, Africa and Asia, where intense LUC (deforestation and expansion of agricultural areas) took place during 1850 - 2005, a clear increase in soil erosion rates is found even in areas with a significant decrease in precipitation due to a higher agricultural area being exposed to erosion. However, in regions where agriculture is already established and has a long history, precipitation changes seem to have more impact than LUC on soil erosion rates. A combination of our assumption that erosion

rates are in steady state with carbon input to the soil at 1850, and minimal agricultural expansion during the last 200 years may be the reasons for this observation.

We also find that summing up the changes in soil erosion rates due to LUC alone and the changes in soil erosion due to climate variability alone do not exactly match the results in the changes in soil erosion obtained when LUC and climate variability are combined (Fig. 4). The non-linear differences between soil erosion rates calculated with

changing land cover fractions in combination with a constant climate ("LUC"), and soil erosion rates calculated by subtracting the erosion simulation "CC" from "CC+LUC" are significant for agricultural land but much smaller for other PFTs and at the global scale. It implies that the LUC effect on erosion depends on the background climate. This is important to keep in mind when evaluating the LUC effect on SOC stocks in the presence of soil erosion.

The decrease in global SOC stocks in simulation S3 are due to the various effects of LUC (without erosion) (Fig.

6A). During 1850 – 2005 LUC has led to a decrease in natural vegetation and an increase in agricultural land. At the global scale, the replacement of natural PFTs by crop results in increased SOC decomposition and decreased carbon input to the soil by litter-fall due to harvest and a lower productivity. Regionally this effect of LUC may be different, depending for example on the natural PFTs that are replaced. Furthermore, the increase in carbon input into the soil after LUC due to increased litter fall when natural vegetation is removed may play a role, but this effect is only

temporary. In addition, wood harvest after deforestation and crop harvest contribute to the decreased carbon input to the soils.

We find that the global SOC stock decreases by 17 Pg C due to LUC only during 1850 – 2005 (Fig. 6A, S3-S4). The overall change in carbon over this period summed up over all biomass, litter, SOC, and wood-product pools due to

LUC without erosion is a loss of 102 Pg C, which lies in the range of cumulative carbon emissions by LUC from estimates of previous studies (Houghton and Nassikas, 2017; Li et al., 2017; Piao et al., 2009). When we use our best estimated soil erosion rates in the SOC-dynamics scheme of the emulator we find that the LUC effect on the global SOC stock is amplified by 4 Pg C or a factor of 1.2 (S1-S2, Fig. 6A). The main reason behind this is the increase in soil erosion rates by expanding agricultural- and grasslands that limits the increase in the global agricultural and grassland SOC stock due to LUC (Fig. 7). This leads to a total change in the overall carbon stock on land due to LUC of -106 Pg. Regionally the amplification of the LUC effect on SOC stocks by the increase in soil erosion ranges between factor of 0.9 and 1.6 (Table 4).

Regionally, changes in precipitation can amplify or offset a large part of the increase in soil erosion due to LUC (Fig. 2 & 5E). Globally we find that the decrease in global total precipitation, especially in the Amazon after 1960AD, partly offsets the increase in soil loss due to land use change (Fig. 4). It should be noted that the uncertainty in precipitation from global climate models for the Amazon is significant making this result uncertain (Mehran *et al.*, 2014). Furthermore, we find that precipitation and temperature changes lead to a small net decrease in SOC stocks at the global scale since 1950 (Fig. 9, S8). This is likely related to the decreased productivity under drought stress (Piao *et al*., 2009). However, soil erosion offsets this decrease by a small net increase of 2 Pg in SOC stocks, mostly due to the decreasing trend in precipitation globally after 1950 AD.

### 4.3 Effects of atmospheric $CO_2$ increase

In the ORCHIDEE model, increasing $CO_2$ leads to a fertilization effect as it increases the NPP, and results in a significant increase in biomass production on land for most PFTs, depending on the temperature and moisture conditions (Arneth *et al.*, 2017; Piao *et al.*, 2009). Figure 9 shows the contribution of this fertilization effect to the cumulative SOC stock change during 1850 – 2005 (S4-S8), which is in the same order of magnitude as the effect of LUC excluding soil erosion. Together with climate variability the atmospheric $CO_2$ increase offsets all the carbon losses by LUC in our model, and leads even to a net cumulative sink of carbon on land over this period of about 30 Pg C (S3). This value is calculated by summing up the changes in all the biomass, litter and SOC pools, and is in line with other assessments that found a net carbon balance that is close to neutral over 1850 - 2005 (Arora *et al*., 2011; Ciais *et al*., 2013; Khatiwala *et al.*, 2009).

In the presence of soil erosion, climate variability and the atmospheric $CO_2$ increase lead to a net cumulative sink of carbon over land of about 28 Pg C (S1), still within the uncertainty of assessed estimates (Arora *et al.*, 2011; Ciais *et al.*, 2013; Khatiwala *et al.*, 2009). Soil erosion can thus slightly change the sink strength by influencing the net effect of LUC on the terrestrial carbon balance.

When the $CO_2$ fertilization effect is absent (S5, S7), we find that the temporal trend in the cumulative change of global SOC stocks is largely determined by the effect of LUC (Fig. 6B), and leads to a cumulative source of carbon on land of 76 Pg C. LUC alone leads to a cumulative decrease in SOC stocks of -14 Pg C (S7-S8), which is 3 Pg C less than the decrease in SOC stocks due to LUC in the presence of increasing atmospheric $CO_2$ concentrations (S3-S4). The overall change in carbon over 1850-2005 summed up over all biomass, litter and SOC pools due to LUC

alone is -74 Pg C in absence of increasing $CO_2$ (S7-S8), which is 27 Pg C less than the LUC effect on carbon stocks under variable atmospheric $CO_2$ (S3-S4). LUC has indeed a smaller effect on carbon stocks in the absence of increasing $CO_2$ concentrations as expected, because the productivity of the vegetation is lower (lower NPP) resulting in less biomass that can be removed by deforestation.

The previously calculated global total soil erosion flux of 47.6 Pg $y^{-1}$ leads to an annual SOC erosion flux of 0.48±0.13 Pg C $y^{-1}$ in the year 2005 in the absence of increasing atmospheric $CO_2$ (S5), which is about 0.04 Pg C $y^{-1}$ less than the SOC erosion flux under increasing $CO_2$ (S1). The global cumulative SOC erosion over the entire time period in the absence of increasing atmospheric $CO_2$ is about 4.78 Pg C less (S5). Although these changes in SOC are small, the effect of LUC on the SOC stocks is amplified by erosion with a factor of 1.26 in absence of increasing

$CO_2$ (S5-S6), which is slightly larger than the effect of LUC with increasing $CO_2$ (S1-S2). This means that the LUC effect in combination with soil erosion has a stronger effect on SOC stocks losses under constant atmospheric $CO_2$ conditions, because the $CO_2$ fertilization effect does not replenish SOC in agricultural lands everywhere.

Finally, it should be mentioned here that the absence of nutrients in the current version of the ORCHIDEE model may result in an overestimation of the $CO_2$ fertilization effect on NPP and may introduce biases in the effect of

erosion on SOC stocks under increasing atmospheric $CO_2$ concentrations. Soil erosion may also lead to significant losses of nutrients in the real world, especially in agricultural areas. For a more complete quantification of the effects of soil erosion on the carbon cycle, nutrients have to be included in future studies.

**4.4 Model limitations, uncertainties and next steps**

One of the uncertainties in our modeling approach is related to the application of the Adj. RUSLE model at the global scale and the estimation of the model parameters for various different environmental conditions and biomes. Although the Adj. RUSLE model was extensively validated using large high-resolution datasets, we calculated an

uncertainty range for the R and C factors of the model to investigate the sensitivity of the emulator to the uncertainty in soil erosion rates. In section 4.1 we show that including soil erosion in the emulator decreases the land carbon sink due to the large SOC losses on agricultural land triggered by erosion that reduce the SOC stocks significantly. Without soil erosion (S3) the global agricultural SOC stock increases by 60 Pg C due to agricultural expansion, while soil erosion reduces this increase by 11 Pg C in the minimum soil erosion scenario (S1 minimum) and by 18

Pg C in the maximum soil erosion scenario (S1 maximum). LUC results thus in a smaller increase in the global agricultural SOC stock under all soil erosion scenarios, while the magnitude of this effect is region-dependent. The larger the soil erosion rates, the lesser carbon can be stored on agricultural land.

Furthermore, the aggregation of the high-resolution soil erosion rates from the Adj.RUSLE model to the resolution of the emulator can induce some uncertainties, as we might not capture correctly the hotspots of carbon erosion and

their effects on the local SOC dynamics in these regions. However, the aggregation was needed to be consistent with the coarse resolution of ORCHIDEE and the limit the computational power of the emulator.

In addition, our soil erosion model is limited to water erosion only. This might result in biases for regions where other types of soil erosion are dominant such as, tillage erosion (Van Oost *et al*., 2007), gully erosion and landslides (Hilton *et al*., 2008; Hilton *et al*., 2011; Valentin *et al.*, 2005).

Although our erosion model runs on a daily time step, the soil erosion rates are calculated on a yearly time step, and thus we might miss extreme climate events triggering large soil losses. In addition, the Adj.RUSLE is not trained for extreme events. The effect of precipitation and temperature change on the SOC stocks under soil erosion might thus be larger than in our model simulations.

Concerning the reconstructed PFT maps, only expansion and abandonment of agriculture is taken into account, but
not soil conservation measures as implemented in Australia and the US to prevent erosion (Chappell *et al.*, 2012; Houghton *et al.*, 1999). Regarding the land use change method that we applied, we only account for net land use change and do not account for shifting cultivation or distinguish between areas that have already seen LUC. Forest regrowth and forest age are also not considered, which could bring uncertainties in our estimates of LUC emissions (Yue *et al*., 2017). To show the potential uncertainty in our results due to uncertainties in underlying land use data
we performed 4 additional simulations (S1 to S4) with a new PFT map using the same methods and data as by Peng et al. (2017), however, where the forest area is not constrained with historical data from Houghton (2003, 2008) and present-day data from satellite land-cover products. In the following we will refer to the new PFT map as the unconstrained PFT map. In the unconstrained PFT map there is a stronger decrease in forest area over the period 1850-2005. Also the grassland shows an increasing trend, while in the PFT map with constrained forest the
grassland shows globally a slight decreasing trend (Fig. S2 of the supporting material).

After calculating soil erosion with the unconstrained PFT map we find that the differences in global average soil erosion rates between the different PFT maps are small (Fig. S3A in the supporting material). This can be related to the fact that the C-factor of the Adj. RUSLE model is similar for forest and dense natural grass. As the change in global agricultural area is not significantly different between the two PFT maps, the overall soil erosion rates are
similar. We expect, however, that the differences in soil erosion rates between the PFT maps can be larger in areas where the change in forest area is substantial over the historical period.

In contrast to the soil erosion rates, the two PFT maps result in significant differences in the SOC erosion rates and cumulative changes in SOC stocks during the transient period (Fig. S4 of the supporting material). The global SOC stock in the equilibrium state without soil erosion (S3) is 8 % higher when the unconstrained PFT map is used, due
to a larger global forest area in this map at 1850. The higher global SOC stock of the unconstrained PFT map leads to higher SOC erosion rates (Fig. S3 B in the supporting material). According to the unconstrained PFT map, soil erosion leads to a total SOC removal of 79 Pg C (S1) over the period 1850-2005, which is 5 Pg C larger than the total SOC removal by soil erosion for the constrained PFT map.

Interestingly, due to the unconstrained PFT map, the global cumulative SOC stock change over 1850-2005 under
soil erosion, climate change and LUC (S1) is 60% smaller than the stock derived using the constrained PFT map. This is most likely due to the higher forest area at the start of the period 1850-2005, leading to a larger increase in SOC stocks by increasing atmospheric $CO_2$ concentrations. We find the global LUC effect on the SOC stocks of both PFT maps to be similar (Fig. S3 C, D in the supporting material).

**5 Conclusions**


In this study we introduced a 4D modeling approach where we coupled soil erosion to the C-cycle of ORCHIDEE and analyzed the potential effects of soil erosion, without sediment deposition or transport, on the global SOC stocks over the period 1850 – 2005. To calculate global potential soil erosion rates we used the Adj.RUSLE model that includes scaling approaches to calculate soil erosion rates at a coarse spatial and temporal resolution. The SOC

dynamics are represented by an emulator that imitates the behavior of the carbon cycle of the ORCHIDEE LSM and enables us to easily couple our soil erosion model to the C-cycle and calculate the effects of soil erosion under different climatic and land use conditions. Although our modeling approach is rather coarse and fairly simple, we found a fair agreement of our soil loss and SOC loss fluxes for the year 2005 with high-resolution estimates from other studies.

When applying the model on the time period 1850-2005 we found a total soil loss flux of 7183±1662, where soil erosion rates increased strongest on agricultural land. This potential soil loss flux mobilized 74±18 Pg of SOC across all PFTs, which compares to 46-74% of the total net flux of carbon lost as $CO_2$ to the atmosphere due to LUC estimated by our study for the same time period. When assuming that all this SOC mobilized is respired we find that the overall SOC change over the period 1850-2005 would increase by 62% and reduce the land carbon sink by 2 Pg

of carbon. The effect of soil erosion on the cumulative SOC change between 1850 and 2005 AD differs significantly between regions, where the largest decrease in SOC due to soil erosion is found in Asia. The expansion of agricultural and grassland is the main driver behind the decreasing SOC stocks by soil erosion. Including soil erosion in the SOC dynamics amplifies the decrease in SOC stocks due to LUC by a factor of 1.2. Overall, the potential effects of soil erosion on the global SOC stocks show that soil erosion needs to be included in future

assessments of the terrestrial C-cycle and LUC.

**Data availability:** Upon request from the authors

**Author contribution**


VN, PC, BG, designed the research, VN, PC, RL and YW performed the research, VN and YW performed the model simulations and data analysis, and all authors wrote the paper. The authors have no conflict of interest to declare.

**Acknowledgements**

Funding was provided by the Laboratory for Sciences of Climate and Environment (LSCE), CEA, CNRS and UVSQ. RL acknowledges funding from the European Union's Horizon 2020 research and innovation program under grant agreement no.703813 for the Marie Sklodowska-Curie European Individual Fellowship "C-Leak". PC

acknowledges support from the European Research Council Synergy grant ERC-2013-SyG-610028 IMBALANCE-

P. BG acknowledges support from the project ERANETMED2-72-209 ASSESS. We also thank A.Chappell and D.S.Goll for their useful comments during the early stages of this research, and D.Zhu for helping out with the model simulations.

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

Table 1: Description of the simulations used in this study. The S1 simulation is also the control simulation (CTR).

S1 and S2 minimum and maximum are the sensitivity simulations using minimum or maximum soil erosion rates.

"Best" stands for the best estimated soil erosion rates using optimal values for the R and C factors of the

Adj.RUSLE model.

| Simulation | Climate change | $CO_2$ change | Land use change | Erosion |
|------------|----------------|---------------|-----------------|---------|
| S1 | Yes | Yes | Yes | Best |
| S1 minimum | Yes | Yes | Yes | minimum |
| S1 maximum | Yes | Yes | Yes | maximum |
| S2 | Yes | Yes | No | Best |
| S2 minimum | Yes | Yes | No | minimum |
| S2 maximum | Yes | Yes | No | maximum |
| S1-S2 | No | Yes | Yes | Best |
| S3 | Yes | Yes | Yes | No |
| S4 | Yes | Yes | No | No |
| S3-S4 | No | Yes | Yes | No |
| S5 | Yes | No | Yes | Best |
| S6 | Yes | No | No | Best |
| S5-S6 | No | No | Yes | Best |
| S7 | Yes | No | Yes | No |
| S8 | Yes | No | No | No |
| S7-S8 | No | No | Yes | No |

Table 2: Area weighted average and standard deviation of soil and SOC erosion rates per PFT for the year 2005AD; the uncertainty range for soil erosion rates is 25 - 53 % and for SOC erosion rates 16-50%.

| PFT | Mean soil erosion (t ha$^{-1}$ y$^{-1}$) | Standard deviation soil erosion (t ha$^{-1}$ y$^{-1}$) | Mean SOC erosion (kg C ha$^{-1}$ y$^{-1}$) | Standard deviation SOC erosion (kg C ha$^{-1}$ y$^{-1}$) |
|---|---|---|---|---|
| Crop | 2.45 | 35 | 10.38 | 466 |
| Grass | 1.80 | 30 | 5.79 | 91 |
| Forest | 0.34 | 3 | 1.34 | 14 |

Table 3: Model estimates per continent of area-weighted average annual soil erosion and SOC erosion rates for the year 2005, their spatial standard deviations, and the changes in average soil and SOC erosion rates since 1851; the uncertainty range for soil and SOC erosion rates is 2 - 36 % and 3-52%, respectively. The uncertainty range for the changes in soil and SOC erosion rates since 1851 is 3 – 83 % and 11-166%, respectively.

| Region | Mean soil erosion rate | Standard deviation soil erosion rate | Change in mean soil erosion rate | Mean SOC erosion rate | Standard deviation SOC erosion rate | Change in mean SOC erosion rate |
|---|---|---|---|---|---|---|
| | 2005 | 2005 | 2005 -1851 | 2005 | 2005 | 2005-1851 |
| | (t ha$^{-1}$ y$^{-1}$) | (t ha$^{-1}$ y$^{-1}$) | (t ha$^{-1}$ y$^{-1}$) | (kg C ha$^{-1}$ y$^{-1}$) | (kg C ha$^{-1}$ y$^{-1}$) | (kg C ha$^{-1}$ y$^{-1}$) |
| Africa | 2.35 | 59.96 | 0.58 | 13.19 | 95.22 | 4.31 |
| Asia | 5.66 | 157.90 | 0.21 | 58.03 | 802.53 | 3.23 |
| Europe | 2.07 | 62.50 | 0.39 | 16.68 | 338.03 | 1.39 |
| Australia | 1.40 | 16.29 | -0.47 | 5.16 | 22.91 | 1.82 |
| South-America | 4.52 | 113.26 | 1.29 | 74.42 | 1515.04 | 38.97 |
| North-America | 2.60 | 58.62 | 0.13 | 32.88 | 556.44 | 3.14 |
| Global | 3.61 | 96.43 | 0.45 | 38.56 | 666.39 | 8.94 |

Table 4: Model estimates per continent of changes in SOC stocks since 1851 from simulations S1, S2, S1-S2, S3, S4, S3-S4.

| Region | Change SOC stocks S1 | Change SOC stocks S2 | Change SOC stocks S1-S2 | Change SOC stocks S3 | Change SOC stocks S4 | Change SOC stocks S3-S4 |
|---|---|---|---|---|---|---|
| | Pg C | Pg C | Pg C | Pg C | Pg C | Pg C |
| Africa | -1.55 | -0.24 | -1.31 | -1.54 | -0.55 | -0.98 |
| Asia | -0.36 | 7.94 | -8.31 | 0.65 | 7.11 | -6.47 |

| | | | | | |
|---|---|---|---|---|---|
| Europe | -3.33 | 1.78 | -5.12 | -4.35 | 1.52 | -5.87 |
| Australia | 0.21 | 0.05 | 0.16 | 0.29 | 0.01 | 0.28 |
| South-America | 2.24 | 2.82 | -0.59 | 3.75 | 2.06 | 1.69 |
| North-America | -2.62 | 3.19 | -5.81 | -2.5 | 3.03 | -5.53 |
| Global | -5.35 | 15.93 | -21.29 | -3.3 | 13.86 | -17.16 |

Table 5: Statistics of a grid cell by grid cell comparison of global SOC stocks between GSDE soil database and simulations S1 (with erosion) and S3 (without erosion). RMSE is the root mean square error and r-value is the correlation coefficient of the linear regression between GSDE and S1 of S3.

| Soil depth (m) | GSDE SOC total (Pg) | S1 SOC total (Pg) | S3 SOC total (Pg) | RMSE S1 | RMSE S3 | r –value S1 | r – value S3 |
|---|---|---|---|---|---|---|---|
| 0.3 | 670 | 428 | 556 | 5218 | 5861 | 0.43 | 0.44 |
| 1 | 1356 | 672 | 846 | 14077 | 10213 | 0.52 | 0.51 |
| 2 | 1748 | 1001 | 1284 | 12968 | 13195 | 0.56 | 0.55 |

Table 6: Comparison of our model estimates of agricultural soil and SOC loss fluxes for the year 2005 with high-resolution model/observation estimates. *Quinton et al. (2010) included also pasture land in their study.

| Study | Soil loss Pg y[-1] | SOC loss Pg C y[-1] |
|---|---|---|
| Van Oost et al. (2007) | 17 | 0.25 |
| Doetterl et al. (2012) | 13 | 0.24 |
| *Quinton et al. (2010) | 28 | 0.5±0.15 |
| Chappell et al. (2015) | 17- 65 | 0.37 – 1.27 |
| Wang et al. (2017) | 17.7±1.70 | 0.44±0.06 |
| This study | 12.28±4.62 | 0.16±0.06 |

Table 7: Comparison of our model best estimates of agricultural soil erosion and SOC erosion rates for the year 2005 with best model/observation estimates from Doetterl et al. (2012) per continent. The uncertainty range for the present-day sediment and SOC fluxes is 18 – 62 % and 5-51%, respectively. The uncertainty range in the values of Doetterl et al. (2012) is 30 - 70%.

| | *Our Study* | | *Doetterl et al. (2012)* | |
|---|---|---|---|---|
| Region | Sediment flux 2005 Pg y[-1] | SOC flux 2005 Tg C y[-1] | Sediment flux 2000 Pg y[-1] | SOC flux 2000 Tg C y[-1] |

| | | | | |
|---|---|---|---|---|
| Africa | 2.6 | 20.3 | 2.4 | 39.5 |
| Asia | 5.4 | 54.3 | 4.9 | 90.0 |
| Europe | 2.1 | 30.8 | 1.9 | 39.5 |
| Australia | 0.2 | 2.5 | 0.3 | 4.3 |
| South-America | 1.6 | 39.0 | 1.4 | 26.7 |
| North-America | 0.7 | 12.4 | 1.6 | 31.5 |
| Total | 12.3 | 161.5 | 12.5 | 231.5 |

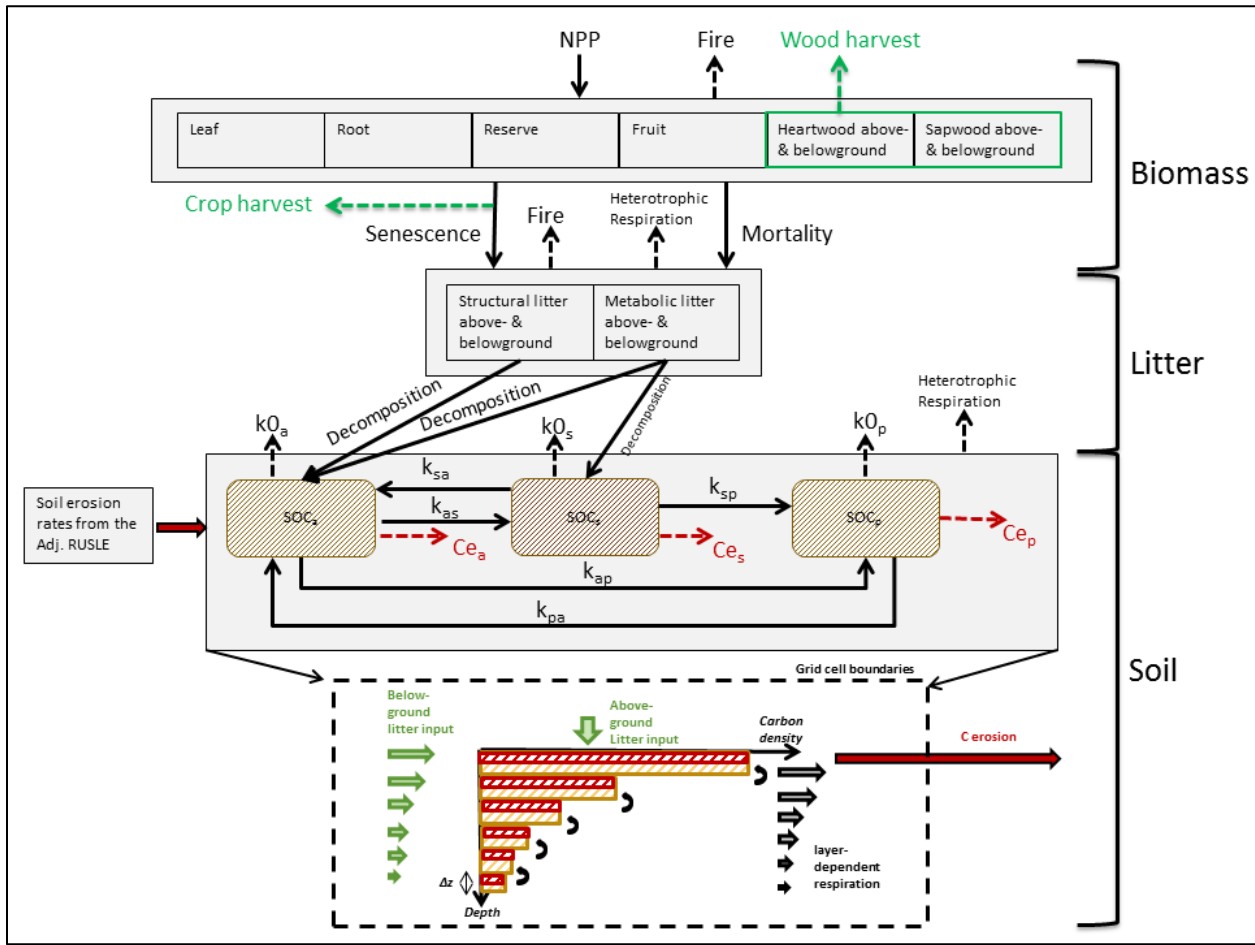

Figure 1(A): The structure of the C emulator (see variable names in the text, paragraph 2.3). The C erosion fluxes
are represented by the red arrows and calculated using the soil erosion rates from the Adj.RUSLE model

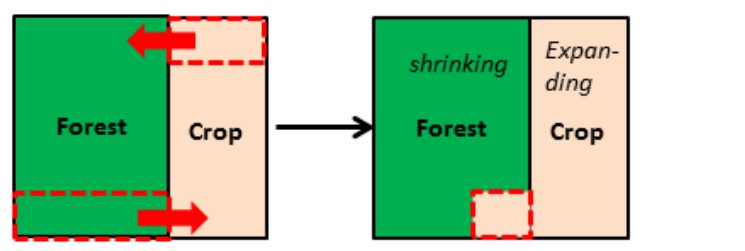

net land use change

1. Read PFT fractions

2. Identify expanding & shrinking PFT's

3. Collect wood products (wood harvest)

4. Sum the biomass of the area lost of all shrinking PFT's

5. Allocate this to the metabolic/structural litter of expanding PFT's

6. Sum the litter of the area lost of all shrinking PFT's

7. Allocate this to the expanding PFT's

8. Distribute biomass and litter over new PFT area

9. Sum the SOC per soil layer of the area lost of all shrinking PFT's

10. Allocate this to the respective soil layer of the expanding PFT's; In this way the SOC soil profile of the expanding and shrinking PFTs gets mixed

11. Distribute SOC over new PFT area

Figure 1(B): The land use change module of the emulator

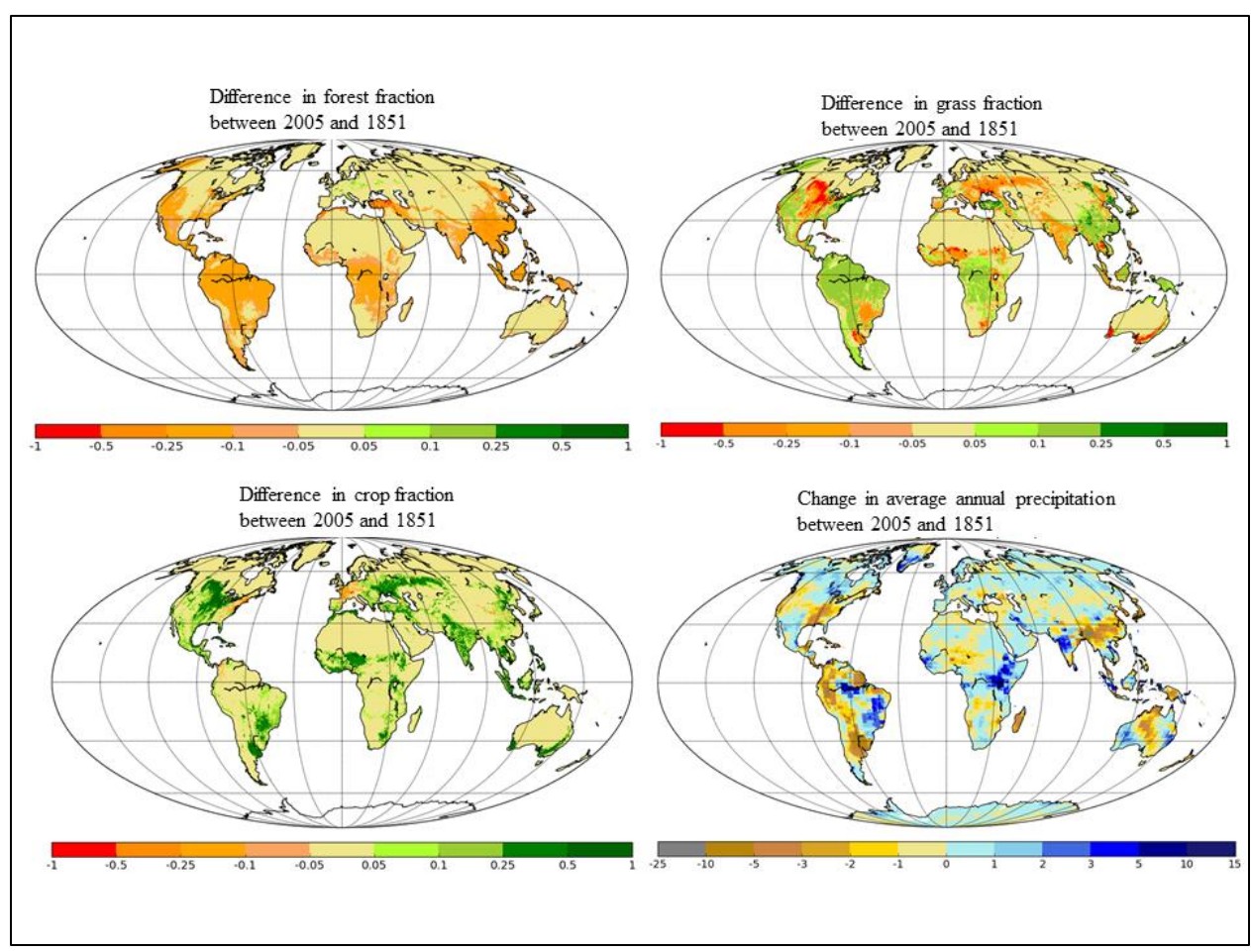

Figure 2: Spatial patterns of the difference in forest, crop and grassland area between 1851 and 2005 represented as a fraction of a grid cell. And spatial patterns of the change in average annual precipitation between 1851 and 2005 in mm y$^{-1}$, calculated as the total change in precipitation over the period 1851 – 2005 and divided by the number of years in this period.

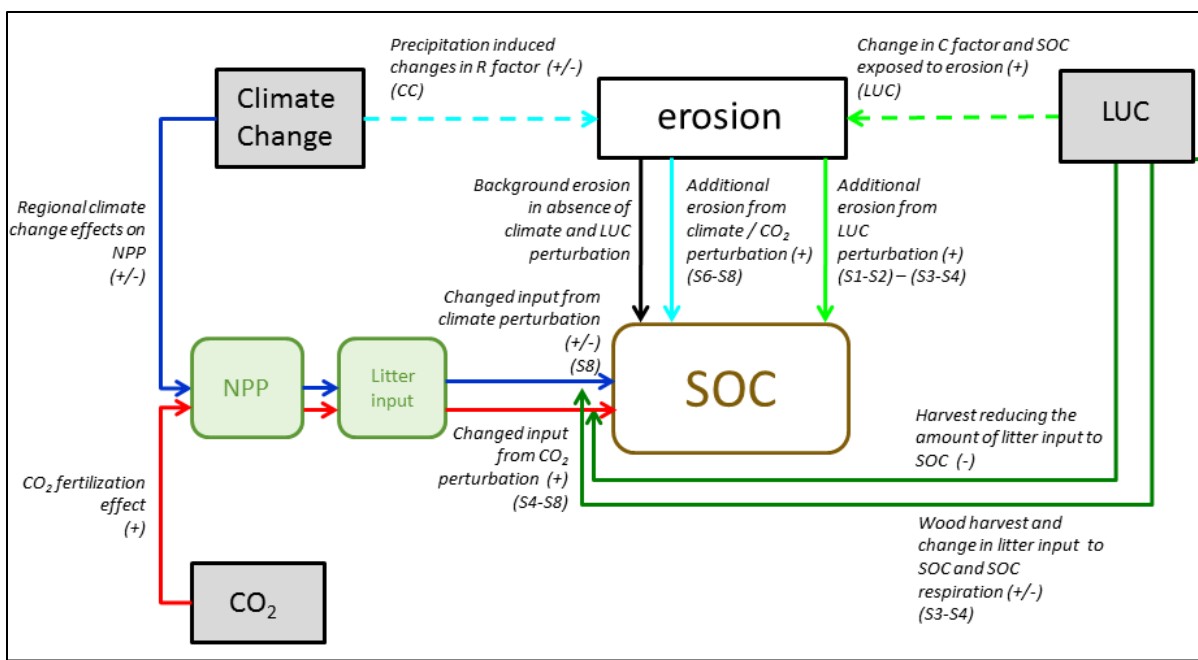

Figure 3: Conceptual diagram of SOC affected by erosion in presence of other perturbations of the carbon cycle, namely climate variability, increasing atmospheric $CO_2$ concentrations and land use change. A separation of these components and of the role of erosion is obtained with the factorial simulations (S1-S8), presented in Table 1

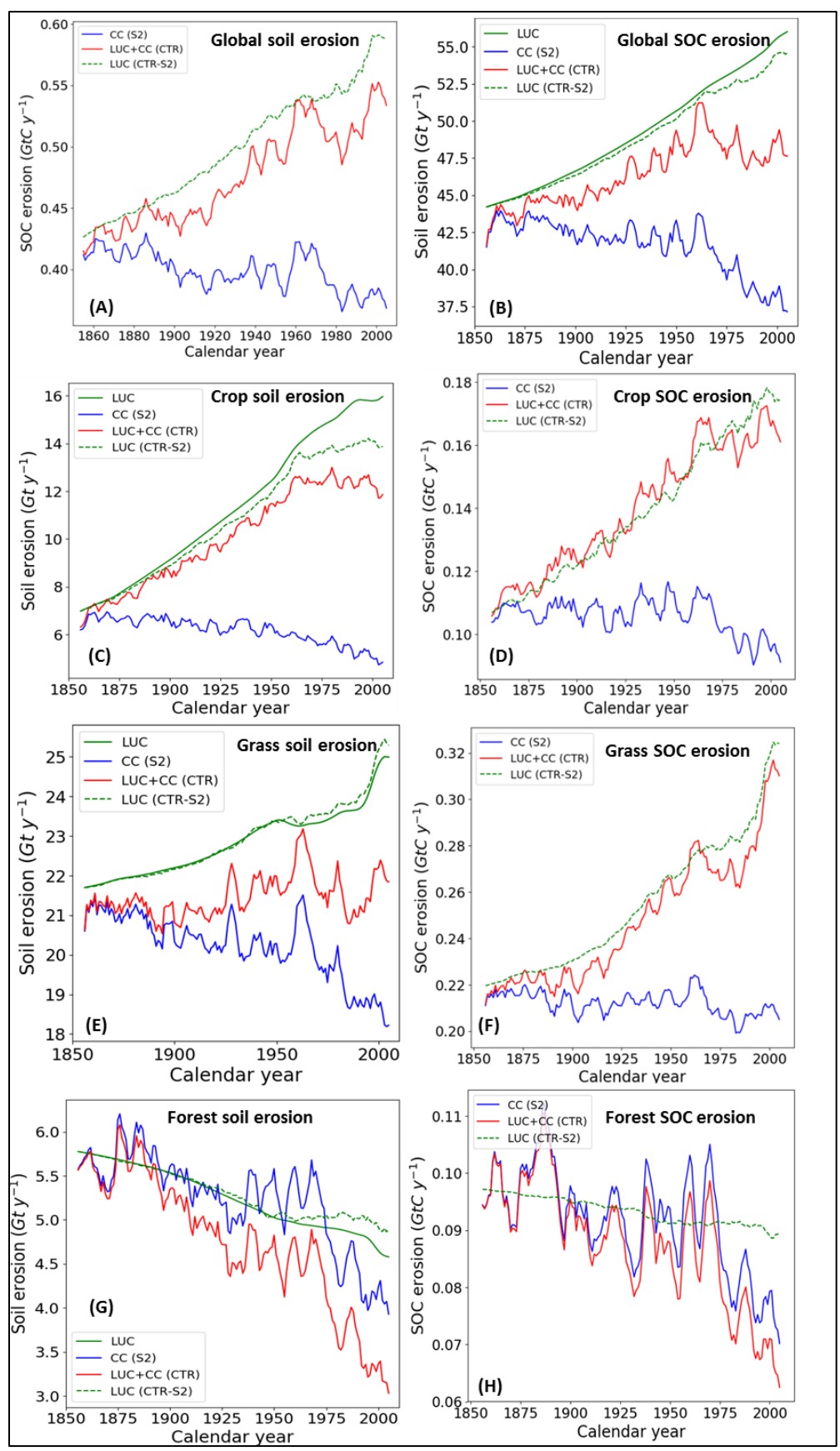

Figure 4: (A) Global annual soil erosion rates, (B) global annual SOC erosion rates, (C) agricultural annual soil erosion rates, (D) agricultural annual SOC erosion rates, (E) grassland annual soil erosion rates, (F) grassland annual SOC erosion rates, (G) forest annual soil erosion rates and (H) forest annual SOC erosion rates over the period 1850-2005 for scenario's with only LUC (green lines), scenario with only climate and $CO_2$ change (blue line) and scenario with LUC, climate and $CO_2$ change (red line). In figures A, C, E, and G the dashed green line is the difference between the CTR and S2 simulations, while the straight green line is the LUC-only simulation with the Adj.RUSLE model.

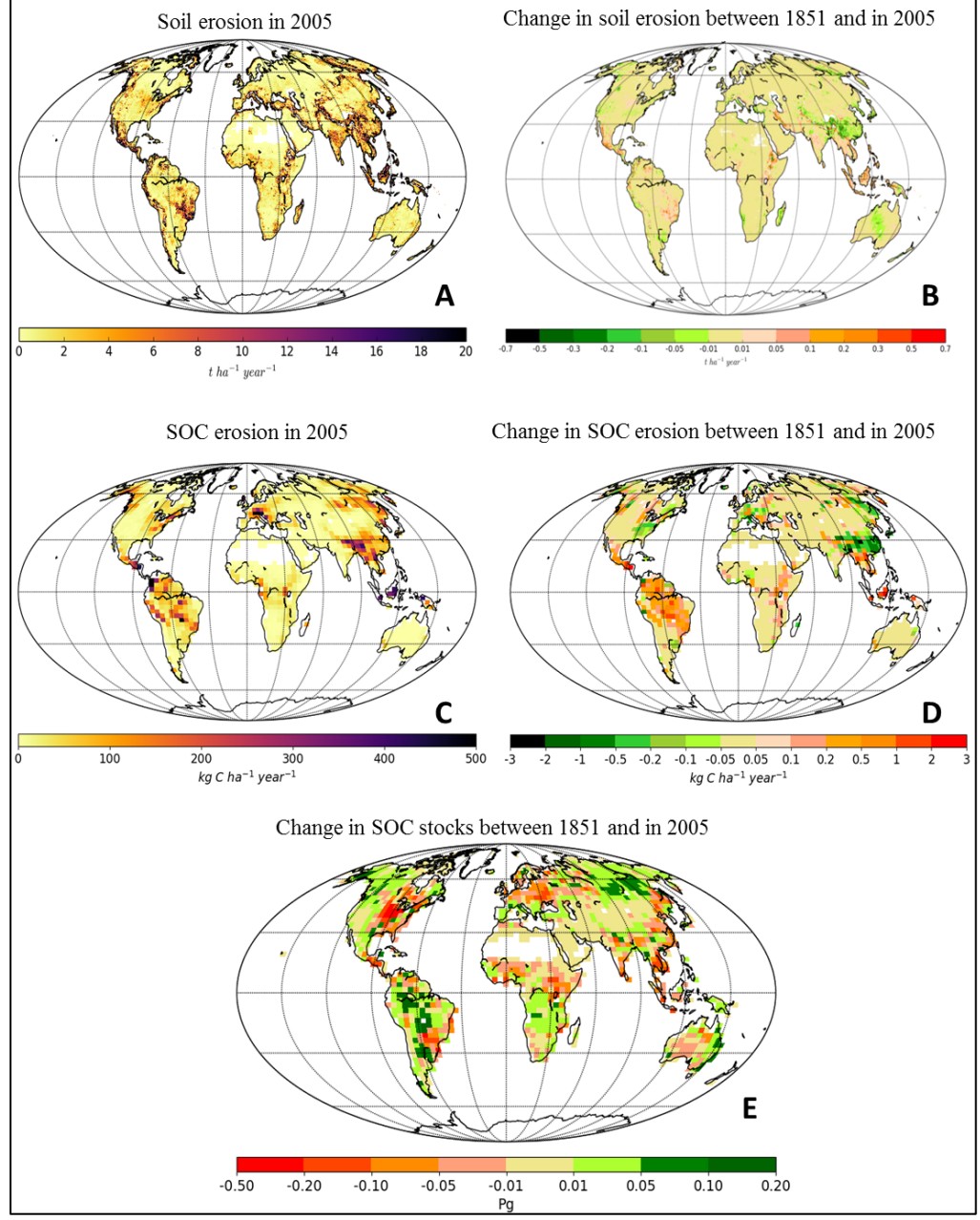

Fig 5: (A) Average annual soil erosion rates at a 5 arcmin resolution in the year 2005, (B) change in average annual soil erosion rates over the period 2005-1850, (C) average annual SOC erosion rates at a resolution of 2.5x3.75 degrees in 2005, (D) change in average annual SOC erosion rates over the period 2005-1850, and ( E ) difference in SOC stocks at a resolution of 2.5x3.75 degrees between the year 2005 and 1850 (CTR simulation). For the SOC stocks positive values (green color) indicate a gain, while negative values (red color) indicating a loss. For the erosion rates positive values (red color) indicate an increase over 1850 - 2005, while negative values (green color) indicate a decrease over 1850 – 2005

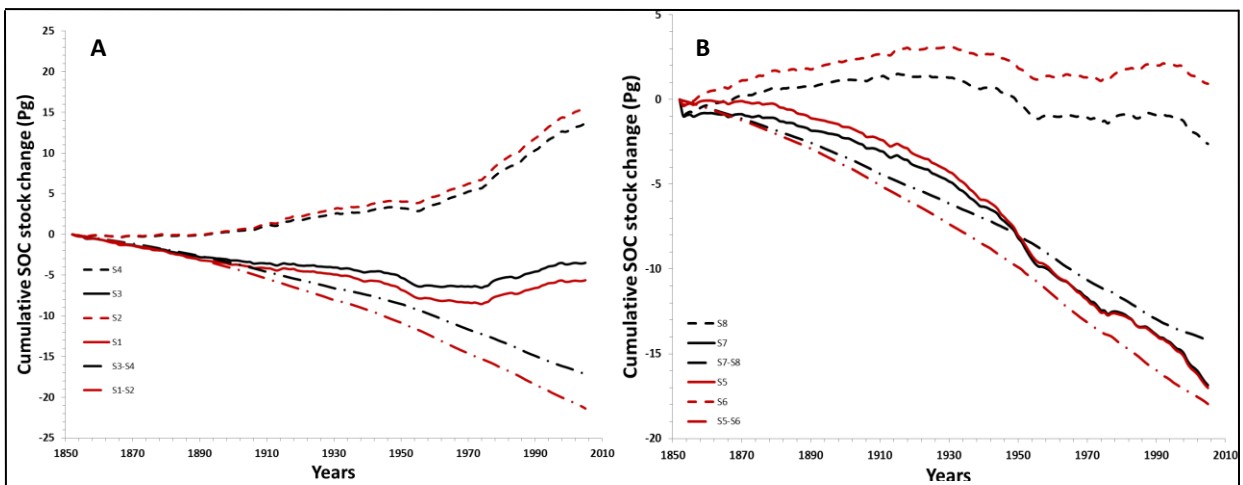

Figure 6: Cumulative SOC stock changes during 1850 – 2005 for (A) simulations with variable atmospheric $CO_2$ concentration,  and (B) for simulations with a constant $CO_2$ concentration, implied by variable land cover alone (dash-dotted lines), by variable climate (dashed lines), and variable land cover and climate (straight lines), without erosion (black lines) and with erosion (red lines).

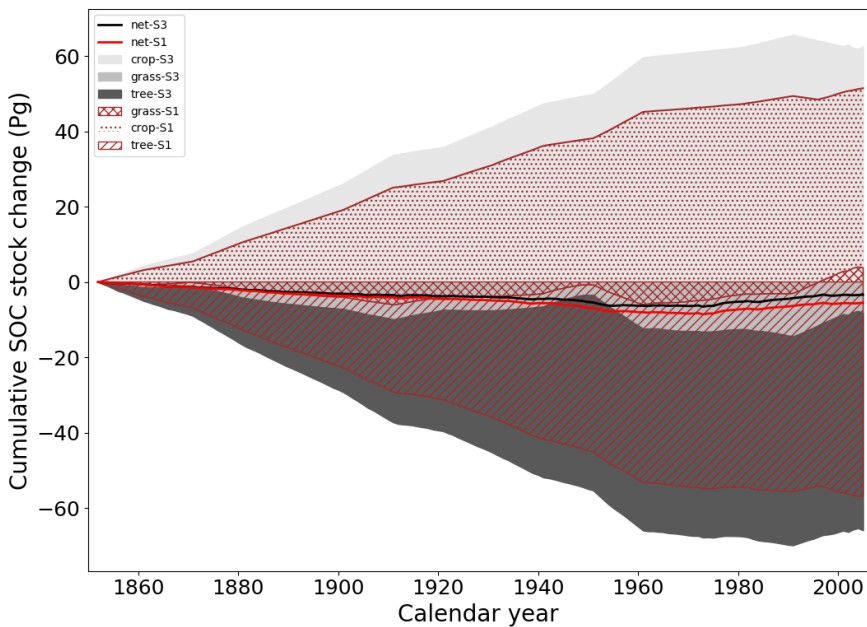

Figure 7: Cumulative SOC stock changes per PFT during 1850 – 2005 implied by variable land cover, climate and $CO_2$, without erosion (grey colors) and with erosion (red colors).

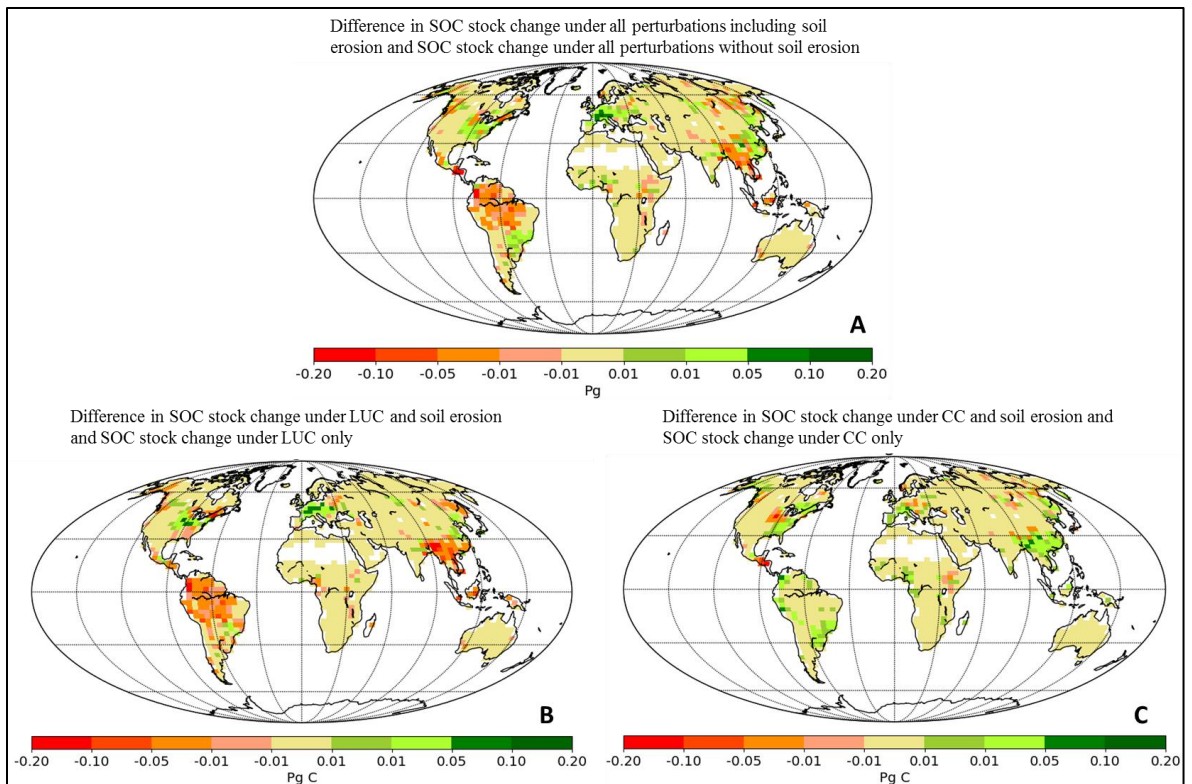

Figure 8: A) Difference between the changes of SOC stocks over the period 1850-2005 under all perturbations including soil erosion and the changes in SOC stocks excluding soil erosion, S1-S3, B) Difference between the changes of SOC stocks under LUC including soil erosion and the changes in SOC stocks excluding soil erosion (S1-S2)-(S3-S4), C) Difference between the changes of SOC stocks under a variable climate and $CO_2$ increase including soil erosion and the changes in SOC stocks excluding soil erosion, S2-S4.

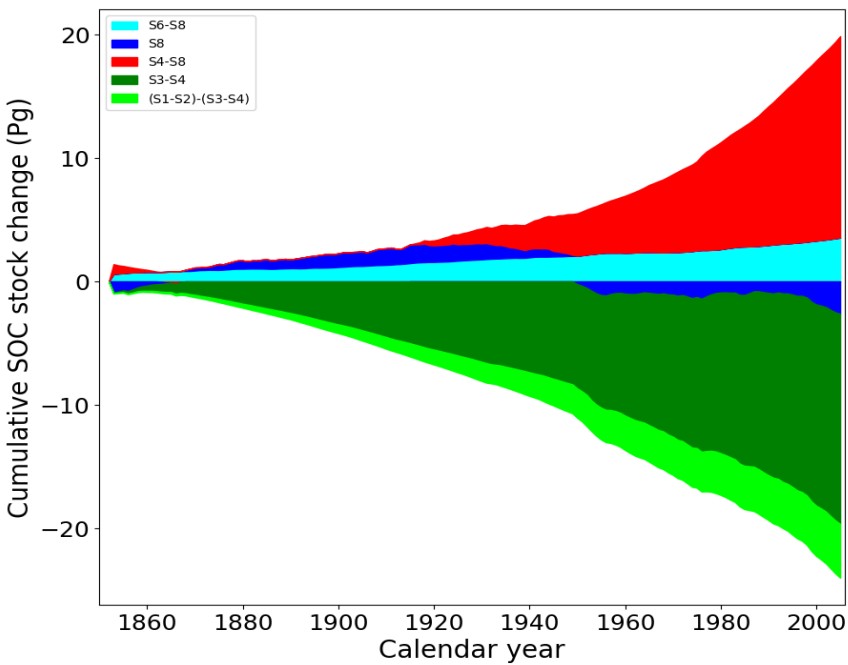

Figure 9: Contribution to the cumulative global SOC stock change over 1850-2005 by $CO_2$ fertilization (red), effect of precipitation and temperature change on the carbon cycle (dark blue), effect of precipitation change on soil erosion ( aqua), LUC effect on the carbon cycle (dark green), and LUC effect on soil erosion (light green)