# Peer review of "Global soil organic carbon removal by water erosion under climate change and land use change during 1850-2005 AD"

_Biogeosciences, 2017_

## Short Comment (SC1) · 30 Jan 2018

Dear Authors, I read with interest your study and I believe that it will contribute meaningfully to the body of knowledge about global soil erosion and SOC dynamics. In the chapter Chapter 3.2 Validation of model results, I found two aspects that you may want to take in consideration.

I would suggest to the Authors to reconsider the use of the term validation. A validation, sensu strictu, is generally performed comparing predicted vs. observed values. I understand that for a global modelling application this is a difficult task to perform and I am ok with model comparisons. However, I would suggest to change the term

validation with model comparison /intercomparison/ evaluation.

The predicted global soil loss flux of 14 Pg y-1 is also close to the global model recently published by Borrelli et al. (2017) in Nature Communication. For the year 2012 a total annual average soil erosion of 17(+1/-0.7) Pg yr−1 is predicted. Considering the high-resolution (ca.250 metre cell size) and the modelling advances introduced by Borrelli et al. (2017), the similar estimates between the two studies suggest that Adj.RUSLE is performing well. Moreover, in Table 6 a wider number of global models, rather different and methodologically independent from each other, providing similar estimates suggest a certain consistency to the predicted order of magnitude.

Congratulations, Pasquale

Borrelli et al. (2017). An assessment of the global impact of 21st century land use change on soil erosion. Nature communications, 8, 2013.

---

## Short Comment (SC2) · 2 Feb 2018

Dear Pasquale, Thank you for the comments. We will reconsider the term 'validation' and include the results from your recent work published in Nature communications in our model comparison.

Regards, Victoria

---

## Referee Comment (RC1) · Anonymous Referee #1 · 9 Feb 2018

General Comments

Naipal et al., present an interesting study that quantified soil organic carbon erosion loss at the global scale from 1850 to 2005. The soil erosion processes are largely underrepresented in current generation earth system land models. I believe this study is making a good step forward, towards better modeling the global carbon cycle. Below are my comments including two major concerns.

Major Comments

1. I was not fully convinced by the vertical discretization approach that the emulator used. First of all, different soil layers have totally different biogeophysical and biogeo-

chemical features. Different layers are experiencing different amount of fresh carbon input (e.g., from fine roots exudates, fine root litter), different microbial community (e.g., fungi/bacteria with different carbon use efficiency), and have different soil structure (e.g., microaggregate, macroagregate).

Secondly, even the idea of summarizing the above-mentioned vertical difference into one single factor (re) is believable, the value of re should be carefully inferred for this model, rather than taking from other studies.

Thirdly, and most importantly, the vertical discretization, artificially, increase total global SOC stock by 44%. This type of artifact should be removed. My suggestions is that, since ORCHIDEE has one single soil layer, k0 of ORCHIDEE is supposed to represent the mean turnover rate of the whole soil column. Therefore, the k0 (equation 5) in the emulator (here aims to represent top soil turnover) should not be k0 from ORCHIDEE. One approach is to change k0 in emulator to offset the total SOC stock artifact until it's removed.

2. Land use change map. The LUC is prescribed by PFT fractional change derived from Peng 2017. Wondered how this LUC dataset differs from Land-Use Harmonization (LUH2), the new CMIP6 land use change dataset. Given that LUC is a dominant factor of SOC erosion, I am curious about the uncertainty of SOC erosion, induced by using different LUC estimate (e.g., Peng 2017 vs LUH2).

Specific Comments

L16 The first sentence gives me an incorrect hint that the paper is going to talk about agriculture activity accelerates soil erosion.

L38 1.0 Pg, does it include fire emission?

L56 what is bookkeeping models?

L70 slow rate of carbon sequestration

L85 In order to better constrain

L96 be able to, remove

L179 in this version of ORCHIDEE model

L23 What's the meaning of randomly projected? A more reasonable way is to repeat 1990-1910 climates during 1850-1900.

L351 comapred to that without soil erosion

L421 "Also, intense soil erosion is typically found in mountainous areas where climate variability has significant impacts, while at the same time these regions are usually poor in SOC." It's not clear in the manuscript whether or not ORCHIDEE has topography information? In another word, if ORCHIDEE simulates a low SOC stock over the grid cells that have mountains, is that because of the topographical feature of this gridcell can not hold a lot of SOC in ORCHIDEE? Or because of other reasons such as climate constraints (e.g., colder in mountain area)?

L465 CO2 fertilization effects on NPP is not fully convincing here, because ORCHIDEE does not have nutrient constraints. OCN might be a better surrogate model to be able to say something about CO2 fertilization effect on NPP.

Figure 4. I do not fully understand why climate change either decrease or not change erosion?

---

## Referee Comment (RC2) · Anonymous Referee #2 · 18 Feb 2018

General comments:

Naipal and coauthors present an interesting study in which they estimated the potential magnitude of carbon loss from soil erosion globally from 1850 to 2005. Soil erosion is often absent from land-use change assessments and earth system models. However, as the authors show, soil erosion can play a significant role in SOC dynamics at the global scale and must be represented more thoroughly in large-scale models.

Major / Specific comments:

The emulator used in this study seems to have various limitations that make the numbers presented quite uncertain – further discussion on, and quantification of, these

uncertainties is warranted and would greatly improve this manuscript. Specifically, I would have liked to see additional support for the SOC model formulation, parameters, and built-in feedbacks chosen for the emulator, as well as support for its vertical discretization and parameterization.

The carbon emulator is supposed to describe the carbon pools and fluxes exactly as in ORCHIDEE, yet the total global SOC stocks from the emulator are 44% higher than that of the original ORCHIDEE model. This is a big difference. What does this tell us about the accuracy and applicability of the emulator, and how do the SOC stocks of the two models compare to the Harmonized World Soil Database (HWSD) and other global SOC databases? Additional major comments/questions, especially those regarding the assumptions and methods used, are detailed below.

* L131: What are the limitations of not including these processes in the emulator? Can it capture all feedbacks and dynamics?

* L141-142: "although originally calculated by complex equations, the dynamic evolution of each pool can be described using the first-order model" – why were the complex equations needed initially then? Again, what are the limitations of this first-order model?

* L180: What does the passive pool correspond to (as a measureable pool)? Why is there no transfer from p to s (k_ps)? Why no input to this pool?

* L190: Does this allow for emergent differences in the relative distribution of the three pools with depth? (e.g., relatively more passive C than active C with depth, etc.)

* L196: "The SOC respiration rates for the topsoil layers are equal to those from ORCHIDEE" but how about subsoil respiration? Does the emulator have more respiration overall then? Please clarify how the models compare.

* L207-208: "total global SOC stock is approximately 44% larger than that from the original ORCHIDEE model" – what does this tell us about the accuracy and applicability

of the emulator? This seems to be a big difference. How do the SOC stocks of the two compare to the HWSD and other global SOC databases?

* L209-210: How are these fractions determined? What are the implications of the uncertainty in this partitioning?

* L269: Why "randomly projected"? Please explain how and why.

* L290: But you used CRU-NCEP for ORCHIDEE... what are the caveats of using different climate datasets for each model?

* L297: Why this dataset? How does it compare to the HWSD and SoilGrids (Hengl et al. PLoS ONE 2014, 2017) datasets?

* L342: How uncertain are these numbers given the model formulation assumptions, land-use maps, and methods used? It would help to see a sensitivity analysis and some uncertainty ranges.

* L517: (Section 4.4) with all of these model limitations, it would be nice to have a rough quantification of uncertainties.

Minor comments / Technical corrections:

L24: Add abbreviation "(RUSLE)"

L25: "soil removal only" as opposed to what here?

L27: 100 PgC +/- ? This is a potential, but it would be helpful to know the uncertainty of this number based on your methods and datasets used.

L31: 5 PgC +/- ?

L45: This paragraph discusses erosion as a C sink, while the following paragraph discusses C source mechanisms. There could be a more clear transition here or a sentence that sets this up before the two contrasting paragraphs.

L57: Please elaborate a little on "bookkeeping models"

L59: Although trivial, state "Net Primary Productivity (NPP)" before using just "NPP" in the text.

L68: Drop "to be able"

L75: Change "been applying" to "applied"

L76: Change "different" to "a range of"

L79: Change "While…" to "In contrast, ..." or combine this sentence with the previous one.

L80: Add "Revised Universal Soil Loss Equation (RUSLE)" to this first use of RUSLE in the text.

L112: Change to "(hereafter simply referred to as)"

L119: Be more consistent with PFTs (instead of PFT's) throughout the manuscript.

L144: "based on the stock and output fluxes"

L149: "harvest index (HI)" state this here before the first use of the abbreviation.

L168: "leaves" instead of "leafs" and be consistent with "below-ground" vs. "below-ground"

L204: How does this compare with Hicks Pries et al. Science, 2017 soil respiration with depth?

L202: max, etc. should be defined earlier where it is used in Eqns. 6 and 7.

L243: "such as those covered by"

L257-258: "to preserve the mass balance... and that new substrate..."

L277: Remove extra "the"

L314: Change to "steady-state"
L318: "CO2" with a subscript

L322: Change to "In the first simulation, we allowed the climate and land cover to vary through..."

L329: "descriptions"

L331: "changes"

L335: "steady-state condition"

L349: "despite the" remove "of"

L355: Add "100%" or exact % change to the "almost doubled" to allow easier comparison of the numbers.

L388: "slightly improves" but because poor agreement with global SOC stocks to begin with?

L408: Remove "in his study"

L409-410: Ok, there were limitations in that study, but this makes it sound like you didn't have limitations with your methods.

L468: Replace "only" with "alone" (?)

L475: "between a factor of" add "a"

L479: "significant, making" add a comma
* * *

---

## Author Comment (AC1) · 10 May 2018

We would like to thank the anonymous reviewer for his or her constructive comments. In this response we provide an answer to all the comments and the indicated changes will be applied in the revised manuscript.

Comment 1: "I was not fully convinced by the vertical discretization approach that the emulator used. First of all, different soil layers have totally different biogeophysical and biogeochemical features. Different layers are experiencing different amount of fresh carbon input (e.g., from fine roots exudates, fine root litter), different microbial community (e.g.,fungi/bacteria with different carbon use efficiency), and have different soil

structure (e.g., microaggregate, macroagregate). Secondly, even the idea of summarizing the above-mentioned vertical difference into one single factor (re) is believable, the value of re should be carefully inferred for this model, rather than taking from other studies. Thirdly, and most importantly, the vertical discretization, artificially, increase total global SOC stock by 44%. This type of artifact should be removed. My suggestion is that, since ORCHIDEE has one single soil layer, k0 of ORCHIDEE is supposed to represent the mean turnover rate of the whole soil column. Therefore, the k0 (equation 5) in the emulator (here aims to represent top soil turnover) should not be k0 from ORCHIDEE. One approach is to change k0 in emulator to offset the total SOC stock artifact until it's removed."

Answer: We understand the reviewer's concern regarding the vertical discretization scheme of the emulator and agree that discretizing the SOC over depth is a complex process that has a large impact on the overall SOC stocks. Vertical discretization of SOC has just recently started to be implemented in land surface models. ORCHIDEE is one of the first global land surface models where recently a vertical SOC scheme has been implemented that includes biological production and consumption of SOC, adsorption and desorption processes and diffusion or vertical mixing (Camino-Serrano et al., 2018; Guimberteau et al. 2018). However, for our study we used a simpler version of the ORCHIDEE model without an explicit vertical SOC discretization scheme. One of the main reasons is the need to balance the complexity of the emulator with the computational speed, meaning that including processes such as diffusion would make the emulator with the carbon balance of each layer more complex and slower, complicating the performance of our erosion simulations. One of the primary reasons for using the C emulator was for its speed and simplicity, so that we could clearly separate and quantify the various effects of soil erosion on the SOC stocks.

However, a vertical SOC profile is necessary to account for smaller SOC erosion rates over time due to the generally decreasing SOC concentration with depth on eroding hillslopes (Hoffmann et al., 2013). Without a vertical SOC profile the removal of SOC

by erosion would be most likely overestimated. To simulate the generally declining SOC with depth (on hillslopes) using the emulator, we let the C input to the soil by belowground litter (roots, shoots) decrease exponentially with depth according to the root-profile exponent 'r' (equations 6 and 8b of the original manuscript), and we let the soil respiration rate also decrease exponentially with depth using the exponent 're'. To stay consistent, we use the same values for the exponent 'r' as in the ORCHIDEE model and make sure that the sum of the belowground litter input to each soil layer is equal to the overall belowground litter input to the soil as simulated by ORCHIDEE. Note that vertical injection of litter from roots in the more sophisticated versions of ORCHIDEE cited above also uses the same root-profile factor 'r'. However, as the OR-CHIDEE model version we used has no vertical soil profile, the values for the exponent 're' have to be determined either from literature or calibrated in such a way that the vertical discretization does not influence the total SOC respiration and thus the total SOC stock as simulated by ORCHIDEE.

In the original manuscript we used a constant global value for the exponent 're' derived from a few local studies in a Belgian landscape. At the same time we assumed that the C pool-dependent soil respiration rate of the original ORCHIDEE model is equal to the surface soil respiration rate 'k0i' of the vertical C profile in the emulator (equation 5 of original manuscript). This setup resulted in a much higher SOC stock simulated by the emulator with vertical soil profile than simulated by the original ORCHIDEE model. We agree with the reviewer that this artifact should be removed, as it may intervene with the separation and quantification of the effects by soil erosion on the global SOC stock. By deriving 'k0i' based on the assumption that the average soil respiration rate over the 2m soil profile is equal to the soil respiration rate of ORCHIDEE per grid cell did not result in similar SOC stocks between the emulator and ORCHIDEE in the case of no soil erosion and no land use change (LUC). Actually, it was rarely possible to find a realistic value for 'k0i' per grid cell under a constant global 're' such that the SOC stocks of the emulator would be similar to those of ORCHIDEE (no soil erosion and no LUC).

Therefore, we decided to calibrate both the exponent 're' and variable 'k0i' for each grid cell and PFT under equilibrium conditions. For this calibration we needed information on the ratios between the SOC stocks of the active, slow and passive pools throughout the soil profile. The old vertical discretization scheme resulted in different ratios between the SOC stocks of the three pools with depth. However, there is very little information or data to constrain the pool ratios globally, mainly because the three SOC pools cannot be directly related to measurements (Elliott et al., 1996). Furthermore, neither the emulator nor the ORCHIDEE model we used include soil processes that may affect these pool ratios with depth, such as vertical mixing by soil organisms, diffusion, changes in soil texture (SOC protection and stabilization by clay particles), limitations by oxygen and by access to deep organic matter by microbes. There is also a lot of discussion on how sensitive SOC is to these other processes. For example, the study of Huang et al. (in revision for the journal Advances in Modeling Earth Systems) who implemented a matric-based approach to assess the sensitivity of SOC showed that equilibrium SOC stocks are more sensitive to input than to mixing for soils in the temperate and high-latitude regions. For all the above-mentioned reasons and to decrease the uncertainty we made the assumption that the ratios between the SOC stocks of the active, slow and passive pools are equal throughout the soil profile in the new vertical soil discretization scheme and similar to the pool ratios derived from ORCHIDEE.

For the transient period (1850-2005), we made 're' remain equal to the equilibrium state values, while 'ki0' was derived at a daily time-step to keep to SOC stocks of the emulator similar to those of ORCHIDEE and preserve the yearly variability in the soil respiration rates due to changes in soil climate (no soil erosion and no LUC). Details of how we calibrated the exponent 're' and variable 'k0i' we describe in the supplement.

Method for calibration of 're' and 'k0i' : We start off by selecting a default value for 're' of 2.5 (average between 0 and 5) and then proceed with deriving the values of 'k0i' according to equations 1-4 described in the supplement. After the derivation of 'k0i'

we test if the total SOC stock per grid cell and PFT is similar to that of ORCHIDEE (difference should be smaller than 1 g m-2). If this is not the case we increase or decrease the value of 're', but make sure that it stays within the range of 0 and 5. If these is no optimized solution for both 're' and 'k0i', we use the values that produce the smallest difference in SOC stocks between emulator and ORCHIDEE.

In the transient period (no land use change or erosion) we assume a time-constant 're' fixed to the equilibrium state. Using the mass-balance approach we can find the daily values for k0a, k0s, k0p per grid cell and PFT with equations 8a,b,c (see supplement). In case there was no solution for the 'k0i' at a certain time-step we took the values from the previous time-step.

Implications of the new vertical discretization scheme: After implementing the above-mentioned empirical adjustments to the vertical SOC discretization scheme of the emulator, we found that the resulting SOC stocks for the equilibrium state are close to those of ORCHIDEE with some small deviations (Fig. S1). It was not always possible to precisely match the SOC stocks of the emulator and ORCHIDEE and at the same time have realistic vertical SOC profiles, where the 're' variable varies between 0 and 5.

Figure S2 shows that also for the transient period of simulation S4 (no erosion or LUC) the total SOC stocks are similar between emulator and the ORCHIDEE model. The difference between the SOC stock of the emulator and ORCHIDEE are between -1 and 0% of ORCHIDEE SOC stocks for most grid cells, however, the maximum difference can reach -10% for some grid cells. The total global SOC stock of the emulator in the year 2005 deviates by -0.5% from the SOC stock of ORCHIDEE. This differences between the emulator and ORCHIDEE are due to the fact that there was not always an optimal and realistic solution for 'k0i' and 're'.

Although the new vertical discretization scheme did change the values of the SOC stocks and the related changes to the SOC stocks during the transient period, the

main trends and findings of this study remain the same. In the following paragraph we describe the changes to our results, figures and tables as will be presented in the revised manuscript.

Changes in the manuscript: We will include the above-mentioned derivation of the variables of the new vertical discretization scheme of the emulator and the argumentation behind it, as explained above, in the revised manuscript in chapter 2.3 after line 200 instead of the paragraph between lines 200 and 208. We will include table S1 with the global mean values for the 're' exponent per PFT in the supporting material, together with the figures S1 and S2 showing the deviations in the SOC stocks due to the vertical discretization scheme of the emulator.

We also change the tables 2 to 7 of the original manuscript, where we include the new values for changes in SOC stocks and SOC erosion rates. In table 2 of the original manuscript we found that the values of SOC erosion rates and their standard deviations were presented in the wrong units, so we corrected them. Also table in 3 we corrected the standard deviations of the continental SOC erosion rates, which had the wrong units. Furthermore, we adapted figures 4B,4D,4F,4H, 5C,5D,5E,6A,7,8A,8B,8C according to the new results of the simulations with the new vertical discretization scheme. All the changes in the manuscript are provided at the end of this document.

We make the following changes to the abstract: L26: "We found that over the period 1850-2005 AD acceleration of soil erosion leads to a total potential SOC removal flux of 73 Pg C of which 60% occurs on agricultural, pasture and natural grass lands" L28: "Including soil erosion in the SOC-dynamics scheme results in an increase of 60% of the cumulative loss of SOC..." L30:" This additional erosional loss decreases the cumulative global carbon sink on land by 2 Pg for this specific period,..."

We make the following changes in the results chapter: L343: "This global soil loss flux (here 'loss' meaning horizontal removal by erosion) leads to a total SOC loss flux of 0.52 Pg C y-1..." L347: "The total soil and SOC losses in the year 2005 are an

increase of 14% and 30%, respectively, compared to 1850" L356: " We found that the total soil erosion flux on agricultural land almost doubled by the year 2005 compared to 1850, while the SOC erosion flux increased by 45% (Fig. 4) and led to a cumulative SOC removal of 22 Pg on agricultural land since 1850 (CTR)." L358" On pasture land and grassland, the soil erosion flux increased by only 8.5%, while the SOC erosion flux increased by 50% (Fig. 4) and led to a cumulative SOC mobilization of 38 Pg since 1850." L364: " In total 7183 Pg of soil and 73 Pg of SOC is mobilized across all PFTs by erosion during the period 1850 - 2005, which is equal to approximately 60% of the total net flux of carbon lost as $CO_2$ to the atmosphere due to LUC..." L372: "...we find that the overall global SOC stock decrease during the period 1850 – 2005 would be larger by 60% compared to a world without soil erosion (Fig. 6A)" L374:"... where the total SOC stock shows a net decrease when the effects of soil erosion are taken into account compared to the net increase under no soil erosion" L383:" We calculated a total global SOC stock for 2005 in the absence of soil erosion (S3) of 1284 Pg, which is a factor of 0.85 lower than the total SOC stock from GSCE (Shangguan et al., 2014) for a soil depth of 2m (Table 5)." L386:" Including soil erosion (S1) leads to a total SOC stock of 1001 Pg for the year 2005 (Table 5). We also find that including soil erosion in the SOC-dynamics scheme slightly improves the root mean square error (RMSE) between the simulated SOC stocks and those from GSDE, for the top 30cm of the soil profile. This improvement in the RMSE occurs especially in highly erosive areas." L396:" This flux is paralleled by a SOC loss flux of 0.16 Pg C y-1 after including soil erosion in the CTR simulation (Fig. 4)." L405:" Furthermore, we find a cumulative soil loss of 1888 Pg and cumulative SOC removal flux of 22 Pg from agricultural land over the entire time period (CTR simulation). "

We make the following changes in the discussion chapter: L468:" We find that globally the SOC stocks decrease by 17 Pg due to LUC only during 1850 – 2005 (Fig. 6A, S3-S4). The overall change in carbon over this period summed up over all biomass, litter, SOC, and wood-product pools due to LUC without erosion is a loss of 101 Pg C which lies in the range of cumulative carbon emissions by LUC from estimates of

previous studies (Houghton and Nassikas, 2017; Li et al., 2017; Piao et al., 2009)"
L471: "The increase in soil erosion by expanding agricultural- and grasslands (S1-S2)
amplifies the decrease of SOC stocks implied by LUC in absence of erosion (S3-S4)
by 4 Pg or a factor of 1.2 (Fig. 6A)" L473: "This leads to a total change in the overall
carbon stock on land of -105 Pg." L492: " . . .and leads even to a net cumulative sink of
carbon on land over this period of about 30 Pg C (S3)" L596:" In the presence of soil
erosion, climate variability and the atmospheric $CO_2$ increase lead to a slightly smaller
net cumulative sink of carbon over land of 28 Pg C (S1). . ."

We make the following changes to the conclusion chapter: L551: "This potential soil
loss flux mobilized 73 Pg of SOC across all PFTs, which compares to 60% of the total
net flux of carbon lost as $CO_2$" L553:" When assuming that all this SOC mobilized is
respired we find that the overall SOC change over the period 1850-2005 would increase
by 60%."

Comment 2: "Land use change map. The LUC is prescribed by PFT fractional change
derived from Peng 2017. Wondered how this LUC dataset differs from Land-Use Har-
monization (LUH2), the new CMIP6 land use change dataset. Given that LUC is a
dominant factor of SOC erosion, I am curious about the uncertainty of SOC erosion,
induced by using different LUC estimate (e.g., Peng 2017 vs LUH2)."

Answer: The PFT fractional map is based on LUHv2 land use dataset, historical forest
area data from Houghton (for large regions) and present day forest area from ESA CCI
satellite land cover data (Peng et al., 2017). The historical forest data from Houghton
and the latest satellite land cover data from ESA are the best estimates that currently
exist on forest area. Figure S3 shows that if the forest is not constrained with methods
described by Peng et al. (2017), there is a stronger decrease in forest area over the
period 1850-2005. Also the grassland shows an increasing trend, while in the PFT
map with constrained forest the grassland shows globally a slight decreasing trend. In
the rest of the text we will refer to the PFT map constrained with data on forest area as
the 'constrained PFT map' and to the other PFT map as the 'unconstrained PFT map'.

We agree with the reviewer that different land use data can result in large uncertainties in both SOC stocks and soil erosion rates. To show the potential uncertainty in our results due to uncertainties in underlying land use data we performed 4 additional simulations (S1 to S4) using the unconstrained PFT map and the new vertical discretization scheme.

Differences in global average soil erosion rates between the different PFT maps are small (Fig. S4), mainly because the C-factor of our Adjusted RUSLE model is similar for forest and dense natural grass. As the change in cropland area globally was not very different between the 2 PFT maps, the overall soil erosion rates were similar. We expect, however, that the changes in soil erosion rates between the 2 PFT maps can be significant in areas where the change in forest area was significant over the historical period.

In contrast to the soil erosion rates, the 2 PFT maps resulted in significant differences in the SOC erosion rates and cumulative changes in SOC stocks during the transient period (Fig. S4). The global SOC stock in the equilibrium state without soil erosion (S3) is 8 % higher when the unconstrained PFT map is used, due to a larger global forest area in this map at 1850. The higher global SOC stock of the PFT map without constrained forest area lead to higher SOC erosion rates (Fig. S4b). According to the unconstrained PFT map, soil erosion leads to a total SOC removal of 79 Pg (S1) over the period 1850-2005, which is 6Pg larger than the total SOC removal by soil erosion under the constrained PFT map.

Interestingly, according to the unconstrained PFT map, the global cumulative SOC stock change over 1850-2005 under soil erosion and LUC (S1) is 60% smaller than the stock derived using the constrained PFT map. This is most likely due to the higher forest area at the start of the period A850-2005, leading to a larger increase in SOC stocks by increasing atmospheric $CO_2$ concentrations. The global LUC effect on the SOC stocks of both PFT maps is found to be similar (Fig 5C, D).

[Figure]

Changes in the manuscript: The uncertainty due to different land use maps, as discussed above, will included in chapter 4.4 of the manuscript after line 535, and we will add the above-mentioned changes to the erosion rates and SOC stocks. The new figures S3 and S4 will be included in the supporting material. Specific comment 1: "L16: The first sentence gives me an incorrect hint that the paper is going to talk about agriculture activity accelerates soil erosion"

Answer: We agree that this may be misleading and changed this sentence to: "Anthropogenic land cover change has not only led to large carbon losses but also accelerated soil erosion rates by rainfall and runoff substantially, mobilizing vast quantities of soil organic carbon (SOC) globally.

Specific comment 2: "L38: 1.0 Pg, does it include fire emission?"

Answer: yes it does

Specific comment 3: "L56 what is bookkeeping models?"

Answer: Bookkeeping models are methods/tools to calculate land use change emissions by keeping track of the carbon stored in different pools before and after land use changes take place. These methods also keep track of the $CO_2$ emitted after land use change has taken place. Specific comment 5: "L23 What's the meaning of randomly projected? A more reasonable way is to repeat 1990-1910 climates during 1850-1900."

Answer: "Randomly projected", means that the climate of the years after 1900 was randomly assigned to the years between 1850 and 1900 because the climate data of CRU-NCEP was only available starting from year 1900. If we would choose to repeat the climates of 1900-1910, we would risk including the effects of extreme climate conditions multiple times.

Changes in the manuscript: After line 270 we will add: "This is done because the climate data of CRU-NCEP was only available starting from year 1900, and to avoid the risk of including the effects of extreme climate conditions multiple times if only a

certain decade would be used repeatedly."

Specific comment 6: "L421 "Also, intense soil erosion is typically found in mountainous areas where climate variability has significant impacts, while at the same time these regions are usually poor in SOC." It's not clear in the manuscript whether or not OR-CHIDEE has topography information? In another word, if ORCHIDEE simulates a low SOC stock over the grid cells that have mountains, is that because of the topographical feature of this gridcell can not hold a lot of SOC in ORCHIDEE? Or because of other reasons such as climate constraints (e.g., colder in mountain area)?"

Answer: ORCHIDEE has no soil depth information and thus cannot simulate low SOC stocks due to the fact that the gridcell cannot hold a lot of SOC. Low SOC might however be a result of the plant productivity, the climate (temperature and precipitation), soil moisture and clay content (which is a constant variable). ORCHIDEE has, however, topographical information such as slope that determines the flow directions for water/runoff and affects hydrological parameters such as soil moisture content.

Changes in the manuscript: "Also, intense soil erosion is typically found in mountainous areas where climate variability has significant impacts, while at the same time these regions are usually poor in SOC due to unfavorable environmental conditions for plant productivity."

Specific comment 7: "L465 CO2 fertilization effects on NPP is not fully convincing here, because ORCHIDEE does not have nutrient constraints. OCN might be a better surrogate model to be able to say something about CO2 fertilization effect on NPP."

Answer: In the ORCHIDEE model version we used the nutrients are indeed absent. Our intention, however, was to show the complete picture of possible direct and indirect interactions of soil erosion with the C cycle with the current model setup. The representation of nutrients in global land surface models is new and the related uncertainties are not well quantified. We work with a more or less simple version of ORCHIDEE and the C emulator to be able to understand and quantify the effects of soil erosion on the C

Interactive
comment

cycle. We will mention in chapter 4 of the revised manuscript some of the uncertainties due to the absence of nutrients.

Changes in the manuscript: L537: "The absence of nutrients in the current version of the ORCHIDEE model may result in an overestimation of the $CO_2$ fertilization effect on NPP and may introduce biases in the effect of erosion on SOC stocks under increasing atmospheric $CO_2$ concentrations. Soil erosion may also lead to significant losses of nutrients, especially in agricultural areas. For a more complete quantification of the effects of soil erosion on the carbon cycle, nutrients have to be included in future studies." Specific comment 8: " Figure 4. I do not fully understand why climate change either decrease or not change erosion?

Answer: With climate change we mean temperature and precipitation changes. For soil erosion only precipitation changes are of interest. Globally we find that average yearly precipitation shows a slightly decreasing trend over the period 1950 – 2005 according to the ISIMIP2b dataset used to calculate soil erosion rates. A global smaller total precipitation with respect to 1850 AD will lead to smaller soil erosion rates when LUC is not included. The decrease in total precipitation over land is mostly coming from the tropics, where due to large precipitation amounts a change in precipitation can alter soil erosion significantly. At the same time precipitation is very variable and might not lead to a significant global net change in soil erosion rates over the total period 1850-2005. This result might be contradictory to the fact that major soil erosion events are caused by storms. But in our case we model only rill and interril erosion, which is usually a slow process and previous studies have shown that land use change is usually the main driver of behind accelerated rates of this type of soil erosion. Furthermore, there are very few studies that have quantified the individual effects of precipitation change versus land use change on soil erosion rates over a sufficiently long time period. Therefore, it is difficult to verify this result. However, our soil erosion model performs well for present-day and therefore any possible biases here could be mainly related to biases in precipitation rates, and soil parameters. We agree that this is an interesting point

raised by the reviewer and will add some additional sentences explaining the trend.

Changes in the manuscript: L438: "The global decrease in precipitation in many regions worldwide, especially in the Amazon, as simulated by ISIMIP2b, lead to a slight decrease in soil and SOC erosion rates (Fig. 4). At the same time precipitation is very variable and might not lead to a significant global net change in soil erosion rates over the total period 1850-2005. This result might be contradictory to the fact that major soil erosion events are caused by storms. But in this study we only simulate rill and interril erosion, which are usually slow processes. In addition, previous studies have shown that land use change is usually the main driver behind accelerated rates of these types of soil erosion. Our study confirms this observation."

Please also note the supplement to this comment:
https://www.biogeosciences-discuss.net/bg-2017-527/bg-2017-527-AC1-supplement.pdf

———————————————

[Figure]

Figure S1: The difference in SOC stocks between the emulator and ORCHIDEE as a ratio (%) of the ORCHIDEE SOC stocks for the equilibrium state of the year 1850 without soil erosion. Positive values indicate larger SOC stocks of the emulator compared to the ORCHIDEE model, and negative values indicate smaller SOC stocks of the emulator compared to the ORCHIDEE model.

[Figure]

[Figure]

Figure S2: The difference in SOC stocks between the emulator and
ORCHIDEE as a ratio (%) of the ORCHIDEE SOC stocks averaged
over the period 1996-2005 for simulation S4 (no erosion, no LUC).
The total Positive values indicate larger SOC stocks of the
emulator compared to the ORCHIDEE model, and negative
values indicate smaller SOC stocks of the emulator compared
to the ORCHIDEE model.

[Figure]

Figure 4: There are no significant changes in the historical trends due to the new vertical discretization scheme, except for the overall values.

[Figure]

**Fig. 3.** Figure4
* * *
Interactive
comment

[Figure]

Figure 5: Only slight changes in the spatial variability of the SOC stocks and SOC erosion rates are observed, which are due to the new vertical discretization scheme.

**Fig. 4.** Figure5

[Figure]

Figure 6A: No significant changes in the temporal trends of the
cumulative SOC stock are observed when the new vertical
discretization scheme is used. However, the overall changes in
the cumulative stocks are smaller.

[Figure]

Figure 7: The cumulative SOC stock change in the original manuscript was
wrongly projected and is corrected here. The grassland SOC stocks should
decrease instead of increase. Overall, the historical trends in the cumulative
SOC stocks follow closely the changes in the respective vegetation fractions.
The overall changes in SOC stocks are smaller here compared to the figure in
the original manuscript due to the new vertical scheme.

**Fig. 5.** Figure 6A and 7

[Figure]

Difference in SOC stock change under all perturbations including soil erosion and SOC stock change under all perturbations without soil erosion

Difference in SOC stock change under LUC and soil erosion and SOC stock change under LUC

Difference in SOC stock change under CC and soil erosion and SOC stock change under CC only

Figure 8: no significant changes are observed in the plots due to the new vertical scheme.

[Figure]

Figure S3: Changes (%) in forest (green), grass (light-green) and crop (red) fractions over the period 1850-2005 with respect to the year 1850AD. The dashed lines represent data from the PFT map without constrained forest, while the straight lines represent data from the PFT map from Peng et al. (2017).

[Figure]

Figure S4: Differences in (A) soil and (B) SOC erosion rates and (C and D) cumulative SOC stock changes between the PFT map that is constrained by forest area data (straight lines) and the PFT map that is not constrained by forest area data (dashed lines).

Table S1: Global area-weighted average '*re*' values per PFT

|  | PFT description | Area-weighted mean re |
|---|---|---|
| 0 | Bare soil | 2.5 |
| 1 | Tropical broad-leaved evergreen | 0.72 |
| 2 | tropical broad-leaved raingreen | 1.33 |
| 3 | temperate needleleaf evergreen | 1.29 |
| 4 | temperate broad-leaved evergreen | 1.25 |
| 5 | Temperate broad-leaved summergreen | 1.09 |
| 6 | boreal needleleaf evergreen | 1.33 |
| 7 | boreal broad-leaved summergreen | 1.13 |
| 8 | boreal needleleaf summergreen | 0.93 |
| 9 | C3 grass | 2.03 |
| 10 | C4 grass | 2.3 |
| 11 | C3 agriculture | 0.73 |
| 12 | C4 agriculture | 0.78 |

*The tables of the original manuscript are changed as following:*

Table2: only SOC erosion rates and their standard deviations are corrected

| PFT | Mean soil erosion $(t\ ha^{-1}\ y^{-1})$ | Standard deviation soil erosion $(t\ ha^{-1}\ y^{-1})$ | Mean SOC erosion $(kg\ C\ ha^{-1}\ y^{-1})$ | Standard deviation SOC erosion $(kg\ C\ ha^{-1}\ y^{-1})$ |
|---|---|---|---|---|
| Crop | 1.71 | 24.95 | 288.17 | 43563 |
| Grass | 1.88 | 32.67 | 83.52 | 4613 |
| Forest | 0.26 | 2.31 | 12.36 | 326 |

Table 3: only SOC erosion rates, their standard deviations and changes are corrected

[revised manuscript text omitted]

In the transient period (no land use change or erosion) we assume a time-constant '*re*' fixed to the equilibrium state. Using the mass-balance approach we can find the daily values for $k_{0a}$, $k_{0s}$, $k_{0p}$ per grid cell and PFT with:

$$\frac{dSOC_a}{dt} = \sum_{z=0}^{z=n}(L_a(z,t) + k_{sa} * soil_s(z,t-1) + k_{pa} * soil_p(z,t-1) - (k_{0a}(t) * e^{-re*z} +$$

$$k_{as} + k_{ap}) * soil_a\ (z,t-1)) \tag{8a}$$

$$\frac{dSOC_s}{dt} = \sum_{z=0}^{z=n}(L_s(z,t) + k_{as} * soil_a(z,t-1) - (k_{0s}(t) * e^{-re*z} + k_{sa} + k_{sp}) * soil_s\ (z,t-$$

$$1)) \tag{8b}$$

$$\frac{dSOC_p}{dt} = \sum_{z=0}^{z=n}(k_{sp} * soil_s(z,t-1) + k_{ap} * soil_a(z,t-1) - (k_{0p}(t) * e^{-re*z} + k_{pa}) *$$

$$soil_p\ (z,t-1)) \tag{8c}$$

In case there was no solution for the '*k0i*' at a certain time-step we took the values from the previous time-step.

---

## Author Comment (AC2) · 10 May 2018

We would like to thank the anonymous reviewer for his or her constructive comments. In this response we provide an answer to all the comments and the indicated changes will be applied in the revised manuscript.

Comment 1: "The emulator used in this study seems to have various limitations that make the numbers presented quite uncertain – further discussion on, and quantification of, these uncertainties is warranted and would greatly improve this manuscript. Specifically, I would have liked to see additional support for the SOC model formulation, parameters, and built-in feedbacks chosen for the emulator, as well as support

for its vertical discretization and parameterization. The carbon emulator is supposed to describe the carbon pools and fluxes exactly as in ORCHIDEE, yet the total global SOC stocks from the emulator are 44% higher than that of the original ORCHIDEE model. This is a big difference. What does this tell us about the accuracy and applicability of the emulator, and how do the SOC stocks of the two models compare to the Harmonized World Soil Database (HWSD) and other global SOC databases? Additional major comments/questions, especially those regarding the assumptions and methods used, are detailed below.

Answer: We modified the vertical discretization scheme of the emulator in such a way that the total SOC stock of each grid cell, PFT and C pool is close to that of ORCHIDEE when soil erosion and land use change is deactivated (0.5% max difference in total global SOC stock). For this we calibrated both the exponent 're' and variable 'k0i' of equation 5 in the manuscript for each grid cell and PFT under equilibrium conditions, such that the total soil respiration per grid cell, PFT, and soil C pool of the emulator would be similar to that of the ORCHIDEE model. For the transient period (1850-2005), we made 're' remain equal to the equilibrium state values, while values for 'ki0' were derived at a daily time-step to keep to SOC stocks of the emulator similar to those of ORCHIDEE and preserve the yearly variability in the soil respiration rates due to changes in soil climate (soil erosion and land use change were deactivated ). Details of how we calibrated the exponent 're' and variable 'k0i' we describe in our response to Reviewer 1. The modified vertical discretization scheme did change the values of the SOC stocks and SOC removal rates, because with this scheme we simulated total SOC such as in the ORCHIDEE model without soil erosion and land use change. However, the overall trends in soil and SOC erosion rates and cumulative changes in SOC stocks during the transient period did not change significantly and the main findings of our study remain unchanged. For more details on the changes in our manuscript related to the modified vertical discretization scheme see our detailed response to reviewer 1.

We performed a comparison of our simulated SOC stocks to the Global Soil Database

for Earth System Modeling (GSDE), as is described in paragraph 3.2 and the new table 5 of the manuscript (with results based on the new vertical discretization scheme). However, we abstained from a more in-depth comparison as our emulator and ORCHIDEE do not include various soil processes that have been proven to affect SOC substantially such as vertical mixing, diffusion, priming, changes in soil texture, C rich organic soils formation, etc.

It should be noted that there are also large uncertainties in the global soil databases (Hengl et al., 2014; Scharlemann et al., 2014; Tifafi et al., 2018), which makes the exact quantification of the uncertainties of the resulting SOC dynamics simulated by our emulator difficult. After applying the modifications to the vertical SOC discretization scheme, we performed a simulation with soil erosion and land use change activated (S1) and compared the resulting SOC stocks with those from the GSDE. Figure S1 shows that our emulator, and the ORCHIDEE model in general, underestimates SOC stocks globally, except for the high-latitudes.

Changes in manuscript: We will put figure S1 in the supporting material. Furthermore, the values of table 5 of the manuscript are modified and the new table that will be included in the manuscript is presented in the supplement. L394:" We abstained from a more in-depth comparison as our emulator and ORCHIDEE do not include various soil processes that have been proven to affect SOC substantially such as vertical mixing, diffusion, priming, changes in soil texture, C rich organic soils formation, etc. It should be noted that there are also large uncertainties in the global soil databases (Hengl et al., 2014; Scharlemann et al., 2014; Tifafi et al., 2018), which makes the exact quantification of the uncertainties of the resulting SOC dynamics simulated by our emulator difficult."

Answer: Some of these processes are already included in the ORCHIDEE model, which is the basis for the C emulator but other feedbacks on SOC are missing in the original ORCHIDEE model such as the effect of SOC on the hydrology or on the thermic of the model. Nevertheless, our main objective here was to present a tool able to

evaluate erosion fluxes at global scale using a 'state-of-art' land surface model outputs and estimate the drivers of erosion at global scale. In addition, this study did not focus on the feedbacks of soil erosion and land use change on NPP, the hydrological cycle or nutrient cycle and therefore it was decided not to incorporate soil erosion processes directly into ORCHIDEE, but rather use the C emulator concept instead. Not including these processes explicitly in the emulator does not change the simulated SOC dynamics in our study. However, the emulator has a flexible structure and could be made more complex depending on the needs, such as including a more sophisticated vertical discretization scheme. The main idea of the emulator was to use a modeling tool that does not require much computational power but that still incorporates the basic processes and variables for simulating large-scale SOC dynamics under soil erosion and land use change. Many simulations were needed to quantify the various effects of soil erosion on the C cycle and to calibrate the model parameters. The C emulator was in this case a convenient tool, as it is fast and its structure allows to easy switch processes on or off.

Changes in manuscript: In chapter 2.1 we will rewrite the text between the lines 131 and 136 as following: "...in the emulator. Some of these processes are already included in the ORCHIDEE model, which is the basis for the C emulator. In addition, this study did not focus on the feedbacks of soil erosion and land use change on NPP, the hydrological cycle or nutrient cycle and therefore it was decided not to incorporate soil erosion processes directly into ORCHIDEE, but rather use the C emulator concept instead. Not including these processes explicitly in the emulator does not change the simulated SOC dynamics in our study. The emulator preserves the structure of the carbon cycle of ORCHIDEE and is able to reproduce the outputs exactly as by the full ORCHIDEE model (Fig 1A). At face value, the emulator merely copies the ORCHIDEE carbon pool dynamics, and for each new atmospheric $CO_2$- and climate scenario a new run of the original LSM is required. Our main objective here was to present a tool able to evaluate erosion fluxes at global scale using a 'state-of-art' land surface model outputs and estimate the drivers of erosion at global scale."

Specific comment 2: "L141-142: although originally calculated by complex equations, the dynamic evolution of each pool can be described using the first-order model" – why were the complex equations needed initially then? Again, what are the limitations of this first-order model?

Answer: The limitations of this first-order model are the incapability to capture feedbacks on the hydrological processes or on the NPP (see answer to previous comment). However, because the SOC is represented by first order equations inside ORCHIDEE and the complex equations only compute the modifier to the default coefficients, and as our study focused on the effects of soil erosion on the SOC dynamics, we decided to use the C emulator, assuming that erosion will not significantly impact soil physics (and in turn decomposition) affecting SOC. The complex equations, such as photosynthesis and hydrological processes are needed to simulate realistically the changes in biomass, litter and soil respiration over time, which is done by ORCHIDEE. In the original ORCHIDEE simulations, these processes are explicitly simulated on a 30 min time step. Such a time step is needed for coupled simulations with a climate model, but makes the model CPU intensive, and there is no need for such high-resolution calculations of 'fast' C fluxes for erosion induced effects on SOC. In the emulator, all C fluxes between ecosystem compartments (with and without erosion) are exactly the same as the original ORCHIDEE, assuming that there is no feedbacks between erosion and these fluxes. The C emulator is much more computational efficient than the original ORCHIDEE because it does not require to compute all 'fast' processes for all simulations. The emulator thus allows us to conduct a lot of simulations (e.g. with and without climate change, with and without $CO_2$ fertilization, with and without land use change, with and without erosion), and at the same time keep the main features (except erosion) of the original ORCHIDEE simulation.

Changes in manuscript: L143:"…Eq.1. The complex equations, such as photosynthesis and hydrological processes are needed to simulate realistically the changes in biomass, litter and soil respiration over time, which is done by ORCHIDEE. In the original ORCHIDEE simulations, these processes are explicitly simulated on a 30 min time step. Such a time step is needed for coupled simulations with a climate model, but makes the model CPU intensive, and there is no need for such high-resolution calculations of 'fast' C fluxes for erosion induced effects on SOC. In the emulator, all C fluxes between ecosystem compartments (with and without erosion) are exactly the same as the original ORCHIDEE, assuming that there is no feedbacks between erosion and these fluxes. Based on the output stock and fluxes of the original ORCHIDEE model, the values of the turnover rates are calculated and archived together with the input fluxes. They are then used..."

  Specific comment 3: "L180: What does the passive pool correspond to (as a measureable pool)? Why is there no transfer from p to s (k_ps)? Why no input to this pool?"

Answer: The distribution of SOC into an active, slow and passive pool and the transfer rates between these pools are based on the work of Parton et al. (1988). These pools are defined by their different residence times. The active, slow and passive SOC pools have a residence time of 1.5, 25 and 1000 years, respectively. That study defines the passive pool as a pool that is very resistant to decomposition and includes physically and chemically stabilized SOM. The proportions of the decomposition products which enter the passive pool from the slow and active pools increase with increasing soil clay content. Passive C is thus not directly produced from litter input but active or slow C has to be stabilized first to become passive C. Then, the original model of Parton et al., (1988) assumes that when the passive pool is decomposed by microorganisms, they produce metabolites corresponding to more labile materials that are released in the soil solution during microbial death and the associated cell lysis. For these reasons, they considered that the decomposed passive pool can only by recycled into the active pool.

Changes in manuscript: L181: "The SOC pools are based on the study of Parton et al. (1988) and are defined by their residence times. The active SOC pool has the lowest

residence time (∼1.5 years) and the passive the highest (1000 years)."

Specific comment 4: "L190: Does this allow for emergent differences in the relative distribution of the three pools with depth? (e.g., relatively more passive C than active C with depth, etc.)

Answer: Yes, the old vertical discretization scheme allowed for different relative distributions of the three pools with depth. However, we changed this aspect by assuming that the ratios between the three pools do not change with depth as we have no data on how these ratios should change (see reply to rev #1). In addition, we do not simulate the underlying processes that would allow for changing ratios between the SOC pools such as changing clay content with depth, diffusion, bioturbation.

Specific comment 5: "L196: "The SOC respiration rates for the topsoil layers are equal to those from ORCHIDEE". But how about subsoil respiration? Does the emulator have more respiration overall then? Please clarify how the models compare."

Answer: We modified the vertical discretization scheme, so that the emulator now has a similar SOC respiration rate as ORCHIDEE without soil erosion or land use change. See our response to the first comment and to the comments of reviewer 1.

Specific comment 6: "L207-208: "total global SOC stock is approximately 44% larger than that from the original ORCHIDEE model" – what does this tell us about the accuracy and applicability of the emulator? This seems to be a big difference. How do the SOC stocks of the two compare to the HWSD and other global SOC databases?"

Answer: We modified the vertical discretization scheme, where the emulator has similar SOC stocks as ORCHIDEE without soil erosion or land use change. For more details see our response to the first comment of reviewer 1.

Specific comment 7: "L209-210: How are these fractions determined? What are the implications of the uncertainty in this partitioning?"

Answer: Above and below-ground litter consists out of plant residues and organic animal excreta that are partitioned into structural and metabolic pools as a function of the lignin to N ratio in the residue (Parton et al., 1988). The lignin and N ratios are usually prescribed per PFT and derived from plant-trait databases. This partitioning is prescribed by Parton et al. (1988) and followed by Krinner et al. (2005). The structural litter pool has a slower decay rate and contains the more recalcitrant molecules, while the metabolic pool has a faster decay rate and contains labile plant material. The decay rates are a function of temperature and humidity (Krinner et al., 2005). The lignin fraction of the plant material does not go through the active pool but is assumed to go directly to the slow C pool as the structural plant material decomposes. This is why part of the decomposed structural litter pool goes the active SOC pool and another to the slow SOC pool. Metabolic litter can be decomposed into active SOC and could also form a mineral-stabilized SOC (slow SOC pool, Cotrufo et al., 2015). The CENTURY model simulates the dynamics of C and nutrients (Parton et al., 1988), and is widely applied and tested in Land Surface Models. There are definitely large uncertainties in the partitioning of the litter pools, however, it is not in the scope of this paper to discuss these uncertainties.

Changes in manuscript: L211: "These litter fractions are based on the Century model as introduced by Parton et al.(1988) and later implemented inside ORCHIDEE (Krinner et al., 2005).."

Specific comment 8: "L269: Why "randomly projected"? Please explain how and why."

Answer: "Randomly projected", means that the climate of the years after 1900 was randomly assigned to the years between 1850 and 1900 because the climate data of CRU-NCEP was only available starting from year 1900. If we would choose to repeat for example the climates of 1900-1910, we would risk including the effects of extreme climate conditions multiple times, which is not the case when a random projection is used. Changes in the manuscript: After line 270 we will add: "This is done because the climate data of CRU-NCEP was only available starting from year 1900, and to avoid the risk of including the effects of extreme climate conditions multiple times if only a

certain decade would be used repeatedly."

Specific comment 9: "L290: But you used CRU-NCEP for ORCHIDEE... what are the caveats of using different climate datasets for each model?"

Answer: We compared the historical trend in yearly total precipitation between CRU-NCEP and ISIMIP2b, see figure S2. We find that although the ISIMIP2b shows a higher overall precipitation amount, the temporal trend and variability are similar to that of CRU-NCEP. If we would use CRU-NCEP to calculate the soil erosion rates, we expect that the new soil erosion rates would fall inside the uncertainty range created by calculation of the R- and the C-factors of the Adj. RUSLE (see answer to the last comment).

Specific comment 10: "L297: Why this dataset? How does it compare to the HWSD and SoilGrids (Hengl et al. PLoS ONE 2014, 2017) datasets?"

Answer: The GSDE is based on the SoilMap of the World (FAO, 1995, 2003) and various regional and national soil databases. It is available at a 1km resolution and at 5 arcmin resolution and contains updated soil information and more soil variables such as nutrients. The GSDE is based on more regional soil maps and is more up to date on soil information than the HWSD but both products compared relatively well since they shared several data (Shangguan et al., 2014). We did not test the SoilGrids data, which is based on a different approach. A recent publication showed that SoilGrids give different results compared to HWSD (Tifafi et al., 2018) but regarding the difference between the products we decided to use only one of them already used to evaluate erosion process and then be more comparable with previous publications (Naipal et al., 2015, 2016).

Specific comment 11: L342: How uncertain are these numbers given the model formulation assumptions, land-use maps, and methods used? It would help to see a sensitivity analysis and some uncertainty ranges.

Answer: We performed 4 additional simulations with a different PFT map, which is also based on the LUH2 land use dataset but where the historical forest area change that is not constrained by data as done by Peng et al. (2017). We used these simulations to show the differences to our results when other land use maps are used. If the forest is not constrained with methods described by Peng et al. (2017), there is a stronger decrease in forest area over the period 1850-2005. Also the grassland shows an increasing trend, while in the PFT map with constrained forest the grassland shows globally a slight decreasing trend. In the rest of the text we will refer to the PFT map constrained with data on forest area as the 'constrained PFT map' and to the other PFT map as the 'unconstrained PFT map'. Differences in global average soil erosion rates between the different PFT maps are small, however, there are significant differences in the SOC erosion rates and cumulative changes in SOC stocks during the transient period. According to the unconstrained PFT map, soil erosion leads to a total SOC removal of 79 Pg (simulation S1 with the new vertical discretization scheme) over the period 1850-2005, which is 6Pg larger than the total SOC removal by soil erosion under the constrained PFT map. Interestingly, according to the unconstrained PFT map, the global cumulative SOC stock change over 1850-2005 under soil erosion and LUC (S1) is 60% smaller than the stock derived using the constrained PFT map. This is most likely due to the higher forest area at the start of the period A850-2005, leading to a larger increase in SOC stocks by increasing atmospheric $CO_2$ concentrations. The global LUC effect on the SOC stocks of both PFT maps is found to be similar. For more details and our changes in the manuscript see our answer to comment 2 of reviewer 1.

Specific comment 12: "L517: (Section 4.4) with all of these model limitations, it would be nice to have a rough quantification of uncertainties."

Answer: We agree with the reviewer that quantifying the uncertainty is important. Therefore, we derived an uncertainty range for our soil erosion rates. First, we varied the R-factor of the Adj.RUSLE model between a maximum and a minimum based on the regression equations derived by Naipal et al. (2015) per climate zone. Then

we varied the C-factor of the Adj.RUSLE model between a maximum and minimum value per land cover type (tree, crop or grass) based on literature. We then used the uncertainty range in the C and R factors to derive the uncertainty range in the soil erosion rates and subsequently in the SOC erosion rates. We performed 4 additional simulations with the emulator, 2 simulations with the setup of S1 and a minimum and maximum soil erosion scenario, and 2 simulations with the setup of S2 with a maximum and minimum soil erosion scenario. The changes in our manuscript due to these uncertainty ranges and the new vertical discretization scheme are described below.

Changes in the manuscript: We make the following changes to the abstract: L26: "We found that over the period 1850-2005 AD acceleration of soil erosion leads to a total potential SOC removal flux of 73±17 Pg C of which 80-97% occurs on agricultural, pasture and natural grass lands" L28: "Including soil erosion in the SOC-dynamics scheme results in an increase of 54-70% of the cumulative loss of SOC..." L30:" This additional erosional loss decreases the cumulative global carbon sink on land by about 2 Pg for this specific period,..."

We make the following changes in the results chapter: L341-346:" After including soil erosion in the ORCHIDEE emulator we obtain a total global soil loss flux of 47.6±10 Pg y-1 for the year 2005 of which about 20-29% is attributed to agricultural land and 51-55% to grassland (natural grass and pasture). This global soil loss flux (here 'loss' meaning horizontal removal by erosion) leads to a total SOC loss flux of 0.52±0.14 Pg C y-1 of which about 26-33% are attributed to agricultural land and 54-64% to grassland (CTR, Fig 4)." L347:" The total soil and SOC losses in the year 2005 are an increase of 11-19% and 23-35%, respectively,..." L355-358:" We found that the total soil erosion flux on agricultural land increased with 55-60% by the year 2005 compared to 1850, while the SOC erosion flux increased by 11-70% (Fig. 4) and led to a cumulative SOC removal of 22±5 Pg on agricultural land since 1850 (CTR). On pasture land and grassland, the soil erosion flux increased by only 8-20%, while the SOC erosion flux increased by 44-54% (Fig. 4) and led to a cumulative SOC mobilization of 38±7

Pg since 1850." It is evident that on agricultural land the uncertainty range in soil erosion leads to a large uncertainty range in SOC erosion compared to grassland. The increase in SOC erosion is much larger than the increase in soil erosion on grasslands because in our model, LUC (without erosion) leads to a significant increase in SOC on grassland, which amplifies the increasing trend in SOC erosion for grassland. This simulated increase for SOC stocks on grasslands..." L364-365: "In total 7183±1662 Pg of soil and 73±17 Pg of SOC is mobilized across all PFTs by erosion during the period 1850 - 2005, which is equal to approximately 46-74% of the total net flux of carbon lost as CO2 to the atmosphere due to LUC ..." L395-397: "Using the Adj.RUSLE model to estimate agricultural soil loss by water erosion for the year 2005 resulted in a global soil loss flux of 12.28±4.62 Pg y-1 (Fig. 4). This flux is paralleled by a SOC loss flux of 0.16±0.06 Pg C y-1 after including soil erosion in the CTR simulation (Fig. 4). " L405-406: "Furthermore, we find a cumulative soil loss of 1888±753 Pg and cumulative SOC removal flux of 22±5 Pg from agricultural land over the entire time period (CTR simulation)."

We make the following changes in the discussion chapter: L473-474:" ...absence of erosion (S3-S4) by 4±2 Pg or a factor of 1.1-1.4 (Fig. 6A). This leads to a total change in the overall carbon stock on land of -105±2 Pg." L493:" ...and leads even to a net cumulative sink of carbon on land over this period of about 30±2 Pg C (S3)."

We make the following changes to the conclusion chapter: L551: "This potential soil loss flux mobilized 73±17 Pg of SOC across all PFTs, which compares to 46-74% of the total net flux of carbon lost as CO2" L553:" When assuming that all this SOC mobilized is respired we find that the overall SOC change over the period 1850-2005 would increase by 54-70%."

The new tables with uncertainty ranges are provided in the supplementary material.

Please also note the supplement to this comment:
https://www.biogeosciences-discuss.net/bg-2017-527/bg-2017-527-AC2-

supplement.pdf

[Figure]

[Figure]

Figure S1: Difference between SOC stocks of the emulator
(simulation S1) and the SOC stocks of GSDE as a percentage of the
SOC stocks of GSDE till 2m depth. Red colors show larger SOC
stocks by the emulator, while blue colors indicate smaller SOC stocks
by the emulator compared to GSDE.

[Figure]

[Figure]

Figure S2: Temporal trend in yearly total, global average
precipitation derived from ISIMIP2b ( staright line) and
CRU-NCEP (dashed line).

**Supplement:**

Table 5

| Soil depth (m) | GSDE SOC total (Pg) | S1 SOC total (Pg) | S3 SOC total (Pg) | RMSE S1 | RMSE S3 | r –value S1 | r – value S3 |
|---|---|---|---|---|---|---|---|
| 0.3 | 670 | 428 | 556 | 5218 | 5861 | 0.43 | 0.44 |
| 1 | 1356 | 672 | 846 | 14077 | 10213 | 0.52 | 0.51 |
| 2 | 1748 | 1001 | 1284 | 12968 | 13195 | 0.56 | 0.55 |

*The tables of the original manuscript are changed as following:*

Table2

| PFT | Mean soil erosion (t ha$^{-1}$ y$^{-1}$) | Standard deviation soil erosion (t ha$^{-1}$ y$^{-1}$) | Mean SOC erosion (kg C ha$^{-1}$ y$^{-1}$) | Standard deviation SOC erosion (kg C ha$^{-1}$ y$^{-1}$) |
|---|---|---|---|---|
| Crop | 1.71±0.66 | 24.95±12.44 | 288.17±64.86 | 43563±11512 |
| Grass | 1.88±0.46 | 32.67±8.58 | 83.52±15.94 | 4613±1231 |
| Forest | 0.26±0.13 | 2.31±1.68 | 12.36±4.26 | 326±92 |

Table 3: only SOC erosion rates, their standard deviations and changes are corrected

| Region | Mean soil erosion rate | Standard deviation soil erosion rate | Change in mean soil erosion rate | Mean SOC erosion rate | Standard deviation SOC erosion rate | Change in mean SOC erosion rate |
|---|---|---|---|---|---|---|
| | 2005 | 2005 | 2005 -1851 | 2005 | 2005 | 2005-1851 |
| | (t ha$^{-1}$ y$^{-1}$) | (t ha$^{-1}$ y$^{-1}$) | (t ha$^{-1}$ y$^{-1}$) | (kg C ha$^{-1}$ y$^{-1}$) | (kg C ha$^{-1}$ y$^{-1}$) | (kg C ha$^{-1}$ y$^{-1}$) |
| Africa | 2.69±0.75 | 68.47±21.84 | 0.69±0.12 | 13.17±5.01 | 100.56±69.56 | 4.31±3.92 |
| Asia | 6.03±1.31 | 167.83±41.62 | 0.23±0.12 | 57.85±14.79 | 831.7±354.05 | 3.22±8.21 |
| Europe | 2.45±0.36 | 73.7±12.9 | 0.48±0.12 | 16.67±3.37 | 347.74±91.1 | 1.39±0.99 |
| Australia | 1.46±0.42 | 16.98±5.66 | -0.5±0.11 | 5.14±1.27 | 23.14±7.45 | 1.81±0.68 |
| South-America | 4.69±1.19 | 117.58±33.24 | 1.35±0.42 | 74.27±21.68 | 1552.04±537.61 | 38.89±16.69 |
| North-America | 2.83±0.55 | 63.68±11.06 | 0.15±0.06 | 32.85±6.24 | 571.24±91.69 | 3.14±2.66 |
| Global | 3.92±0.89 | 104.48±26.99 | 0.5±0.07 | 38.49±10.12 | 690.96±264.47 | 8.92±6.31 |

Table 4

| Region | Change SOC stocks S1 | Change SOC stocks S2 | Change SOC stocks S1-S2 | Change SOC stocks S3 | Change SOC stocks S4 | Change SOC stocks S3-S4 |
|---|---|---|---|---|---|---|
| | Pg C | Pg C | Pg C | Pg C | Pg C | Pg C |
| Africa | -1.55±0.01 | -0.24±0.03 | -1.31±0.04 | -1.54 | -0.55 | -0.98 |
| Asia | -0.36±0.77 | 7.94±0.001 | -8.31±0.77 | 0.65 | 7.11 | -6.47 |
| Europe | -3.33±0.91 | 1.78±0.11 | -5.12±0.81 | -4.35 | 1.52 | -5.87 |
| Australia | 0.21±0.01 | 0.05±0.003 | 0.16±0.01 | 0.29 | 0.01 | 0.28 |
| South-America | 2.24±0.22 | 2.82±0.07 | -0.59±0.29 | 3.75 | 2.06 | 1.69 |
| North-America | -2.62±0.10 | 3.19±0.05 | -5.81±0.95 | -2.5 | 3.03 | -5.53 |
| Global | -5.35±0.27 | 15.93±0.21 | -21.29±2.48 | -3.3 | 13.86 | -17.16 |

Table 7: only SOC fluxes from our study are corrected

| Region | Our Study | | Doetterl et al. (2012) | |
|---|---|---|---|---|
| | Sediment flux 2005 Pg y$^{-1}$ | SOC flux 2005 Tg C y$^{-1}$ | Sediment flux 2000 Pg y$^{-1}$ | SOC flux 2000 Tg C y$^{-1}$ |
| Africa | 2.6 | 20.29±8.28 | 2.4 | 39.5 |
| Asia | 5.4 | 54.34±20.63 | 4.9 | 90.0 |
| Europe | 2.1 | 30.8±6.40 | 1.9 | 39.5 |
| Australia | 0.2 | 2.45±0.52 | 0.3 | 4.3 |
| South-America | 1.6 | 39.03±15.74 | 1.4 | 26.7 |
| North-America | 0.7 | 12.37±3.39 | 1.6 | 31.5 |
| Total | 12.6 | 161.53±55.74 | 12.5 | 231.5 |

---

## Author Response (AR1)

**Author's Response to Reviewer 1**

We would like to thank the anonymous reviewer for his or her constructive comments. In this response we provide an answer to all the comments and the indicated changes will be applied in the revised manuscript.

**Comment 1:** *"I was not fully convinced by the vertical discretization approach that the emulator used. First of all, different soil layers have totally different biogeophysical and biogeochemical features. Different layers are experiencing different amount of fresh carbon input (e.g., from fine roots exudates, fine root litter), different microbial community (e.g.,fungi/bacteria with different carbon use efficiency), and have different soil structure (e.g., microaggregate, macroagregate).*

*Secondly, even the idea of summarizing the above-mentioned vertical difference into one single factor (re) is believable, the value of re should be carefully inferred for this model, rather than taking from other studies.*

*Thirdly, and most importantly, the vertical discretization, artificially, increase total global SOC stock by 44%. This type of artifact should be removed. My suggestion is that, since ORCHIDEE has one single soil layer, k0 of ORCHIDEE is supposed to represent the mean turnover rate of the whole soil column. Therefore, the k0 (equation 5) in the emulator (here aims to represent top soil turnover) should not be k0 from ORCHIDEE. One approach is to change k0 in emulator to offset the total SOC stock artifact until it's removed."*

**Answer:** We understand the reviewer's concern regarding the vertical discretization scheme of the emulator and agree that discretizing the SOC over depth is a complex process that has a large impact on the overall SOC stocks. Vertical discretization of SOC has just recently started to be implemented in land surface models. ORCHIDEE is one of the first global land surface models where recently a vertical SOC scheme has been implemented that includes biological production and consumption of SOC, adsorption and desorption processes and diffusion or vertical mixing (Camino-Serrano et al., 2018; Guimberteau et al. 2018). However, for our study we used a simpler version of the ORCHIDEE model without an explicit vertical SOC discretization scheme. One of the main reasons is the need to balance the complexity of the emulator with the computational speed, meaning that including processes such as diffusion would make the

emulator with the carbon balance of each layer more complex and slower, complicating the performance of our erosion simulations. One of the primary reasons for using the C emulator was for its speed and simplicity, so that we could clearly separate and quantify the various effects of soil erosion on the SOC stocks.

However, a vertical SOC profile is necessary to account for smaller SOC erosion rates over time due to the generally decreasing SOC concentration with depth on eroding hillslopes (Hoffmann et al., 2013). Without a vertical SOC profile the removal of SOC by erosion would be most likely overestimated.

To simulate the generally declining SOC with depth (on hillslopes) using the emulator, we let the C input to the soil by belowground litter (roots, shoots) decrease exponentially with depth according to the root-profile exponent *'r'* (equations 6 and 8b of the original manuscript), and we let the soil respiration rate also decrease exponentially with depth using the exponent *'re'*. To stay consistent, we use the same values for the exponent *'r'* as in the ORCHIDEE model and make sure that the sum of the belowground litter input to each soil layer is equal to the overall belowground litter input to the soil as simulated by ORCHIDEE. Note that vertical injection of litter from roots in the more sophisticated versions of ORCHIDEE cited above also uses the same root-profile factor '*r*'. However, as the ORCHIDEE model version we used has no vertical soil    profile, the values for the exponent *'re'* have to be determined either from literature or calibrated in such a way that the vertical discretization does not influence the total SOC respiration and thus the total SOC stock as simulated by ORCHIDEE.

In the original manuscript we used a constant global value for the exponent '*re*' derived from a few local studies in a Belgian landscape. At the same time we assumed that the C pool-dependent soil respiration rate of the original ORCHIDEE model is equal to the surface soil respiration rate '$k_{0i}$' of the vertical C profile in the emulator (equation 5 of original manuscript). This setup resulted in a much higher SOC stock simulated by the emulator with vertical soil profile than simulated by the original ORCHIDEE model. We agree with the reviewer that this artifact should be removed, as it may intervene with the separation and quantification of the effects by soil erosion on the global SOC stock. By deriving '$k_{0i}$' based on the assumption that the average soil respiration rate over the 2m soil profile is equal to the soil respiration rate of ORCHIDEE per grid cell did not result in similar SOC stocks between the emulator and

ORCHIDEE in the case of no soil erosion and no land use change (LUC). Actually, it was rarely possible to find a realistic value for '$k_{0i}$' per grid cell under a constant global '$re$' such that the SOC stocks of the emulator would be similar to those of ORCHIDEE (no soil erosion and no LUC).

Therefore, we decided to calibrate both the exponent '$re$' and variable '$k_{0i}$' for each grid cell and PFT under equilibrium conditions. For this calibration we needed information on the ratios between the SOC stocks of the active, slow and passive pools throughout the soil profile. The old vertical discretization scheme resulted in different ratios between the SOC stocks of the three pools with depth. However, there is very little information or data to constrain the pool ratios globally, mainly because the three SOC pools cannot be directly related to measurements (Elliott et al., 1996). Furthermore, neither the emulator nor the ORCHIDEE model we used include soil processes that may affect these pool ratios with depth, such as vertical mixing by soil organisms, diffusion, changes in soil texture (SOC protection and stabilization by clay particles), limitations by oxygen and by access to deep organic matter by microbes. There is also a lot of discussion on how sensitive SOC is to these other processes. For example, the study of Huang et al. (in revision for the journal Advances in Modeling Earth Systems) who implemented a matric-based approach to assess the sensitivity of SOC showed that equilibrium SOC stocks are more sensitive to input than to mixing for soils in the temperate and high-latitude regions. For all the above-mentioned reasons and to decrease the uncertainty we made the assumption that the ratios between the SOC stocks of the active, slow and passive pools are equal throughout the soil profile in the new vertical soil discretization scheme and similar to the pool ratios derived from ORCHIDEE.

For the transient period (1850-2005), we made '$re$' remain equal to the equilibrium state values, while '$k_{i0}$' was derived at a daily time-step to keep to SOC stocks of the emulator similar to those of ORCHIDEE and preserve the yearly variability in the soil respiration rates due to changes in soil climate (no soil erosion and no LUC). Details of how we calibrated the exponent 're' and variable '$k_{0i}$' we describe in the section below.

**Method for calibration of '$re$' and '$k_{0i}$'**

We start off by selecting a default value for '$re$' of 2.5 (average between 0 and 5) and then proceed with deriving the values of '$k_{0i}$' according to equations 1-4 described in the following paragraph. After the derivation of '$k_{0i}$' we test if the total SOC stock per grid cell and PFT is

similar to that of ORCHIDEE (difference should be smaller than 1 g m$^{-2}$). If this is not the case we increase or decrease the value of '$re$', but make sure that it stays within the range of 0 and 5. If these is no optimized solution for both '$re$' and '$k0i$', we use the values that produce the smallest difference in SOC stocks between emulator and ORCHIDEE.

After selecting a value for '$re$' for a certain grid cell and PFT we first calculated the respiration rate of the surface soil layer ($k_0$) when all SOC pools are in an equilibrium state, with the following equation:

$$SOC_{orchidee} = \sum_{z=0}^{z=n} \frac{L(z)}{k0*e^{-re*z}} \tag{1}$$

Where, $SOC_{orchidee}$ is the total equilibrium SOC stock derived from ORCHIDEE for a certain grid cell and PFT. $L(z)$ is the total litter input to the soil for a certain soil layer discretized according to the root profile.

Then we derived the equilibrium SOC stocks per soil layer as:

$$SOC(z) = \frac{L(z)}{k0*e^{-re*z}} \tag{2}$$

Assuming that the ratios between the active, slow and passive SOC pools do not change with depth and are equal to the ratios derived from ORCHIDEE, we can calculate the SOC stocks of each pool with the following equation:

$$1 + \frac{soil_s(z)}{soil_a(z)} + \frac{soil_p(z)}{soil_a(z)} = \frac{SOC(z)}{soil_a(z)} \tag{3}$$

Where, $soil_a(z)$, $soil_s(z)$, $soil_p(z)$ are the emulator derived active, slow and passive SOC stock per soil layer, grid cell and PFT. Now, for the equilibrium state the input is equal to the output, so we can derive $k_{0a}$, $k_{0s}$ and $k_{0p}$ from the following equations:

$$\sum_{z=0}^{z=n} \left( \frac{L_a(z)+k_{sa}*soil_s(z)+k_{pa}*soil_p(z)}{k_{0a}*e^{-re*z}+k_{as}+k_{ap}} \right) = SOC_a \tag{4a}$$

$$\sum_{z=0}^{z=n} \left( \frac{L_s(z)+k_{as}*soil_a(z)}{k_{0s}*e^{-re*z}+k_{sa}+k_{sp}} \right) = SOC_s \tag{4b}$$

$$\sum_{z=0}^{z=n} \left( \frac{k_{sp}*soil_s(z)+k_{ap}*soil_a(z)}{k_{0p}*e^{-re*z}+k_{pa}} \right) = SOC_p \tag{4c}$$

Where, *La* is the total litter input to the active SOC pool, *Ls* is the total litter input to the slow SOC pool. $SOC_a$, $SOC_s$, $SOC_p$ are the total active, slow and passive SOC per grid cell and PFT, respectively, derived from ORCHIDEE. $k_{as}$, $k_{ap}$, $k_{sa}$, $k_{sp}$, $k_{pa}$ are the coefficients determining the fluxes between the SOC pools.

In the transient period (no land use change or erosion) we assume a time-constant '*re*' fixed to the equilibrium state. Using the mass-balance approach we can find the daily values for $k_{0a}$, $k_{0s}$, $k_{0p}$ per grid cell and PFT with:

$$\frac{dSOC_a}{dt} = \sum_{z=0}^{z=n}(L_a(z,t) + k_{sa} * soil_s(z,t-1) + k_{pa} * soil_p(z,t-1) - (k_{0a}(t) * e^{-re*z} + k_{as} + k_{ap}) * soil_a(z,t-1)) \tag{8a}$$

$$\frac{dSOC_s}{dt} = \sum_{z=0}^{z=n}(L_s(z,t) + k_{as} * soil_a(z,t-1) - (k_{0s}(t) * e^{-re*z} + k_{sa} + k_{sp}) * soil_s(z,t-1)) \tag{8b}$$

$$\frac{dSOC_p}{dt} = \sum_{z=0}^{z=n}(k_{sp} * soil_s(z,t-1) + k_{ap} * soil_a(z,t-1) - (k_{0p}(t) * e^{-re*z} + k_{pa}) * soil_p(z,t-1)) \tag{8c}$$

In case there was no solution for the '*k0i*' at a certain time-step we took the values from the previous time-step.

**Implications of the new vertical discretization scheme**

After implementing the above-mentioned empirical adjustments to the vertical SOC discretization scheme of the emulator, we found that the resulting SOC stocks for the equilibrium state are close to those of ORCHIDEE with some small deviations (Fig. S1). It was not always possible to precisely match the SOC stocks of the emulator and ORCHIDEE and at the same time have realistic vertical SOC profiles, where the '*re*' variable varies between 0 and 5.

[Figure]

| -0.5 | -0.1 | -0.01 | 0.01 |

Figure S1: The difference in SOC stocks between the emulator and ORCHIDEE as a ratio (%) of the ORCHIDEE SOC stocks for the equilibrium state of the year 1850 without soil erosion. Positive values indicate larger SOC stocks of the emulator compared to the ORCHIDEE model, and negative values indicate smaller SOC stocks of the emulator compared to the ORCHIDEE model.

| PFT number | PFT description | Area-weighted mean re |
|---|---|---|
| 0 | Bare soil | 2.5 |
| 1 | Tropical broad-leaved evergreen | 0.72 |
| 2 | tropical broad-leaved raingreen | 1.33 |
| 3 | temperate needleleaf evergreen | 1.29 |
| 4 | temperate broad-leaved evergreen | 1.25 |
| 5 | Temperate broad-leaved summergreen | 1.09 |
| 6 | boreal needleleaf evergreen | 1.33 |
| 7 | boreal broad-leaved summergreen | 1.13 |
| 8 | boreal needleleaf summergreen | 0.93 |
| 9 | C3 grass | 2.03 |
| 10 | C4 grass | 2.3 |
| 11 | C3 agriculture | 0.73 |
| 12 | C4 agriculture | 0.78 |

Table S1: Global area-weighted average '*re*' values per PFT

Figure S2 shows that also for the transient period of simulation S4 (no erosion or LUC) the total SOC stocks are similar between emulator and the ORCHIDEE model. The difference between the SOC stock of the emulator and ORCHIDEE are between -1 and 0% of ORCHIDEE SOC

stocks for most grid cells, however, the maximum difference can reach -10% for some grid cells. The total global SOC stock of the emulator in the year 2005 deviates by -0.5% from the SOC stock of ORCHIDEE. This differences between the emulator and ORCHIDEE are due to the fact that there was not always an optimal and realistic solution for '*k0i*' and '*re*'.

[Figure]

Figure S2: The difference in SOC stocks between the emulator and ORCHIDEE as a ratio (%) of the ORCHIDEE SOC stocks averaged over the period 1996-2005 for simulation S4 (no erosion, no LUC).The total Positive values indicate larger SOC stocks of the emulator compared to the ORCHIDEE model, and negative values indicate smaller SOC stocks of the emulator compared to the ORCHIDEE model.

Although the new vertical discretization scheme did change the values of the SOC stocks and the related changes to the SOC stocks during the transient period, the main trends and findings of this study remain the same. In the following paragraph we describe the changes to our results, figures and tables as will be presented in the revised manuscript.

**Changes in the manuscript:** We included the above-mentioned derivation of the variables of the new vertical discretization scheme of the emulator and the argumentation behind it, as explained above, in the revised manuscript in chapter 2.3. We include table S1 with the global mean values for the '*re*' exponent per PFT in the supplementary material.

We also changed the tables 2 to 7 of the original manuscript, where we include the new values for changes in SOC stocks and SOC erosion rates. Furthermore, we adapted figures 4B,4D,4F,4H, 5C,5D,5E,6A,7,8A,8B,8C and 9 according to the new results of the simulations with the new vertical discretization scheme. The main changes to the results in the abstract,

chapter 3, 4 and 5 of the manuscript are provided below. The uncertainty ranges provided in the results are related to comment 12 of reviewer 2.

*We make the following changes to the abstract:*
L27: "We found that over the period 1850-2005 AD acceleration of soil erosion leads to a total potential SOC removal flux of 74±18 Pg C of which 79-85% occurs on agricultural, pasture and natural grass lands"
L29: "Including soil erosion in the SOC-dynamics scheme results in an increase of 62% of the cumulative loss of SOC…"
L31:" This additional erosional loss decreases the cumulative global carbon sink on land by 2 Pg for this specific period,…"

*We make the following changes in the results chapter:*
L417: "This global soil loss flux (here 'loss' meaning horizontal removal by erosion) leads to a total SOC loss flux of 0.52±0.14 Pg C y$^{-1}$ of which 26 to 33% are attributed to agricultural land and 54 to 64% to grassland (CTR, Fig 4)"
L421: "The total soil and SOC losses in the year 2005 are an increase of 11-19% and 23-35%, respectively, compared to 1850"
L429: " We found that the total soil erosion flux on agricultural land increased with 55-58% by the year 2005 compared to 1850, while the SOC erosion flux increased with 11-70% (Fig. 4) and led to a cumulative SOC removal of 22±5 Pg."
L431" On pasture land and grassland, the soil erosion flux increased only with 8-20%, while the SOC erosion flux increased with44-54% (Fig. 4) and led to a cumulative SOC mobilization of 38±7 Pg since 1850."
L442: " In total 7183±1662 Pg of soil and 74±18 Pg of SOC is mobilized across all PFTs by erosion during the period 1850 - 2005, which is equal to approximately 46-74% of the total net flux of carbon lost as CO2 to the atmosphere due to LUC…"
L462:" We calculated a total global SOC stock for 2005 in the absence of soil erosion (S3) of 1284 Pg, which is a factor of 0.73 lower than the total SOC stock from GSCE (Shangguan *et al*., 2014) for a soil depth of 2m (Table 5)."

L466:" Including soil erosion (S1) leads to a total SOC stock of 1001±58 Pg for the year 2005 (Table 5). We also find that including soil erosion in the SOC-dynamics scheme slightly improves the root mean square error (RMSE) between the simulated SOC stocks and those from GSDE, for the top 30cm of the soil profile. This improvement in the RMSE occurs especially in highly erosive areas."

L486:" This flux is paralleled by a SOC loss flux of 0.16±0.06 Pg C y-1 after including soil erosion in the CTR simulation (Fig. 4)."

L495:" Furthermore, we find a cumulative soil loss of 1888±753 Pg and cumulative SOC removal flux of 22±5 Pg from agricultural land over the entire time period (CTR simulation). "

*We make the following changes in the discussion chapter:*

L507: "…When considering our best estimated soil erosion rates and assuming that the SOC mobilized by soil erosion in the CTR simulation is all respired, we find an overall global SOC stock decrease that is 62 % larger compared to a world without soil erosion…"

L582:" We find that the global SOC stock decreases by 17 Pg due to LUC only during 1850 – 2005 (Fig. 6A, S3-S4). The overall change in carbon over this period summed up over all biomass, litter, SOC, and wood-product pools due to  LUC without erosion is a loss of 102 Pg C which lies in the range of cumulative carbon emissions by LUC from estimates of previous studies (Houghton and Nassikas, 2017; Li et al., 2017; Piao et al., 2009)"

L585: "When we use our best estimated soil erosion rates in the SOC-dynamics scheme of the emulator we find that the LUC effect on the global SOC stock is amplified by 4Pg or a factor of 1.2 (S1-S2, Fig. 6A)"

L591: "This leads to a total change in the overall carbon stock on land of -106 Pg."

L610: " …and leads even to a net cumulative sink of carbon on land over this period of about 30 Pg C (S3)"

L613:" In the presence of soil erosion, climate variability and the atmospheric $CO_2$ increase lead to a slightly smaller net cumulative sink of carbon over land of 28 Pg C (S1)…"

*We make the following changes to the conclusion chapter:*

L713: "This potential soil loss flux mobilized 74±18 Pg of SOC across all PFTs, which compares to 60% of the total net flux of carbon lost as $CO_2$"

L715:" When assuming that all this SOC mobilized is respired we find that the overall SOC change over the period 1850-2005 would increase by 62% and reduce the land carbon sink by 2 Pg."

Figure 4: There are no significant changes in the historical trends due to the new vertical discretization scheme, except for the overall values.

Figure 5: Only slight changes in the spatial variability of the SOC stocks    and SOC erosion rates are observed, which are due to the new vertical discretization scheme.

Figure 6A: No significant changes in the temporal trends of the cumulative SOC stock are observed when the new vertical discretization scheme is used. However, the overall changes in the cumulative stocks are smaller.

Figure 7: The cumulative SOC stock change in the original manuscript was wrongly projected and is corrected here. The grassland SOC stocks should decrease instead of increase. Overall, the historical trends in the cumulative SOC stocks follow closely the changes in the respective vegetation fractions. The overall changes in SOC stocks are smaller here compared to the figure in the original manuscript due to the new vertical scheme.

Figure 8: no significant changes are observed in the plots due to the new vertical scheme.

Figure 9: Some slight changes are observed in the plot due to the new vertical scheme.

**Comment 2:** *"Land use change map. The LUC is prescribed by PFT fractional change derived from Peng 2017. Wondered how this LUC dataset differs from Land-Use Harmonization (LUH2), the new CMIP6 land use change dataset. Given that LUC is a dominant factor of SOC erosion, I am curious about the uncertainty of SOC erosion, induced by using different LUC estimate (e.g., Peng 2017 vs LUH2)."*

**Answer:** The PFT fractional map is based on LUHv2 land use dataset, historical forest area data from Houghton (for large regions) and present day forest area from ESA CCI satellite land cover data (Peng et al., 2017). The historical forest data from Houghton and the latest satellite land cover data from ESA are the best estimates that currently exist on forest area. Figure S3 shows

that if the forest is not constrained with methods described by Peng et al. (2017), there is a stronger decrease in forest area over the period 1850-2005. Also the grassland shows an increasing trend, while in the PFT map with constrained forest the grassland shows globally a slight decreasing trend. In the rest of the text we will refer to the PFT map constrained with data on forest area as the 'constrained PFT map' and to the other PFT map as the 'unconstrained PFT map'.

We agree with the reviewer that different land use data can result in large uncertainties in both SOC stocks and soil erosion rates. To show the potential uncertainty in our results due to uncertainties in underlying land use data we performed 4 additional simulations (S1 to S4) using the unconstrained PFT map and the new vertical discretization scheme.

Differences in global average soil erosion rates between the different PFT maps are small (Fig. S4), mainly because the C-factor of our Adjusted RUSLE model is similar for forest and dense natural grass. As the change in cropland area globally was not very different between the 2 PFT maps, the overall soil erosion rates were similar. We expect, however, that the changes in soil erosion rates between the 2 PFT maps can be significant in areas where the change in forest area was significant over the historical period.

In contrast to the soil erosion rates, the 2 PFT maps resulted in significant differences in the SOC erosion rates and cumulative changes in SOC stocks during the transient period (Fig. S4). The global SOC stock in the equilibrium state without soil erosion (S3) is 8 % higher when the unconstrained PFT map is used, due to a larger global forest area in this map at 1850. The higher global SOC stock of the PFT map without constrained forest area lead to higher SOC erosion rates (Fig. S4b). According to the unconstrained PFT map, soil erosion leads to a total SOC removal of 79 Pg (S1) over the period 1850-2005, which is 6Pg larger than the total SOC removal by soil erosion under the constrained PFT map.

Interestingly, according to the unconstrained PFT map, the global cumulative SOC stock change over 1850-2005 under soil erosion and LUC (S1) is 60% smaller than the stock derived using the constrained PFT map. This is most likely due to the higher forest area at the start of the period A850-2005, leading to a larger increase in SOC stocks by increasing atmospheric $CO_2$ concentrations. The global LUC effect on the SOC stocks of both PFT maps is found to be similar (Fig 5C, D).

Figure S3: Changes (%) in forest (green), grass (light-green) and crop (red) fractions over the period 1850-2005 with respect to the year 1850AD. The dashed lines represent data from the PFT map without constrained forest, while the straight lines represent data from the PFT map from Peng et al. (2017).

[Figure]

Figure S4: Differences in (A) soil and (B) SOC erosion rates and (C and D) cumulative SOC stock changes between the PFT map that is constrained by forest area data (straight lines) and the PFT map that is not constrained by forest area data (dashed lines).

**Changes in the manuscript:**

The uncertainty due to different land use maps, as discussed above, will be discussed in chapter 4.4 of the manuscript, and we will mention the above-mentioned changes to the erosion rates and SOC stocks. The new figures S3 and S4 will be included in the supplementary material.

**Specific comment 1:** "L16: The first sentence gives me an incorrect hint that the paper is going to talk about agriculture activity accelerates soil erosion"

**Answer:** We agree that this may be misleading and changed this sentence to: "Erosion is an Earth System process that transports carbon laterally across the land surface, and is currently accelerated by anthropogenic activities. Anthropogenic land cover change has accelerated soil erosion rates by rainfall and runoff substantially, mobilizing vast quantities of soil organic carbon (SOC) globally."

**Specific comment 2:** "L43: 1.0 Pg, does it include fire emission?"

**Answer:** yes it does

**Specific comment 3:** "L61 what is bookkeeping models?"

**Answer:** Bookkeeping models are methods/tools to calculate land use change emissions by keeping track of the carbon stored in different pools before and after land use changes take place. These methods also keep track of the CO2 emitted after land use change has taken place.

**Specific comment 5:** "L337 What's the meaning of randomly projected? A more reasonable way is to repeat 1990-1910 climates during 1850-1900."

**Answer:** "Randomly projected", means that the climate of the years after 1900 was randomly assigned to the years between 1850 and 1900 because the climate data of CRU-NCEP was only available starting from year 1900. If we would choose to repeat the climates of 1900-1910, we would risk including the effects of extreme climate conditions multiple times.

**Changes in the manuscript:**

In L337 we will add: "The random projection of the climate data is necessary to avoid the risk of including the effects of extreme climate conditions multiple times when only a certain decade is used repeatedly."

**Specific comment 6: "**L556 "Also, intense soil erosion is typically found in mountainous areas where climate variability has significant impacts, while at the same time these regions are usually poor in SOC." It's not clear in the manuscript whether or not ORCHIDEE has topography information? In another word, if ORCHIDEE simulates a low SOC stock over the grid cells that have mountains, is that because of the topographical feature of this gridcell can not hold a lot of SOC in ORCHIDEE? Or because of other reasons such as climate constraints (e.g., colder in mountain area)?"

**Answer:** ORCHIDEE has no soil depth information and thus cannot simulate low SOC stocks due to the fact that the gridcell cannot hold a lot of SOC. Low SOC might however be a result of the plant productivity, the climate (temperature and precipitation), soil moisture and clay content (which is a constant variable). ORCHIDEE has, however, topographical information such as slope that determines the flow directions for water/runoff and affects hydrological parameters such as soil moisture content.

**Changes in the manuscript:** L556: "Also, intense soil erosion is typically found in mountainous areas where climate variability has significant impacts, while at the same time these regions are usually poor in SOC due to unfavorable environmental conditions for plant productivity."

**Specific comment 7: "**L577 $CO_2$ fertilization effects on NPP is not fully convincing here, because ORCHIDEE does not have nutrient constraints. OCN might be a better surrogate model to be able to say something about $CO_2$ fertilization effect on NPP."

**Answer:** In the ORCHIDEE model version we used the nutrients are indeed absent. Our intention, however, was to show the complete picture of possible direct and indirect interactions of soil erosion with the C cycle with the current model setup. The representation of nutrients in global land surface models is new and the related uncertainties are not well quantified. We work with a more or less simple version of ORCHIDEE and the C emulator to be able to understand and quantify the effects of soil erosion on the C cycle.

**Changes in the manuscript:**

L636: "Finally, it should be mentioned here that the absence of nutrients in the current version of the ORCHIDEE model may result in an overestimation of the CO2 fertilization effect on NPP and may introduce biases in the effect of erosion on SOC stocks under increasing atmospheric CO2 concentrations. Soil erosion may also lead to significant losses of nutrients, especially in agricultural areas. For a more complete quantification of the effects of soil erosion on the carbon cycle, nutrients have to be included in future studies."

**Specific comment 8: "** Figure 4. I do not fully understand why climate change either decrease or not change erosion?

**Answer:** With climate change we mean temperature and precipitation changes. For soil erosion only precipitation changes are of interest. Globally we find that average yearly precipitation shows a slightly decreasing trend over the period 1950 – 2005 according to the ISIMIP2b dataset used to calculate soil erosion rates. A global smaller total precipitation with respect to 1850 AD will lead to smaller soil erosion rates when LUC is not included. The decrease in total precipitation over land is mostly coming from the tropics, where due to large precipitation amounts a change in precipitation can alter soil erosion significantly. At the same time precipitation is very variable and might not lead to a significant global net change in soil erosion rates over the total period 1850-2005. This result might be contradictory to the fact that major soil erosion events are caused by storms. But in our case we model only rill and interril erosion, which is usually a slow process and previous studies have shown that land use change is usually the main driver of behind accelerated rates of this type of soil erosion. Furthermore, there are very few studies that have quantified the individual effects of precipitation change versus land use change on soil erosion rates over a sufficiently long time period. Therefore, it is difficult to verify this result. However, our soil erosion model performs well for present-day and therefore any possible biases here could be mainly related to biases in precipitation rates, and soil parameters. We agree that this is an interesting point raised by the reviewer and added some additional sentences explaining the trend in the beginning of chapter 4.2.

**Changes in the manuscript:**

L536: "The global decrease in precipitation in many regions worldwide, especially in the Amazon, as simulated by ISIMIP2b, lead to a slight decrease in soil and SOC erosion rates (Fig.

4). At the same time precipitation is very variable and might not lead to a significant global net change in soil erosion rates over the total period 1850-2005. This result might be contradictory to the fact that major soil erosion events are caused by storms. But in this study we only simulate rill and interril erosion, which are usually slow processes. In addition, previous studies have shown that land use change is usually the main driver behind accelerated rates of these types of soil erosion. Our study confirms this observation."

**Author's Response to Reviewer 2**

We would like to thank the anonymous reviewer for his or her constructive comments. In this response we provide an answer to all the comments and the indicated changes will be applied in the revised manuscript.

**Comment 1:** *"The emulator used in this study seems to have various limitations that make the numbers presented quite uncertain – further discussion on, and quantification of, these uncertainties is warranted and would greatly improve this manuscript. Specifically, I would have liked to see additional support for the SOC model formulation, parameters, and built-in feedbacks chosen for the emulator, as well as support for its vertical discretization and parameterization.*

The carbon emulator is supposed to describe the carbon pools and fluxes exactly as in ORCHIDEE, yet the total global SOC stocks from the emulator are 44% higher than that of the original ORCHIDEE model. This is a big difference. What does this tell us about the accuracy and applicability of the emulator, and how do the SOC stocks of the two models compare to the Harmonized World Soil Database (HWSD) and other global SOC databases? Additional major comments/questions, especially those regarding the assumptions and methods used, are detailed below."

**Answer:** We modified the vertical discretization scheme of the emulator in such a way that the total SOC stock of each grid cell, PFT and C pool is close to that of ORCHIDEE when soil erosion and land use change is deactivated (0.5% max difference in total global SOC stock). For this we calibrated both the exponent *'re'* and variable *'$k_{0i}$'* of equation 8 in the manuscript for each grid cell and PFT under equilibrium conditions, such that the total soil respiration per grid cell, PFT, and soil C pool of the emulator would be similar to that of the ORCHIDEE model. For the transient period (1850-2005), we made *'re'* remain equal to the equilibrium state values,

while values for '$k_{i0}$' were derived at a daily time-step to keep to SOC stocks of the emulator similar to those of ORCHIDEE and preserve the yearly variability in the soil respiration rates due to changes in soil climate (soil erosion and land use change were deactivated ). Details of how we calibrated the exponent '$re$' and variable '$k_{0i}$' we describe in our response to Reviewer 1 and in the manuscript after line 274. The modified vertical discretization scheme did change the values of the SOC stocks and SOC removal rates, because with this scheme we simulated total SOC such as in the ORCHIDEE model without soil erosion and land use change. However, the overall trends in soil and SOC erosion rates and cumulative changes in SOC stocks during the transient period did not change significantly and the main findings of our study remain unchanged. For more details on the changes in our manuscript related to the modified vertical discretization scheme see our detailed response to reviewer 1.

We performed a comparison of our simulated SOC stocks to the Global Soil Database for Earth System Modeling (GSDE), as is described in paragraph 3.2 and the new table 5 of the manuscript (with results based on the new vertical discretization scheme). However, we abstained from a more in-depth comparison as our emulator and ORCHIDEE do not include various soil processes that have been proven to affect SOC substantially such as vertical mixing, diffusion, priming, changes in soil texture, C rich organic soils formation, etc.

It should be noted that there are also large uncertainties in the global soil databases (Hengl et al., 2014; Scharlemann et al., 2014; Tifafi et al., 2018), which makes the exact quantification of the uncertainties of the resulting SOC dynamics simulated by our emulator difficult.

After applying the modifications to the vertical SOC discretization scheme, we performed a simulation with soil erosion and land use change activated (S1) and compared the resulting SOC stocks with those from the GSDE. Figure S1 shows that our emulator, and the ORCHIDEE model in general, underestimates SOC stocks globally, except for the high-latitudes.

[Figure]

Figure S1: Difference between SOC stocks of the emulator (simulation S1) and the SOC stocks of GSDE as a percentage of the SOC stocks of GSDE till 2m depth. Red colors show larger SOC stocks by the emulator, while blue colors indicate smaller SOC stocks by the emulator compared to GSDE.

**Changes in manuscript:** We put figure S1 in the supplementary material. Furthermore, the values of table 5 of the manuscript are updated and presented in the revised manuscript.

L493:" We did not perform a more in-depth comparison as our emulator and ORCHIDEE do not include various soil processes that have been proven to affect SOC substantially such as vertical mixing, diffusion, priming, changes in soil texture, carbon rich organic soils formation, etc. The ORCHIDEE model we use also lacks processes such as nitrogen and phosphorus limitations and priming, which affect the productivity and SOC decomposition (Goll *et al.*, 2017; Guenet *et al.*, 2016). The emulator also misses the SOC transport and deposition after erosion, and there is a general uncertainty in the simulation of underlying processes that govern the SOC dynamics (Todd-Brown *et al.*, 2014). Finally, large uncertainties in the global soil databases (Hengl et al., 2014; Scharlemann et al., 2014; Tifafi et al., 2018), complicate the exact quantification of the uncertainties of the resulting SOC dynamics simulated by our emulator."

**Specific comment 1: "**L143: What are the limitations of not including these processes in the emulator? Can it capture all feedbacks and dynamics?"

**Answer:** Some of these processes are already included in the ORCHIDEE model, which is the basis for the C emulator but other feedbacks on SOC are missing in the original ORCHIDEE model such as the effect of SOC on the hydrology or on the thermic of the model. Nevertheless, our main objective here was to present a tool able to evaluate erosion fluxes at global scale using a 'state-of-art' land surface model outputs and estimate the drivers of erosion at global scale. In addition, this study did not focus on the feedbacks of soil erosion and land use change on NPP, the hydrological cycle or nutrient cycle and therefore it was decided not to incorporate soil erosion processes directly into ORCHIDEE, but rather use the C emulator concept instead. Not including these processes explicitly in the emulator does not change the simulated SOC dynamics in our study.  However, the emulator has a flexible structure and could be made more complex depending on the needs, such as including a more sophisticated vertical discretization scheme.

The main idea of the emulator was to use a modeling tool that does not require much computational power but that still incorporates the basic processes and variables for simulating large-scale SOC dynamics under soil erosion and land use change. Many simulations were needed to quantify the various effects of soil erosion on the C cycle and to calibrate the model parameters. The C emulator was in this case a convenient tool, as it is fast and its structure allows to easy switch processes on or off.

**Changes in manuscript:** See our answer to the next comment.

Section 2.1 L158: "Our main objective here is to present a tool able to evaluate erosion fluxes at global scale using a 'state-of-art' land surface model outputs and estimate the drivers of erosion at global scale."

**Specific comment 2:** "L142-143: although originally calculated by complex equations, the dynamic evolution of each pool can be described using the first-order model" – why were the complex equations needed initially then? Again, what are the limitations of this first-order model?

**Answer:** The limitations of this first-order model are the incapability to capture feedbacks on the hydrological processes or on the NPP (see answer to previous comment). However, because the SOC is represented by first order equations inside ORCHIDEE and the complex equations only

compute the modifier to the default coefficients, and as our study focused on the effects of soil erosion on the SOC dynamics, we decided to use the C emulator, assuming that erosion will not significantly impact soil physics (and in turn decomposition) affecting SOC. The complex equations, such as photosynthesis and hydrological processes are needed to simulate realistically the changes in biomass, litter and soil respiration over time, which is done by ORCHIDEE. In the original ORCHIDEE simulations, these processes are explicitly simulated on a 30 min time step. Such a time step is needed for coupled simulations with a climate model, but makes the model CPU intensive, and there is no need for such high-resolution calculations of 'fast' C fluxes for erosion induced effects on SOC. In the emulator, all C fluxes between ecosystem compartments (with and without erosion) are exactly the same as the original ORCHIDEE, assuming that there is no feedbacks between erosion and these fluxes. The C emulator is much more computational efficient than the original ORCHIDEE because it does not require to compute all 'fast' processes for all simulations. The emulator thus allows us to conduct a lot of simulations (e.g. with and without climate change, with and without $CO_2$ fertilization, with and without land use change, with and without erosion), and at the same time keep the main features (except erosion) of the original ORCHIDEE simulation.

**Changes in manuscript:** L144:"…Eq.1. Complex equations, such as photosynthesis and hydrological processes are needed to simulate realistically the carbohydrates input to carbon pools and the moisture and temperature conditions controlling litter and soil carbon decomposition over time. All the processes that determine surface and soil temperature and soil moisture, are calculated by the ORCHIDEE LSM on a 30 minute time-step. Such a time-step is needed for coupled simulations with a climate model, but makes the LSM model CPU intensive. However, there is no need for such high temporal resolution calculations of 'fast' carbon and energy fluxes to account for erosion-induced effects on SOC stocks. The addition of erosion is here supposed to impact only carbon pools, and to have no feedbacks on soil moisture, soil temperature and photosynthesis. Therefore, we decided to use the emulator concept rather than incorporating erosion processes directly into ORCHIDEE. For each carbon pool…"

**Specific comment 3:** "L196: What does the passive pool correspond to (as a measureable pool)? Why is there no transfer from p to s (k_ps)? Why no input to this pool?"

**Answer:** The distribution of SOC into an active, slow and passive pool and the transfer rates between these pools are based on the work of Parton et al. (1988). These pools are defined by their different residence times. The active, slow and passive SOC pools have a residence time of 1.5, 25 and 1000 years, respectively. That study defines the passive pool as a pool that is very resistant to decomposition and includes physically and chemically stabilized SOM. The proportions of the decomposition products which enter the passive pool from the slow and active pools increase with increasing soil clay content. Passive C is thus not directly produced from litter input but active or slow C has to be stabilized first to become passive C. Then, the original model of Parton et al., (1987) assumes that when the passive pool is decomposed by microorganisms, they produce metabolites corresponding to more labile materials that are released in the soil solution during microbial death and the associated cell lysis. For these reasons, they considered that the decomposed passive pool can only by recycled into the active pool.

**Changes in manuscript:** L196: "The SOC pools are based on the study of  Parton et al. (1988) and are defined by their residence times. The active SOC pool has the lowest residence time (~1.5 years) and the passive the highest (1000 years)."

**Specific comment 4:** "L209: Does this allow for emergent differences in the relative distribution of the three pools with depth? (e.g., relatively more passive C than active C with depth, etc.)"

**Answer:** Yes, the old vertical discretization scheme allowed for different relative distributions of the three pools with depth. However, we changed this aspect by assuming that the ratios between the three pools do not change with depth so that the relative distribution is the same. We made this assumption as we have not enough data to clearly determine how the ratios between the pools change should change with depth (see reply to rev #1). In addition, we do not simulate the

underlying processes that would allow for changing ratios between the SOC pools such as changing clay content with depth, diffusion, bioturbation.

**Specific comment 5:** "L196 (old manuscript): "The SOC respiration rates for the topsoil layers are equal to those from ORCHIDEE". But how about subsoil respiration? Does the emulator have more respiration overall then? Please clarify how the models compare."

**Answer:** We modified the vertical discretization scheme, so that the emulator now has a similar SOC respiration rate as ORCHIDEE without soil erosion or land use change. See our response to the first comment and to the comments of reviewer 1.

**Specific comment 6:** "L256: "total global SOC stock is approximately 44% larger than that from the original ORCHIDEE model" – what does this tell us about the accuracy and applicability of the emulator? This seems to be a big difference. How do the SOC stocks of the two compare to the HWSD and other global SOC databases?"

**Answer:** We modified the vertical discretization scheme, where the emulator has similar SOC stocks as ORCHIDEE without soil erosion or land use change. For more details see our response to the first comment of reviewer 1.

**Specific comment 7: "**L226: How are these fractions determined? What are the implications of the uncertainty in this partitioning?"

**Answer:** Above and below-ground litter consists out of plant residues and organic animal excreta that are partitioned into structural and metabolic pools as a function of the lignin to N ratio in the residue (Parton et al., 1988). The lignin and N ratios are usually prescribed per PFT and derived from plant-trait databases. This partitioning is prescribed by Parton et al. (1988) and followed by Krinner et al. (2005). The structural litter pool has a slower decay rate and contains the more recalcitrant molecules, while the metabolic pool has a faster decay rate and contains labile plant material. The decay rates are a function of temperature and humidity (Krinner et al., 2005). The lignin fraction of the plant material does not go through the active pool but is

assumed to go directly to the slow C pool as the structural plant material decomposes. This is why part of the decomposed structural litter pool goes the active SOC pool and another to the slow SOC pool. Metabolic litter can be decomposed into active SOC and could also form a mineral-stabilized SOC (slow SOC pool, Cotrufo et al., 2015). The CENTURY model simulates the dynamics of C and nutrients (Parton et al., 1988), and is widely applied and tested in Land Surface Models. There are definitely large uncertainties in the partitioning of the litter pools, however, it is not in the scope of this paper to discuss these uncertainties.

**Changes in manuscript:** L230: "These litter fractions are based on the Century model as introduced by Parton et al.(1988) and later implemented inside ORCHIDEE (Krinner et al., 2005)."

**Specific comment 8:** "L337: Why "randomly projected"? Please explain how and why."

**Answer:** "Randomly projected", means that the climate of the years after 1900 was randomly assigned to the years between 1850 and 1900 because the climate data of CRU-NCEP was only available starting from year 1900. If we would choose to repeat for example the climates of 1900-1910, we would risk including the effects of extreme climate conditions multiple times, which is not the case when a random projection is used.

**Specific comment 9:** "L359: But you used CRU-NCEP for ORCHIDEE... what are the caveats of using different climate datasets for each model?"

**Answer:** We compared the historical trend in yearly total precipitation between CRU-NCEP and ISIMIP2b, see figure S2. We find that although the ISIMIP2b shows a higher overall precipitation amount, the temporal trend and variability are similar to that of CRU-NCEP. If we would use CRU-NCEP to calculate the soil erosion rates, we expect that the new soil erosion rates would fall inside the uncertainty range created by calculation of the R- and the C-factors of the Adj. RUSLE (see answer to the last comment).

[Figure]

Figure S2: Temporal trend in yearly total, global average precipitation derived from ISIMIP2b ( straight line) and CRU-NCEP (dashed line).

**Specific comment 10: "**L366: Why this dataset? How does it compare to the HWSD and SoilGrids (Hengl et al. PLoS ONE 2014, 2017) datasets?"

**Answer:** The GSDE is based on the SoilMap of the World (FAO, 1995, 2003) and various regional and national soil databases. It is available at a 1km resolution and at 5 arcmin resolution and contains updated soil information and more soil variables such as nutrients. The GSDE is based on several regional soil maps and is more up to date on soil information than the HWSD but both products compared relatively well since they shared several data (Shangguan et al., 2014). We did not test the SoilGrids data, which is based on a different approach. A recent publication showed that SoilGrids give different results compared to HWSD (Tifafi et al., 2018) but regarding the difference between the products we decided to use only one of them already used to evaluate erosion process and then be more comparable with previous publications (Naipal et al., 2015, 2016).

**Specific comment 11:** L415: How uncertain are these numbers given the model formulation assumptions, land-use maps, and methods used? It would help to see a sensitivity analysis and some uncertainty ranges.

**Answer:** We performed 4 additional simulations with a different PFT map, which is also based on the LUH2 land use dataset but where the historical forest area change that is not constrained by data as done by Peng et al. (2017). We used these simulations to show the differences to our results when other land use maps are used. If the forest is not constrained with methods described by Peng et al. (2017), there is a stronger decrease in forest area over the period 1850-2005. Also the grassland shows an increasing trend, while in the PFT map with constrained forest the grassland shows globally a slight decreasing trend. In the rest of the text we will refer to the PFT map constrained with data on forest area as the 'constrained PFT map' and to the other PFT map as the 'unconstrained PFT map'. Differences in global average soil erosion rates between the different PFT maps are small, however, there are significant differences in the SOC erosion rates and cumulative changes in SOC stocks during the transient period. According to the unconstrained PFT map, soil erosion leads to a total SOC removal of 79 Pg (simulation S1 with the new vertical discretization scheme) over the period 1850-2005, which is 6Pg larger than the total SOC removal by soil erosion under the constrained PFT map.

Interestingly, according to the unconstrained PFT map, the global cumulative SOC stock change over 1850-2005 under soil erosion and LUC (S1) is 60% smaller than the stock derived using the constrained PFT map. This is most likely due to the higher forest area at the start of the period A850-2005, leading to a larger increase in SOC stocks by increasing atmospheric $CO_2$ concentrations. The global LUC effect on the SOC stocks of both PFT maps is found to be similar. For more details and our changes in the manuscript see our answer to comment 2 of reviewer 1.

**Specific comment 12: "**L645: (Section 4.4) with all of these model limitations, it would be nice to have a rough quantification of uncertainties."

**Answer:** We agree with the reviewer that quantifying the uncertainty is important. Therefore, we derived an uncertainty range for our soil erosion rates. First, we varied the R-factor of the Adj.RUSLE model between a maximum and a minimum based on the regression equations derived by Naipal et al. (2015) per climate zone. Then we varied the C-factor of the Adj.RUSLE model between a maximum and minimum value per land cover type (tree, crop or grass) based

on literature. We then used the uncertainty range in the C and R factors to derive the uncertainty range in the soil erosion rates and subsequently in the SOC erosion rates. We performed 4 additional simulations with the emulator, 2 simulations with the setup of S1 and a minimum and maximum soil erosion scenario, and 2 simulations with the setup of S2 with a maximum and minimum soil erosion scenario. The results can be found in sections 3 and 4.

**Changes in manuscript:** We present the resulting soil and SOC erosion range with an uncertainty estimate that is related to the variation in the C and R factors of the Adj. RUSLE model. Furthermore, we discuss the effect of soil erosion uncertainty on the land carbon sink in section 4.4.

**Global soil organic carbon removal by water erosion under climate change and land use change during 1850-2005 AD**

Victoria Naipal[1], Philippe Ciais[1], Yilong Wang[1], Ronny Lauerwald[2, 3], Bertrand Guenet[1], Kristof Van Oost[4]

[1]Laboratoire des Sciences du Climat et de l'Environnement, CEA CNRS UVSQ, Gif-sur-Yvette 91191, France

[2]Department of Geoscience, Environment & Society, Université Libre de Bruxelles, Brussels, Belgium

[3]Department of Mathematics, College of Engineering, Mathematics and Physical Sciences, University of Exeter, Exeter, UK

[4]Université catholique de Louvain,TECLIM - Georges Lemaître Centre for Earth and Climate Research, Louvain-la-Neuve, Belgium

*Correspondence to*: Victoria Naipal (victoria.naipal@lsce.ipsl.fr)

**Abstract**

Erosion is an Earth System process that transports carbon laterally across the land surface, and is currently accelerated by anthropogenic activities.  Anthropogenic land cover change has accelerated soil erosion rates by rainfall and runoff substantially, mobilizing vast quantities of soil organic carbon (SOC) globally. At timescales of decennia to millennia this mobilized SOC can significantly alter previously estimated carbon emissions from land use change (LUC). However, a full understanding of the impact of erosion on land-atmosphere carbon exchange is still missing. The aim of  this study is to better constrain the terrestrial carbon fluxes by developing methods compatible with  Land Surface Models (LSMs) in order to explicitly represent the links between soil erosion by rainfall and runoff and carbon dynamics. For this we use an emulator that represents the carbon cycle of a  LSM, in combination with the Revised Universal Soil Loss Equation model. We applied this modeling framework at the global scale to evaluate the effects of potential soil erosion (soil removal only) in the presence of other perturbations of the carbon cycle: elevated atmospheric $CO_2$, climate variability, and LUC. We found that over the period 1850-2005 AD acceleration of soil erosion leads to a total potential SOC removal flux of  74±18 Pg C of which 79-85% occurs on agricultural  and  grass-land. Using our best estimates for soil erosion we find that Including soil erosion in the SOC-dynamics scheme  results in an increase of 62% of the cumulative loss of SOC over 1850 – 2005 due to the combined effects of climate variability, increasing atmospheric $CO_2$ and LUC. This additional erosional loss decreases the cumulative global carbon sink on land by 2 Pg of carbon for this specific period, with the largest effects found for the tropics, where deforestation

**Comment [VN1]:**
**Reviewer #1:** "L16: The first sentence gives me an incorrect hint that the paper is going to talk about agriculture activity accelerates soil erosion"

**Answer:** We agree that this may be misleading and changed this sentence.

[revised manuscript text omitted]

**Comment [VN2]:**
**Reviewer #2:** "What are the limitations of not including these processes in the emulator? Can it capture all feedbacks and dynamics?"

**Answer:** Some of these processes are already included in the ORCHIDEE model, which is the basis for the C emulator but other feedbacks on SOC are missing in the original ORCHIDEE model such as the effect of SOC on the hydrology or on the thermic of the model. Nevertheless, our main objective here was to present a tool able to evaluate erosion fluxes at global scale using a 'state-of-art' land surface model outputs and estimate the drivers of erosion at global scale. In addition, this study did not focus on the feedbacks of soil erosion and land use change on NPP, the hydrological cycle or nutrient cycle and therefore it was decided not to incorporate soil erosion processes directly into ORCHIDEE, but rather use the C emulator concept instead. Not including these processes explicitly in the emulator does not change the simulated SOC dynamics in our study. However, the emulator has a flexible structure and could be made more complex depending on the needs, such as including a more sophisticated vertical discretization scheme.
The main idea of the emulator was to use a modeling tool that does not require much computational power but that still incorporates the basic processes and variables for simulating large-scale SOC dynamics under soil erosion and land use change. Many simulations were needed to quantify the various effects of soil erosion on the C cycle and to calibrate the model parameters. The C emulator was in this …

**Comment [VN3]:**
**Reviewer #2:** " although originally calculated by complex equations, the dynamic evolution of each pool can be described using the first-order model" – why were the complex equations needed initially then? Again, what are the limitations of this first-order model?

**Answer:** The limitations of this first-order model are the incapability to capture feedbacks on the hydrological processes or on the NPP (see answer to previous comment). However, because the SOC is represented by first order equations inside ORCHIDEE and the complex equations only compute the modifier to the default coefficients, and as our study focused on the effects of soil erosion on the SOC dynamics, we decided to use the C emulator, assuming that erosion will not significantly impact soil physics (and in turn decomposition) affecting SOC. The complex equations, such as photosynthesis and hydrological processes are needed to simulate realistically the changes in biomass, litter and soil respiration over time, which is done by ORCHIDEE. In the original ORCHIDEE simulations, these processes are explicitly simulated on a 30 min time step. Such a time step is needed for coupled simulations with a climate model, but makes the model CPU intensive, and there is no need for such high-resolution calculations of 'fast' C fluxes for erosion induced effects on SOC. In the emulator, all C fluxes between ecosystem compartments (with and without erosion) are exactly the same as the original ORCHIDEE, assuming that there is no feedbacks between erosion and these fluxes. The …

[revised manuscript text omitted]

**Comment [VN4]:**
**Reviewer #2**: "What does the passive pool correspond to (as a measureable pool)? Why is there no transfer from p to s (k_ps)? Why no input to this pool?"

**Answer:** The distribution of SOC into an active, slow and passive pool and the transfer rates between these pools are based on the work of Parton et al. (1988). These pools are defined by their different residence times. The active, slow and passive SOC pools have a residence time of 1.5, 25 and 1000 years, respectively. That study defines the passive pool as a pool that is very resistant to decomposition and includes physically and chemically stabilized SOM. The proportions of the decomposition products which enter the passive pool from the slow and active pools increase with increasing soil clay content. Passive C is thus not directly produced from litter input but active or slow C has to be stabilized first to become passive C. Then, the original model of Parton et al., (1988) assumes that when the passive pool is decomposed by microorganisms, they produce metabolites corresponding to more labile materials that are released in the soil solution during microbial death and the associated cell lysis. For these reasons, they considered that the decompose ...

**Comment [VN5]:**
**Reviewer #1**: "I was not fully convinced by the vertical discretization approach that the emulator used. First of all, different soil layers have totally different biogeophysical and biogeochemical features. Different layers are experiencing different amount of fresh carbon input (e.g., from fine roots exudates, fine root litter), different microbial community (e.g.,fungi/bacteria with different carbon use efficiency), and have different soil structure (e.g., microagregate, macroagregate).
Secondly, even the idea of summarizing the above-mentioned vertical difference into one single factor (re) is believable, the value of re should be carefully inferred for this model, rather than taking from other studies.
Thirdly, and most importantly, the vertical discretization, artificially, increase total global SOC stock by 44%. This type of artifact should be removed. My suggestion is that, since ORCHIDEE has one single soil layer, k0 of ORCHIDEE is supposed to represent the mean turnover rate of th ...

**Comment [VN6]:**
**Reviewer #2:** "Does this allow for emergent differences in the relative distribution of the three pools with depth? (e.g., relatively more passive C than active C with depth, etc.)"

**Answer:** Yes, the old vertical discretization scheme allowed for different relative distributions of the three pools with depth. However, we changed this aspect by assuming that the ratios between the three pools do not change with depth so that the relative distribution is the same. We made this assumption as we have not enough data to clearly determine how the ratios between the pools change should change with depth (see reply to rev #1). In addition, we do not simulate the underlying processes that would allow for changing ratios between the SOC pools such as changing clay content with depth, diffusion, bioturbation.

Also, we do not include processes such as bioturbation or leaching of litter or SOC.

In the ~vertical discretization scheme of the emulatoremulator, the soil profile is divided into thin layers of 1 cm thickness down to a depth of 2 m, which is the soil depth used by ORCHIDEE to calculate SOC. The first 10 cm of the soil profile are referred to as the "topsoil", where we assume that the SOC content is homogeneously distributed. The rest of the soil profile is referred to as the subsoil. The topsoil receives carbon from above- and below-ground litter, and each of the soil layers in the topsoil receives an equal fraction of both litter types.

The below-ground litter input for the active SOC pool is the sum of a fraction of the below-ground structural and metabolic litter pools fromof ORCHIDEE being re-calculated by the emulator, while the below-ground litter input for slow SOC pool is equal to a fraction of the below-ground structural litter pool only. This setting is consistent with the structure of the SOC module of the ORCHIDEE LSM to ensure that the emulator reproduces the same C pool dynamics than the LSM. The litter fractions are based on the Century model as introduced by Parton *et al.* (1987) and later implemented inside ORCHIDEE (Krinner *et al.*, 2005). We assume that the subsoil receives carbon only from below-ground litter, and that this input decreases exponentially with depth following the root profile of ORCHIDEE. This discretization of the total below-ground litter input ($lit_{be}$) is the same for both SOC pools and can then be represented as:

$$lit_{be} = \int_{z=0}^{z=z_{max}} I_{0be} * e^{-r*z} \, dz \tag{5}$$

where $I_{0be}$ is the below-ground litter input to the surface layer, and is equal to:

$$I_{0be} = lit_{be} * \frac{r}{1 - e^{-r*z_{max}}} \tag{6}$$

The homogeneously distributed below-ground litter input ($I_{be}$) to the layers of the topsoil is equal to:

$$\frac{\sum_{z=0}^{z=10} I_{be0} * e^{-r*z}}{0.1} \tag{7a}$$

The below-ground litter input to the layers of the subsoil is equal to:

$$I_{be}(z) = I_{be0} * e^{-r*z} \tag{7b}$$

where $z_{max}$ is the maximum soil depth equal to 2 m, and $dz$ is the soil layer discretization (1 cm); $r$ is the PFT-specific vertical root-density attenuation coefficient as used in ORCHIDEE.

The SOC respiration rates for the topsoil layers are equal to those from ORCHIDEE and are determined by average soil temperature, moisture and texture. For the rest of the soil profile the respiration rates of all three SOC pools decrease exponentially with depth:

$$k_i(z) = k_{0\,i} * e^{-r_e z} \tag{85}$$

Here $k_{0\,i}$ is the SOC respiration rate at the surface layer for each SOC pool (i = a, s, p). and as derived by the emulator based on the original ORCHIDEE rates. $r_e$ (m$^{-1}$) is an exponential decreasing coefficient representing the impact of external factors, such as oxygen availability, which reduceing SOC mineralization rate with depth ($z$). After performing sensitivity simulations where the values of $r_e$ differ per PFT, we found no significant difference in the resulting SOC stocks. The value of $r_e$ was then set to 1.2 m$^{-1}$, so that the respiration rate decreases with a factor of 2-3 from the surface to 1 m depth consistent with SOC profile observations (Bouchoms *et al.*, 2017; Van Oost *et al.*, 2005; Wang *et al.*, 2015a). This vertical discretization in the emulator leads to a total global SOC stock that is

**Comment [VN7]:**
**Reviewer #2:** "How are these fractions determined? What are the implications of the uncertainty in this partitioning?""

**Answer:** Above and below-ground litter consists out of plant residues and organic animal excreta that are partitioned into structural and metabolic pools as a function of the lignin to N ratio in the residue (Parton et al., 1988). The lignin and N ratios are usually prescribed per PFT and derived from plant-trait databases. This partitioning is prescribed by Parton et al. (1988) and followed by Krinner et al. (2005). The structural litter pool has a slower decay rate and contains the more recalcitrant molecules, while the metabolic pool has a faster decay rate and contains labile plant material. The decay rates are a function of temperature and humidity (Krinner et al., 2005). The lignin fraction of the plant material does not go through the active pool but is assumed to go directly to the slow C pool as the structural plant material decomposes. This is why part of the decomposed structural litter pool goes the active SOC pool and another to the slow SOC pool. Metabolic litter can be decomposed into active SOC and could also form a mineral-stabilized SOC (slow SOC pool, Cotrufo et al., 2015). The CENTURY model simulates the dynamics of C and nutrients (Parton et al., 1988), and is widely applied and tested in Land Surface Models. There are definitely large uncertainties in the partitioning of the litter pools, however, it is not in the scope of this paper to discuss these uncertainties.

[revised manuscript text omitted]

**Comment [VN8]:**
**Reviewer #1:** "What's the meaning of randomly projected? A more reasonable way is to repeat 1990-1910 climates during 1850-1900."

**Answer:** "Randomly projected", means that the climate of the years after 1900 was randomly assigned to the years between 1850 and 1900 because the climate data of CRU-NCEP was only available starting from year 1900. If we would choose to repeat the climates of 1900-1910, we would risk including the effects of extreme climate conditions multiple times.

**Comment [VN9]:**
**Reviewer #2:** "But you used CRU-NCEP for ORCHIDEE... what are the caveats of using different climate datasets for each model?"

**Answer:** We compared the historical trend in yearly total precipitation between CRU-NCEP and ISIMIP2b. We find that although the ISIMIP2b shows a higher overall precipitation amount, the temporal trend and variability are similar to that of CRU-NCEP. If we would use CRU-NCEP to calculate the soil erosion rates, we expect that the new soil erosion rates would fall inside the uncertainty range created by calculation of the R- and the C-factors of the Adj. RUSLE

interpolation method to the resolution of the Adj.RUSLE model, before being used to calculate the R-factor. This was necessary because the erosivity equations from the Adj.RUSLE model are calibrated at this specific resolution (Naipal *et al.*, 2015).

364

Data on soil bulk density and other soil parameters to calculate the soil erodibility factor (K), available at the resolution of 1 km, have been taken from the Global Soil Dataset for use in Earth System Models (GSDE) (Shangguan *et al*, 2014). The K factor has been calculated at the resolution of 1 km before being regridded to 5 arcmin using the bilinear interpolation method. We also used the SOC concentration in the soil from GSDE, which was derived using the "aggregating first" approach, to compare to our SOC stocks from simulations with the emulator. Finally, the slope steepness factor (S), which was originally estimated at the resolution of 1 km, was also

371

regridded to the resolution of 5 arcmin using the bilinear interpolation method.

Using the above-mentioned data, soil erosion rates were first calculated at the resolution of 5 arcmin, and afterwards aggregated to the coarse resolution of the emulator (2.5° x 3.75° ) to calculate daily SOC erosion rates.

**2.5 Model simulations**

To be able to understand and estimate the different direct and indirect effects of soil erosion on the SOC dynamics,

378

we propose a factorial simulation framework (Fig. 3 and Table 1). This framework allows isolating or combining the main processes that link soil erosion to the SOC pool, namely the influence from climate variability, LUC, and atmospheric $CO_2$ increase. The different model simulations described in this section will be based on this framework.

We performed two different simulations with the full ORCHIDEE model to produce the required data input for the emulator for the period 1850-2005. For this we first performed a spinup with ORCHIDEE to get stead-state carbon pools for the year 1850. We chose the period 1850-2005 based on the ISIMIP2b precipitation data availability and

385

the fact that this period underwent a significant intensification in agriculture globally and a substantial rise in atmospheric $CO_2$ concentrations. In the first simulation of ORCHIDEE the global atmospheric $CO_2$ concentration was fixed to the year 1850 to calculate time-varying NPP not impacted by CO2 fertilization and subsequent carbon pools, while in the second simulation the atmospheric $CO_2$ concentration was made variable. In both simulations, climate is variable from CRU-NCEP (Fig. 3).

Furthermore, we performed  7 simulations with the Adj.RUSLE model to pre-calculate the soil erosion rates that will be used as input to the ORCHIDEE emulator. Three of the 7 erosion simulations used best estimates for

392

each model parameter, and the rest used either the minimum or maximum values for the R- and C-factors to derive an uncertainty range for our soil erosion rates and to analyze the sensitivity of the emulator.  In the first simulation with the best estimated model parameters we kept the climate and land cover variable through time (the "CC+LUC" simulation). In the second simulation we only varied the climate through time and kept land cover fractions fixed to 1850 (the "CC" simulation, Fig. 3). In the third simulation we only varied the land cover through time and kept the climate constant

**Comment [VN10]:**
**Reviewer #2:** *"Why this dataset? How does it compare to the HWSD and SoilGrids (Hengl et al. PLoS ONE 2014, 2017) datasets?"*

**Answer:** The GSDE is based on the SoilMap of the World (FAO, 1995, 2003) and various regional and national soil databases. It is available at a 1km resolution and at 5 arcmin resolution and contains updated soil information and more soil variables such as nutrients. The GSDE is based on several regional soil maps and is more up to date on soil information than the HWSD but both products compared relatively well since they shared several data (Shangguan et al., 2014). We did not test the SoilGrids data, which is based on a different approach. A recent publication showed that SoilGrids give different results compared to HWSD (Tifafi et al., 2018) but regarding the difference between the products we decided to use only one of them already used to evaluate erosion process and then be more comparable with previous publications (Naipal et al., 2015, 2016).

to the average cyclic variability of the period 1850-1859 (the "LUC" simulation, Fig. 3). The erosion simulations with either minimum or maximum model parameters were either a "CC+LUC" or a "CC" type of simulation. From the two simulations of ORCHIDEE (with variable and constant $CO_2$) and the 7 soil erosion simulations of the Adj.RUSLE, we constructed 4 versions of the emulator to perform 8 main simulations and 4 sensitivity simulations. The different simulations and their description are given in table 1 and figure 3. In the simulations without LUC (S2, S4, S6 and S8), the PFT fractions and the harvest index are constant and equal to those in the year 1850. In the simulations with LUC (S1, S3, S5 and S7) the harvest index increases and the PFT fraction change with time during 1850 - 2005. In each emulator-simulation we first calculated the equilibrium carbon stocks analytically before calculating the change of the carbon stocks in time depending on the perturbations during the transient period (1851-2005). In simulations with erosion, the equilibrium state of the SOC pools has been calculated using the average erosion rates of the period 1850-1859, assuming erosion to be constant before 1850 and a steady state condition where erosion fluxes are equal to new input from litter.

**3 Results**

**3.1 Erosion versus no erosion**

After including soil erosion in the ORCHIDEE emulator we obtain a total global soil loss flux of 47.6±10 Pg C $y^{-1}$ for the year 2005 of which 20 to 29% is attributed to agricultural land and 51 to 554% to grassland (natural grass and pasture). This global soil loss flux (here 'loss' meaning horizontal removal of SOC by erosion) leads to a total SOC loss flux of 0.52±0.1467 Pg C $y^{-1}$ of which 26 to 331% are attributed to agricultural land and 54 to 640% to grassland (CTR, Fig 4). Grassland and agricultural land thus have much larger annual average soil and SOC erosion rates compared to forest (Table 2).

The total soil and SOC losses in the year 2005 are show an increase of 11-194% and 23-356.5%, respectively, compared to 1850 (CTR, Fig. 4) with the largest increases found in the tropics (Fig. 5B, D). The largest increase in soil and SOC erosion during 1850 – 2005 is found in South-America (Table 3) despite of the significant decreases in simulated precipitation leading to less intense erosion rates in this region. One should keep in mind that due to uncertainties in the simulated LUC and climate variability for certain regions and the assumptions made in our modeling framework, these trends in soil and SOC erosion rates are linked to some uncertainty. However, it is difficult to assess this uncertainty, mainly due to the lack of observations for the past in regions such as the tropics and the lack of model-testing in these regions.

We found that the total soil erosion flux on agricultural land increased withby 55-58% almost doubled byin the year 2005 compared to 1850, while the SOC erosion flux increased by withby only 11-7062% (Fig. 4) and led to a cumulative SOC removal of 22±57 Pg C on agricultural land since 1850 (CTR). On pasture land and grassland, the soil erosion flux increased by only with 8-20.5%, while the SOC erosion flux increased withby 44-5434% (Fig. 4) and led to a cumulative SOC mobilization of 38±752 Pg C since 1850. It is evident that on agricultural land the uncertainty range inof soil erosion leads to a large uncertainty range in SOC erosion compared to grassland. The

**Comment [VN11]:**
**Reviewer #2:** How uncertain are these numbers given the model formulation assumptions, land-use maps, and methods used? It would help to see a sensitivity analysis and some uncertainty ranges.

**Answer:** We performed 4 additional simulations with a different PFT map, which is also based on the LUH2 land use dataset but where the historical forest area change that is not constrained by data as done by Peng et al. (2017). We used these simulations to show the differences to our results when other land use maps are used. If the forest is not constrained with methods described by Peng et al. (2017), there is a stronger decrease in forest area over the period 1850-2005. Also the grassland shows an increasing trend, while in the PFT map with constrained forest the grassland shows globally a slight decreasing trend. In the rest of the text we will refer to the PFT map constrained with data on forest area as the 'constrained PFT map' and to the other PFT map as the 'unconstrained PFT map'. Differences in global average soil erosion rates between the different PFT maps are small, however, there are significant differences in the SOC erosion rates and cumulative changes in SOC stocks during the transient period. According to the unconstrained PFT map, soil erosion leads to a total SOC removal of 79 Pg (simulation S1 with the new vertical discretization scheme) over the period 1850-2005, which is 6Pg larger than the total SOC removal by soil erosion under the constrained PFT map. Interestingly, according to the unconstrained PFT map, the global cumulative SOC stock change over 1850-2005 under soil erosion and LUC (S1) is 60% smaller than the stock derived using the constrained PFT map. This is most likely due to the higher forest area at the start of the period A850-2005, leading to a larger increase in SOC stocks by increasing atmospheric CO2 concentrations. The global LUC effect on the SOC stocks of both PFT maps is found to be similar. For more details and our changes in the manuscript see our answer to comment 2 of reviewer 1.
We also derived an uncertainty range for our soil erosion rates. First, we varied the R-factor of the Adj.RUSLE model between a maximum and a minimum based on the regression equations derived by Naipal et al. (2015) per climate zone. Then we varied the C-factor of the Adj.RUSLE model between a maximum and minimum value per land cover type (tree, crop or grass) based on literature. We then used the uncertainty range in the C and R factors to derive the uncertainty range in the soil erosion rates and subsequently in the SOC erosion rates. We performed 4 additional simulations with the emulator, 2 simulations with the setup of S1 and a minimum and maximum soil erosion scenario, and 2 simulations with the setup of S2 with a maximum and minimum soil erosion scenario.

[revised manuscript text omitted]

**Comment [VN13]:**
**Reviewer #2:** *"The emulator used in this study seems to have various limitations that make the numbers presented quite uncertain – further discussion on, and quantification of, these uncertainties is warranted and would greatly improve this manuscript. Specifically, I would have liked to see additional support for the SOC model formulation, parameters, and built-in feedbacks chosen for the emulator, as well as support for its vertical discretization and parameterization. The carbon emulator is supposed to describe the carbon pools and fluxes exactly as in ORCHIDEE, yet the total global SOC stocks from the emulator are 44% higher than that of the original ORCHIDEE model. This is a big difference. What does this tell us about the accuracy and applicability of the emulator, and how do the SOC stocks of the two models compare to the Harmonized World Soil Database (HWSD) and other global SOC databases? Additional major comments/questions, especially those regarding the assumptions and methods used, are detailed below."*

**Answer:** We modified the vertical discretization scheme of the emulator in such a way that the total SOC stock of each grid cell, PFT and C pool is close to that of ORCHIDEE when soil erosion and land use change is deactivated (0.5% max difference in total global SOC stock). For this we calibrated both the exponent '*re*' and variable '$k_{0i}$' of equation 8 in the manuscript for each grid cell and PFT under equilibrium conditions, such that the total soil respiration per grid cell, PFT, and soil C pool of the emulator would be similar to that of the ORCHIDEE model. For the transient period (1850-2005), we made '*re*' remain equal to the equilibrium state values, while values for '$k_{0i}$' were derived at a daily time-step to keep to SOC stocks of the emulator similar to those of ORCHIDEE and preserve the yearly variability in the soil respiration rates due to changes in soil climate (soil erosion and land use change were deactivated ). Details of how we calibrated the exponent '*re*' and variable '$k_{0i}$' we describe in our response to Reviewer 1**.** The ...

**Comment [VN14]:**
**Reviewer #2:** *"*(Section 4.4) with all of these model limitations, it would be nice to have a rough quantification of uncertainties.*"*

**Answer:** We agree with the reviewer that quantifying the uncertainty is important. Therefore, we derived an uncertainty range for our soil erosion rates. First, we varied the R-factor of the Adj.RUSLE model between a maximum and a minimum based on the regression equations derived by Naipal et al. (2015) per climate zone. Then we varied the C-factor of the Adj.RUSLE model between a maximum and minimum value per land cover type (tree, crop or grass) based on literature. We then used the uncertainty range in the C and R factors to derive the uncertainty range in the soil erosion rates and subsequently in the SOC erosion rates. We performed 4 additional simulations with the emulator, 2 simulations with the setup of S1 and a minimum and maximum soil erosion scenario, and 2 simulations with the setup of S2 with a maximum and minimum soil erosion scenario. The results can be found in chapter 3 
[revised manuscript text omitted]

**Comment [VN15]:**
**Reviewer #1:** "Also, intense soil erosion is typically found in mountainous areas where climate variability has significant impacts, while at the same time these regions are usually poor in SOC." It's not clear in the manuscript whether or not ORCHIDEE has topography information? In another word, if ORCHIDEE simulates a low SOC stock over the grid cells that have mountains, is that because of the topographical feature of this gridcell can not hold a lot of SOC in ORCHIDEE? Or because of other reasons such as climate constraints (e.g., colder in mountain area)?"

**Answer:** ORCHIDEE has no soil depth information and thus cannot simulate low SOC stocks due to the fact that the gridcell cannot hold a lot of SOC. Low SOC might however be a result of the plant productivity, the climate (temperature and precipitation), soil moisture and clay content (which is a constant variable). ORCHIDEE has, however, topographical information such as slope that determines the flow directions for water/runoff and affects hydrological parameters such as soil moisture content.

**Comment [VN16]:**
**Reviewer #1:** "CO2 fertilization effects on NPP is not fully convincing here, because ORCHIDEE does not have nutrient constraints. OCN might be a better surrogate model to be able to say something about CO2 fertilization effect on NPP."

**Answer:** In the ORCHIDEE model version we used the nutrients are indeed absent. Our intention, however, was to show the complete picture of possible direct and indirect interactions of soil erosion with the C cycle with the current model setup. The representation of nutrients in global land surface models is new and the related uncertainties are not well quantified. We work with a more or less simple version of ORCHIDEE and the C emulator to be able to understand and quantify the effects of soil erosion on the C cycle. We mention the uncertainties due to he absence of nutrients in the next chapter.

[revised manuscript text omitted]

**Comment [VN17]:**
**Reviewer #1:** *"Land use change map. The LUC is prescribed by PFT fractional change derived from Peng 2017. Wondered how this LUC dataset differs from Land-Use Harmonization (LUH2), the new CMIP6 land use change dataset. Given that LUC is a dominant factor of SOC erosion, I am curious about the uncertainty of SOC erosion, induced by using different LUC estimate (e.g., Peng 2017 vs LUH2)."*

**Answer:** The PFT fractional map is based on LUHv2 land use dataset, historical forest area data from Houghton (for large regions) and present day forest area from ESA CCI satellite land cover data (Peng et al., 2017). The historical forest data from Houghton and the latest satellite land cover data from ESA are the best estimates that currently exist on forest area. Figure S… shows that if the forest is not constrained with methods described by Peng et al. (2017), there is a stronger decrease in forest area over the period 1850-2005. Also the grassland shows an increasing trend, while in the PFT map with constrained forest the grassland shows globally a slight decreasing trend. In the rest of the text we will refer to the PFT map constrained with data on forest area as the 'constrained PFT map' and to the other PFT map as the 'unconstrained PFT map'.
We agree with the reviewer that different land use data can result in large uncertainties in both SOC stocks and soil erosion rates. To show the potential uncertainty in our results due to uncertainties in underlying land use data we performed 4 additional simulations (S1 to S4) using the unconstrained PFT map and the new vertical discretization scheme. The results are described here in the revised manuscript.

[revised manuscript text omitted]

**Comment [VN19]:** No significant differences to figures 4-9 due to changes in the vertical discretization scheme

**Comment [VN20]:**
**Reviewer #1: "** Figure 4. I do not fully understand why climate change either decrease or not change erosion?

**Answer:** With climate change we mean temperature and precipitation changes. For soil erosion only precipitation changes are of interest. Globally we find that average yearly precipitation shows a slightly decreasing trend over the period 1950 – 2005 according to the ISIMIP2b dataset used to calculate soil erosion rates. A global smaller total precipitation with respect to 1850 AD will lead to smaller soil erosion rates when LUC is not included. The decrease in total precipitation over land is mostly coming from the tropics, where due to large precipitation amounts a change in precipitation can alter soil erosion significantly. At the same time precipitation is very variable and might not lead to a significant global net change in soil erosion rates over the total period 1850-2005. This result might be contradictory to the fact that major soil erosion events are caused by storms. But in our case we model only rill and interril erosion, which is usually a slow process and previous studies have shown that land use change is usually the main driver of behind accelerated rates of this type of soil erosion. Furthermore, there are very few studies that have quantified the individual effects of precipitation change versus land use change on soil erosion rates over a sufficiently long time period. Therefore, it is difficult to verify this result. However, our soil erosion model performs well for present-day and therefore any possible biases here could be mainly related to biases in precipitation rates, and soil parameters. We agree that this is an interesting point raised by the reviewer and added some additional sentences explaining the trend in beginning of chapter 4.2.

Fig 5: (A) Average annual soil erosion rates at a 5 arcmin resolution in the year 2005, (B) change in average annual soil erosion rates over the period 2005-1850, (C) average annual SOC erosion rates at a resolution of 2.5x3.75 degrees in 2005, (D) change in average annual SOC erosion rates over the period 2005-1850, and ( E ) difference in SOC stocks at a resolution of 2.5x3.75 degrees between the year 2005 and 1850 (CTR simulation). For the SOC

980      stocks positive values (green color) indicate a gain, while negative values (red color) indicating a loss. For the erosion rates positive values (red color) indicate an increase over 1850 - 2005, while negative values (green color) indicate a decrease over 1850 - 2005

[Figure]

Figure 6: Cumulative SOC stock changes during 1850 – 2005 for (A) simulations with variable atmospheric $CO_2$ concentration,  and (B) for simulations with a constant $CO_2$ concentration, implied by variable land cover alone (dash-dotted lines), by variable climate (dashed lines), and variable land cover and climate (straight lines), without erosion (black lines) and with erosion (red lines).

[Figure]

987    Figure 7: Cumulative SOC stock changes per PFT during 1850 – 2005 implied by variable land cover, climate and CO$_2$, without erosion (grey colors) and with erosion (red colors).

Figure 8: A) Difference between the changes of SOC stocks over the period 1850-2005 under all perturbations including soil erosion and the changes in SOC stocks excluding soil erosion, S1-S3, B) Difference between the changes of SOC stocks under LUC including soil erosion and the changes in SOC stocks excluding soil erosion (S1-S2)-(S3-S4), C) Difference between the changes of SOC stocks under a variable climate and CO$_2$ increase including

994    soil erosion and the changes in SOC stocks excluding soil erosion, S2-S4.

[Figure]

Figure 9: Contribution to the cumulative global SOC stock change over 1850-2005 by $CO_2$ fertilization (red), effect of precipitation and temperature change on the carbon cycle (dark blue), effect of precipitation change on soil erosion ( aqua), LUC effect on the carbon cycle (dark green), and LUC effect on soil erosion (light green)